# Multiway Multislice PHATE:
# Visualizing Hidden Dynamics of RNNs through Training

**Jiancheng Xie**                                                        *xiejc@bu.edu*
*Department of Biomedical Engineering*
*Boston University*

**Lou C. Kohler-Voinov**                                    *lou.kohlervoinov@epfl.ch*
*Department of Life Sciences*
*Ecole Polytechnique Fédérale de Lausanne*

**Noga Mudrik**                                                      *nmudrik1@jhu.edu*
*Department of Biomedical Engineering, Kavli NDI, Center for Imaging Science*
*Johns Hopkins University*

**Gal Mishne**                                                          *gmishne@ucsd.edu*
*Halıcıoğlu Data Science Institute*
*University of California San Diego*

**Adam S. Charles**                                                      *adamsc@jhu.edu*
*Department of Biomedical Engineering, The Mathematical Institute for Data Science, Center for Imaging Science*
*Johns Hopkins University*

**Reviewed on OpenReview:** *https: // openreview. net/ forum? id= 9Yr4V7iZsq*

## Abstract

Recurrent neural networks (RNNs) are a widely used tool for sequential data analysis; however, they are still often seen as black boxes. Visualizing the internal dynamics of RNNs is a critical step toward understanding their functional principles and developing better architectures and optimization strategies. Prior studies typically emphasize network representations only after training, overlooking how those representations evolve during learning. Here, we present Multiway Multislice PHATE (MM-PHATE), a graph-based embedding method for visualizing the evolution of RNN hidden states across the multiple dimensions spanned by RNNs: time, training epoch, and units. Across controlled synthetic benchmarks and real RNN applications, MM-PHATE preserves hidden-representation community structure among units and reveals training-phase changes in representation geometry. In controlled synthetic systems spanning multiple bifurcation families and smooth state-space warps, MM-PHATE recovers qualitative dynamical progression while distinguishing family-level differences. In task-trained RNNs, the embedding identifies information-processing and compression-related phases during training, and time-resolved geometric and entropy-based summaries align with linear probes, time-step ablations, and label–state mutual information. These results show that MM-PHATE provides an intuitive and comprehensive way to inspect RNN hidden dynamics across training and to better understand how model architecture and learning dynamics relate to performance.

## 1 Introduction

Recurrent neural networks (RNNs) are designed to handle sequential data by modeling input sequences and retaining memory of past elements through recurrent connections or memory units (Lipton et al., 2015; Kaur & Mohta, 2019). Unlike feedforward neural networks (FNNs), which process each input independently, RNNs

update their internal state dynamically, enabling them to capture contextual relationships across sequences. This makes RNNs particularly effective for tasks that rely on the order and relationships between elements, such as time-series analysis and action recognition (Hewamalage et al., 2021).

Since gaining popularity in the 1990s, various RNN variants and training strategies have been developed to improve training stability and long-range dependency learning. Architectures such as Long Short-Term Memory (LSTM) networks (Hochreiter & Schmidhuber, 1996) and Structurally Constrained Recurrent Networks (SCRN)(Mikolov et al., 2014) were designed to address challenges like the vanishing gradient problem(Salehinejad et al., 2018). Additionally, RNNs excel at handling irregular or incomplete sequential data due to their flexibility with variable-length inputs. These properties, along with ongoing architectural improvements (Salehinejad et al., 2018), have enabled RNNs to achieve exceptional performance across domains such as natural language processing (NLP), neuroscience, and biomedical signal processing (Barak, 2017; Chen & Li, 2021; Khalifa et al., 2021). Notably, RNNs remain the state-of-the-art for neural decoding in intracortical brain-computer interfaces (Deo et al., 2024).

Despite their extensive use, the learning dynamics and internal representations of RNNs remain difficult to interpret. This opacity complicates performance evaluation, model design, and parameter selection, hindering the development of more effective architectures. While significant progress has been made in interpreting FNNs (Wang et al., 2021; Yu et al., 2014), similar advances for RNNs have been limited. Most RNN analyses focus on either fixed-point analysis in dynamical systems (Sussillo & Barak, 2013) or performance evaluations across different architectures and components (Chung et al., 2014; Greff et al., 2017). Consequently, there is a clear need for new methods that facilitate the interpretation of RNNs' latent representations during training.

In explainable deep learning, dimensionality reduction is widely used to visualize high-dimensional data, helping researchers gain intuition about its structure and relationships (Karpathy et al., 2015; Hidaka & Kurita, 2017; Rauber et al., 2016; Gigante et al., 2019). However, existing techniques often have limitations. Methods like t-SNE emphasize local structure at the expense of global patterns (Maaten & Hinton, 2008), while methods such as PCA and Isomap focus on global structures and may miss important local details (Maćkiewicz & Ratajczak, 1993; Tenenbaum et al., 2000). These methods can also be sensitive to noise and outliers (Moon et al., 2019). Visualizing RNNs is particularly challenging since they require preserving structures across multiple dimensions, including time, epochs, and hidden units. Consequently, traditional dimensionality reduction techniques often fail to capture the complexity of RNN learning dynamics.

To address these challenges, Gigante et al.(Gigante et al., 2019) introduced Multislice PHATE (M-PHATE), which visualizes FNN hidden states during training. M-PHATE constructs a multislice graph where each slice represents the network's state at a particular training epoch, capturing both inter-epoch relationships and community structures through PHATE(Moon et al., 2019). This approach effectively captures performance-related features, such as task-related specialization, without requiring external validation data, making it particularly useful in data-limited settings.

However, M-PHATE is designed for FNNs and does not account for the sequential nature of RNNs, where hidden states across all time-steps play a critical role in representation learning (Su & Shlizerman, 2020). To address this limitation, we propose Multiway Multislice PHATE (MM-PHATE), which extends M-PHATE by capturing RNN hidden states across both time-steps and epochs. MM-PHATE provides a comprehensive view of RNN learning dynamics, enabling us to track how hidden representations evolve during training and how they support task performance. These training dynamics are reminiscent of ideas from information bottleneck theory, where intermediate representations are encouraged to retain task-relevant information about the target while discarding irrelevant variability in the input (Fischer, 2020; Tishby & Zaslavsky, 2015; Cheng et al., 2019). In contrast to classical information bottleneck approaches, we do not optimize or estimate a formal bottleneck objective; instead, we empirically characterize "expansion" and "compression" phases via the geometry and entropy of hidden-state trajectories revealed by MM-PHATE. We evaluate MM-PHATE on controlled synthetic benchmarks and real RNN applications, including synthetic systems spanning multiple bifurcation families and smooth state-space warps, and compare against existing methods such as PCA (Wold et al., 1987), t-SNE (Maaten & Hinton, 2008), Isomap (Tenenbaum et al., 2000), Locally Linear Embedding (LLE) (Roweis & Saul, 2000), UMAP (McInnes et al., 2020), and M-PHATE.

Our main contributions are as follows:

- We introduce MM-PHATE, a multiway multislice visualization framework for RNN hidden dynamics across time-steps and training epochs, providing a unified view of representation evolution during learning.

- We evaluate MM-PHATE on controlled synthetic benchmarks (Hopf and Pitchfork bifurcations, with and without smooth state-space warps), showing that it recovers qualitative dynamical progression and preserves family-level distinctions under the tested conditions.

- We show on task-trained RNNs that MM-PHATE preserves hidden-unit community structure and reveals time-resolved representational changes whose geometric and entropy-based summaries align with linear probes, time-step ablations, and label–state mutual information.

## 2 Related Work

Existing methods for interpreting RNNs can be categorized into performance-oriented and application-oriented post-training analyses. Performance-oriented approaches focus on evaluating network-level performance by comparing architectural components and training parameters. For example, Chung et al. (2014) compared gated RNNs (e.g., GRUs and LSTMs), while Greff et al. (2017) conducted a detailed analysis of LSTM components. However, these studies primarily emphasize performance outcomes and provide limited insights into the hidden state dynamics and learned representations within RNNs.

In contrast, application-oriented approaches focus on visualizing and interpreting hidden state activations after training, often in the context of specific tasks. In NLP, Karpathy et al. (2015) overlaid activation maps on texts, revealing interpretable unit behaviors such as tracking text structure. Li et al. (2016) used saliency heat maps to identify critical words in learned representations, while Strobelt et al. and Ming et al. developed interactive tools to correlate hidden state patterns with phrases (Strobelt et al., 2018; Ming et al., 2017). Similar techniques have been applied to domains such as speech recognition (Tang et al., 2017), earth sciences (Titos et al., 2022), and medical applications (Kwon et al., 2019). While these studies provide intuitive, task-specific insights, they often lack generalizability and do not capture training dynamics over time.

Other works have explored general RNN behavior using techniques such as Proper Orthogonal Decomposition (POD) to analyze Seq2Seq internal states (Su & Shlizerman, 2020) and PCA to link recurrent activations to model generalization (Farrell et al., 2022). However, these approaches focus on post-training analysis and do not visualize how hidden states evolve across both time-steps and epochs during training. To our knowledge, no existing methods provide a unified view of RNN hidden dynamics over temporal and training dimensions.

## 3 Background

**PHATE:** PHATE is a data visualization technique that can capture both the local and global structure of data using diffusion processes (Moon et al., 2019). The PHATE algorithm optimizes the diffusion kernel (Coifman & Lafon, 2006) for the visualization of high-dimensional data. Let $\boldsymbol{x}_i$ be a point in a high-dimensional dataset. PHATE begins by computing the Euclidean distance matrix $\boldsymbol{E}$ between all data points, where $\boldsymbol{E}_{ij} = \|\boldsymbol{x}_i - \boldsymbol{x}_j\|_2$. These distances are then transformed into affinities using an adaptive $\alpha$-decay kernel $\boldsymbol{K}_{k,\alpha}(\boldsymbol{x}_i, \boldsymbol{x}_j) = \frac{1}{2}\exp\left(-\left(\frac{\boldsymbol{E}_{ij}}{\epsilon_k(\boldsymbol{x}_i)}\right)^\alpha\right) + \frac{1}{2}\exp\left(-\left(\frac{\boldsymbol{E}_{ij}}{\epsilon_k(\boldsymbol{x}_j)}\right)^\alpha\right)$, which adapts to the data density around each point and captures local information. The parameters $\epsilon_k(x_i)$ and $\epsilon_k(x_j)$ are the $k$-nearest-neighbor distance of $x_i$ and $x_j$, and $\alpha$ controls the decay rate. The affinities are then row-normalized to obtain the diffusion operator $\boldsymbol{P} = \boldsymbol{D}^{-1}\boldsymbol{K}_{k,\alpha}$ that represents the single-step transition probabilities between data points, where $\boldsymbol{D}$ is a diagonal matrix whose entries are row sums of $\boldsymbol{K}_{k,\alpha}$. PHATE calculates the information distance between points based on their transition probabilities: $\text{dist}_{ij} = \sqrt{\|\log \boldsymbol{P}_i^t - \log \boldsymbol{P}_j^t\|^2}$, where $\boldsymbol{P}^t$ captures the transition probabilities of a diffusion process on the data over $t$ steps and $i$ and $j$ are rows in the matrix. These distances are embedded into low dimensions using Multidimensional Scaling (MDS) (Ramsay,

1966) for visualization. Local and global distances within the data's manifold are represented in PHATE by multistep diffusion probabilities. The diffusion probability of each point captures its local context, enabling pairwise comparisons between all points (both neighboring and distant points) that represent the entire global context. For further details, see (Moon et al., 2019). We will use PHATE to embed RNN training dynamics, however we alter the initial graph construction to emphasize certain structures in the data we wish to visualize.

**M-PHATE:** Gigante et al. (2019) model the evolution of the hidden units in a feedforward neural network and their community structure using a multislice graph. Each slice corresponds to the network at an epoch during training, and the collection of graphs represents the dynamical system resulting from the evolution of the network's hidden states. Let $\boldsymbol{F}$ be an FNN with a total of $m$ hidden units, and let $\boldsymbol{F}^{(\tau)}$ be the representation of the network after being trained for $\tau \in \{1, ..., n\}$ epochs on the training data $\boldsymbol{X}$ sampled from a larger dataset $\boldsymbol{\Pi}$. The algorithm first calculates a shared feature space using the normalized activations of all hidden units $i \in \{1, \ldots, m\}$ on the input data, as a 3-dimensional tensor:

$$\boldsymbol{T}(\tau, i, k) = \frac{\boldsymbol{F}_i^{(\tau)}(\boldsymbol{Y}_k) - \frac{1}{p}\sum_{\ell} \boldsymbol{F}_i^{(\tau)}(\boldsymbol{Y}_\ell)}{\sqrt{\mathrm{Var}_\ell[\boldsymbol{F}_i^{(\tau)}(\boldsymbol{Y}_\ell)]}},$$

where $\boldsymbol{F}_i^{(\tau)}(\boldsymbol{Y}_k) : \mathbb{R}^d \to \mathbb{R}$ denotes the activation of the $i$-th hidden unit of $\boldsymbol{F}$ for the $k^{th}$ sample from input data $\boldsymbol{Y}$. Here, $\boldsymbol{Y}$ is a subset of $p$ samples from the $d$-dimensional training data $\boldsymbol{X}$, with an equal number of samples from each input class, and $p \ll |\boldsymbol{X}|$. This activation tensor $\boldsymbol{T}$ is then used to calculate intraslice affinities between pairs of hidden units within an epoch $\tau$ during the training, as well as the interslice affinities between a hidden unit $i$ and itself at different epochs:

$$\boldsymbol{K}_{\text{intraslice}}^{(\tau)}(i, j) = \exp\left(\frac{-\|\boldsymbol{T}(\tau, i) - \boldsymbol{T}(\tau, j)\|_2^\alpha}{\sigma_{(\tau, i)}^\alpha}\right)$$

$$\boldsymbol{K}_{\text{interslice}}^{(i)}(\tau, \upsilon) = \exp\left(\frac{-\|\boldsymbol{T}(\tau, i) - \boldsymbol{T}(\upsilon, i)\|_2^2}{\epsilon^2}\right)$$

where $\alpha$ is the $\alpha$-decay parameter, $\sigma_{(\tau, i)}$ is the intraslice bandwidth for unit $i$ in epoch $\tau$, and $\epsilon$ is the fixed interslice bandwidth. These matrices are combined to form an $nm \times nm$ multislice kernel matrix $\boldsymbol{K}$, which is then symmetrized, row-normalized, and visualized using PHATE in 2D or 3D.

## 4  Multiway Multislice PHATE

M-PHATE was shown to be a powerful tool for visualizing FNNs. However, to effectively visualize the evolution of RNNs' hidden representations, we need to consider hidden state dynamics across *time-steps* within the sequence and training epochs concurrently (Su & Shlizerman, 2020). In RNNs, the output from previous *time-steps* is fed as an input to current *time-steps*. This is useful in the treatment of sequences and building a memory of the previous inputs into the network. The network iteratively updates a hidden state $h$. At each *time-step* $t$, the next hidden state $h_{t+1}$ is computed using the input $x_t$ and the current hidden state $h_t$. Importantly, the network uses the same weights $W$ and biases $b$ for each *time-step*. Thus the output $y_t$ at *time-step* $t$ is $y_t = f(W \cdot h_t + b)$, where $f$ is some activation function.

Let $\boldsymbol{R}^{(\tau)}$ be the representation of an $m$-unit RNN after being trained for $\tau \in \{1, \ldots, n\}$ epochs on the training data $\boldsymbol{X} \subset \boldsymbol{\Pi}$. We denote $\boldsymbol{R}_{i,w}^{(\tau)}(\boldsymbol{Y}_k) : \mathbb{R}^d \to \mathbb{R}$ the activation of the $i$-th hidden unit of $\boldsymbol{R}$ at time-step $w \in \{1, \ldots, s\}$ in epoch $\tau$ for the $k^{th}$ sample of $\boldsymbol{Y}$, where $\boldsymbol{Y}$ consists of $p$ samples from the training data $\boldsymbol{X}$. We construct the 4-way tensor $\boldsymbol{T}$ using the hidden unit activations as a shared feature space, which we use to calculate unit affinities across all epochs and time-steps. The tensor $\boldsymbol{T}$ is an $n \times s \times m \times p$ tensor containing the activations at each epoch $\tau \in \{1, \ldots, n\}$ and time-step $w \in \{1, \ldots, s\}$ of each hidden unit $\boldsymbol{R}_i$ ($i \in \{1, \ldots, m\}$) with respect to each sample $\boldsymbol{Y}_k \subset \boldsymbol{X}$. To eliminate the variability in $\boldsymbol{T}$ due to the bias term

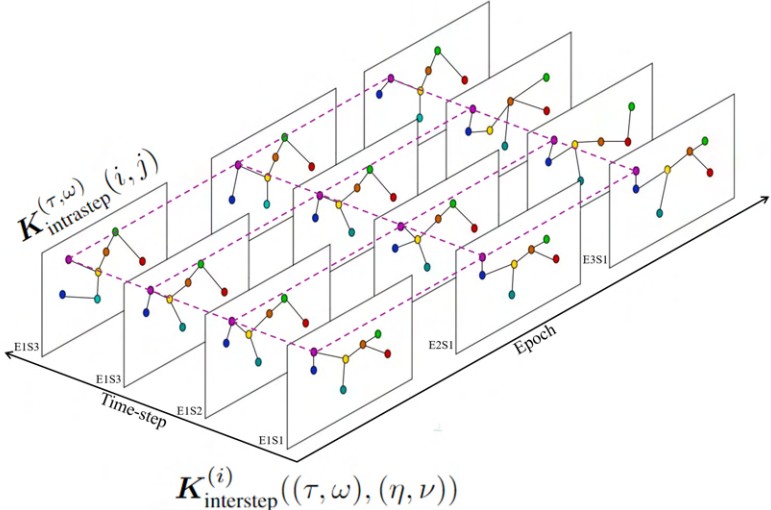

Figure 1: Example schematic of the multiway multislice graph used in MM-PHATE for RNNs. The intra-step kernels represent the similarities between the graph nodes at the same time-steps. The inter-step kernels represent the similarities between the nodes and themselves at different time-steps and epochs.

$b$, we $z$-score the activation of each hidden unit at time-step $\omega$ and epoch $\tau$:

$$\boldsymbol{T}(\tau, \omega, i, k) = \frac{\boldsymbol{R}_{i,\omega}^{(\tau)}(\boldsymbol{Y}_k) - \frac{1}{p}\sum_\ell \boldsymbol{R}_{i,\omega}^{(\tau)}(\boldsymbol{Y}_\ell)}{\sqrt{Var_\ell[\boldsymbol{R}_{i,\omega}^{(\tau)}(\boldsymbol{Y}_\ell)]}}. \tag{1}$$

We construct a kernel over $\boldsymbol{T}$ utilizing our prior knowledge of the temporal aspect of $\boldsymbol{T}$ to capture its dynamics over epochs and time-steps. This constructed kernel, denoted $\boldsymbol{K}$, represents the weighted edges in the multislice graph of the hidden units (Fig. 1). In this representation, each unit has two types of connections: edges between the unit to itself across epochs and time-steps and, within a fixed epoch and time-step, edges between a unit and its community—the other units which have the most similar representation. The edges are weighted by the similarity in activation pattern. We define $\boldsymbol{K}$ as a $nsm \times nsm$ kernel matrix between all $m$ hidden units at all $s$ time-steps in all $n$ training epochs. The $((\tau-1)sm + (\omega-1)m + j)_{th}$ row or column of $\boldsymbol{K}$ refers to the $j_{th}$ unit at time-step $\omega$ in epoch $\tau$. We henceforth refer to the row as $\boldsymbol{K}((\tau, \omega, j), :)$ and the column as $\boldsymbol{K}(:, (\tau, \omega, j))$. In order to capture the evolution of hidden units of $R$ across time-steps and epochs, while preserving the unit's community structure, we construct a multiway multislice kernel matrix reflecting two types of connections simultaneously. Given the $\alpha$-decay parameter $\alpha$, the intra-step bandwidth for unit $i$ at time-step $\omega$ and epoch $\tau$: $\sigma_{(\tau, \omega, i)}$, and the fixed inter-step bandwidth $\epsilon$, we define:

- Intra-step affinities between hidden units $i$ and $j$ at time-step $\omega$ in epoch $\tau$:

$$\boldsymbol{K}_{\text{intra-step}}^{(\tau, \omega)}(i, j) = \exp\left(-\frac{\parallel \boldsymbol{T}(\tau, \omega, i) - \boldsymbol{T}(\tau, \omega, j) \parallel_2^\alpha}{\sigma_{(\tau, \omega, i)}^\alpha}\right)$$

- Inter-step affinities between a hidden unit $i$ and itself at different time-steps and epochs:

$$\boldsymbol{K}_{\text{inter-step}}^{(i)}((\tau, \omega), (\eta, \nu)) = \exp\left(-\frac{\parallel \boldsymbol{T}(\tau, \omega, i) - \boldsymbol{T}(\eta, \nu, i) \parallel_2^2}{\epsilon^2}\right)$$

The bandwidth $\sigma_{(\tau, \omega, i)}$ of the $\alpha$-decay kernel is set to be the distance of unit $i$ at time-step $\omega$ from epoch $n$ to its $k^{th}$ nearest neighbor across units at that time-step and epoch: $\sigma_{(\tau, \omega, i)} = d_k(\boldsymbol{T}(\tau, \omega, i), \boldsymbol{T}(\tau, \omega, :))$,

where $d_k(z, Z)$ denotes the $\ell_2$ distance from $z$ to its $k^{th}$ nearest neighbor in $Z$. We used $k = 5$ in all the results presented. The use of this adaptive bandwidth means the kernel is not symmetric and thus requires symmetrization. In the inter-step affinities $\boldsymbol{K}_{\text{inter-step}}^{(i)}$, we use a fixed-bandwidth Gaussian kernel $\epsilon = \frac{1}{nsm} \sum_{\tau=1}^{n} \sum_{\omega=1}^{s} \sum_{i=1}^{m} d_k(\boldsymbol{T}(\tau, \omega, i), \boldsymbol{T}(:, :, i))$, the average across all time-steps in all epochs and all units of the distance of unit $i$ at time-step $t$ to its $k_{th}$ nearest neighbor among the set consisting of the same unit $i$ at all steps.

The combined kernel matrix of these two matrices contains one row and column for each unit at each time-step in each epoch, such that the intra-step affinities form a block diagonal matrix and the inter-step affinities form off-diagonal blocks composed of diagonal matrices (Fig. A.1).

$$\boldsymbol{K}((\tau, \omega, i), (\eta, \nu, j)) = \begin{cases} \boldsymbol{K}_{\text{intra-step}}^{(\tau,\omega)}(i, j) & \text{if } (\tau, \omega) = (\eta, \nu) \\ \boldsymbol{K}_{\text{inter-step}}^{(i)}((\tau, \omega), (\eta, \nu)) & \text{if } i = j \\ 0 & \text{otherwise} \end{cases} \tag{2}$$

We symmetrize this final kernel as $\boldsymbol{K}' = \frac{1}{2}(\boldsymbol{K} + \boldsymbol{K}^T)$, and row-normalize it to obtain $\boldsymbol{P} = \boldsymbol{D}^{-1}\boldsymbol{K}'$, where $\boldsymbol{D}$ is a diagonal matrix whose entries are row sums of $\boldsymbol{K}'$ and which $\boldsymbol{P}$ represents a random walk over all units at all time-steps in all epochs, where propagating from one state to another is conditional on the transition probabilities between time-step $\omega$ in epoch $\tau$ and time-step $\nu$ in epoch $\eta$. PHATE is applied to $\boldsymbol{P}$ to visualize the tensor $\boldsymbol{T}$ in two or three dimensions. This resulting visualization thus simultaneously captures information regarding the evolution of the units across both time-steps and epochs.

## 5 Results

We evaluate MM-PHATE in two stages. First, we use *controlled synthetic benchmarks* to test whether the embedding preserves known dynamical structure and remains informative under smooth state-space perturbations. This stage provides a ground-truth anchor for interpreting the geometry and entropy summaries used later. Second, we analyze *task-trained RNNs* (Area2Bump neural spiking Chowdhury et al. (2022) and HAR kinematics Reyes-Ortiz et al. (2012)) to assess whether MM-PHATE reveals learning-relevant representational changes across time-steps and training epochs.

Throughout, we compare MM-PHATE against standard embeddings (PCA, t-SNE, Isomap, LLE, UMAP). We additionally report M-PHATE as a qualitative baseline in Appendix A.4. However, because M-PHATE operates on a single time-slice and does not model within-sequence dynamics, it embeds a different mathematical object (units × epochs rather than units × time × epochs), making direct quantitative comparison inappropriate.

### 5.1 Controlled Synthetic Benchmarks: Bifurcation Families and Smooth State-Space Warps

Interpreting RNN training dynamics from a visualization is challenging because the latent geometry is typically unknown, and many embeddings are dominated by activation magnitude or coordinate parameterization rather than the underlying dynamics. To test whether an embedding preserves *dynamically meaningful* structure, we use controlled synthetic benchmarks built from low-dimensional bifurcation systems with known qualitative geometry.

We use two-dimensional canonical systems (Hopf and Pitchfork) to generate noisy hidden-unit readouts across epochs and time-steps (Appendix A.5). These benchmarks are designed to address two complementary questions:

1. whether the embedding distinguishes *dynamical-family differences* (Hopf vs. Pitchfork), and

2. whether the resulting geometric organization remains interpretable under smooth state-space warps.

We first use the Hopf bifurcation as an analytically interpretable *anchor benchmark* to validate geometric and entropy interpretations in a setting with known latent phases. We then use a broader synthetic suite to test family-level distinctions and warp perturbations.

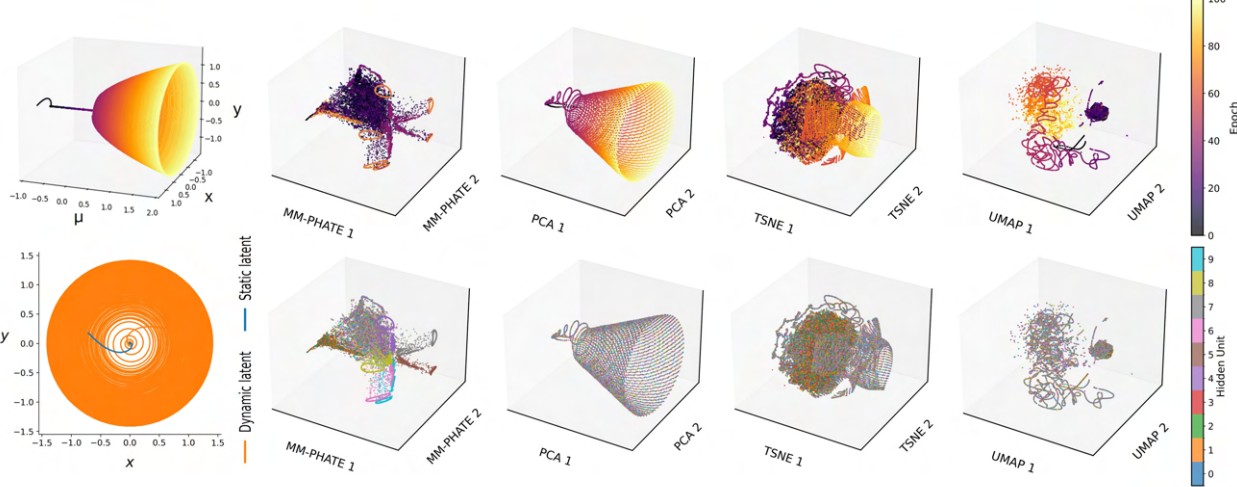

Figure 2: **Visualization of supercritical Hopf bifurcation dynamics and their embeddings.** The first column summarizes the synthetic construction used for the Hopf anchor benchmark: *top*—a representative latent Hopf trajectory for the swept-$\mu$ (dynamic) group, plotted in $(x, y, \mu)$ and colored by training epoch, illustrating the transition from a contracting fixed point ($\mu < 0$) to a stable limit cycle ($\mu > 0$) as $\mu$ crosses 0; *bottom*—the corresponding latent phase portrait in $(x, y)$, showing representative trajectories from the static control group ($\mu \equiv -1$ across epochs) alongside the same representative dynamic trajectory, colored by unit group (static vs. dynamic). The latent trajectories are then mapped through a scalar hidden-unit readout to simulate the model activations used for visualization. Remaining columns show embeddings produced by MM-PHATE, PCA, t-SNE, and UMAP, applied to the full hidden-state tensor and colored by epoch (top row) or unit identity (bottom row). MM-PHATE recovers the underlying bifurcation geometry as a single coherent post-bifurcation orbit, whereas conventional methods primarily reflect changes in activation magnitude.

**Hopf anchor benchmark: validating geometric and entropy interpretations.** In the supercritical Hopf system, the control parameter $\mu$ governs a transition from a contracting fixed point ($\mu < 0$) to loss of stability at $\mu = 0$, followed by convergence to a stable limit cycle for $\mu > 0$ with radius $r^*(\mu) = \sqrt{\mu}$. This gives a controlled analogue of training-phase motifs that often appear in RNNs: contraction, transition/expansion, and stabilization on a low-dimensional manifold. For this Hopf anchor benchmark, we use two groups generated from the same supercritical Hopf system but with different control-parameter schedules: a *dynamic* (swept-$\mu$) group with 6 units, for which $\mu$ is linearly swept from $-1$ to $2$ across epochs, and a *static* (fixed-$\mu$) control group with 4 units, for which $\mu \equiv -1$ at all epochs. Thus, only the dynamic group crosses the bifurcation, while the static group remains in the contracting fixed-point regime. This provides an internal control for distinguishing bifurcation-driven geometric reorganization from changes driven primarily by readout magnitude or coordinate effects.

Figure 2 shows this Hopf *dynamic-vs.-static* anchor benchmark and compares embeddings produced by MM-PHATE and representative baselines. Each point corresponds to a hidden unit at a specific time-step and epoch. MM-PHATE recovers the qualitative dynamical progression: post-bifurcation epochs align onto a coherent periodic orbit, epochs remain globally ordered, and units follow parallel trajectories on a shared manifold. Importantly, MM-PHATE deemphasizes amplitude growth as a nuisance dimension and organizes points by geometry and temporal progression of the underlying dynamical system.

By contrast, standard embeddings in the full comparison (PCA, t-SNE, Isomap, LLE, UMAP; see Appendix A.5.4) largely encode the increase in oscillation amplitude as $\mu$ grows, yielding families of concentric or inflated loops across epochs. In this regime, the dominant embedding axis reflects readout magnitude rather than the shared periodic topology, obscuring the transition and mixing static and dynamic units. When a tanh nonlinearity is applied to the readout, these variance-driven distortions become more pronounced due to saturation, whereas MM-PHATE remains qualitatively stable (Fig. A.24).

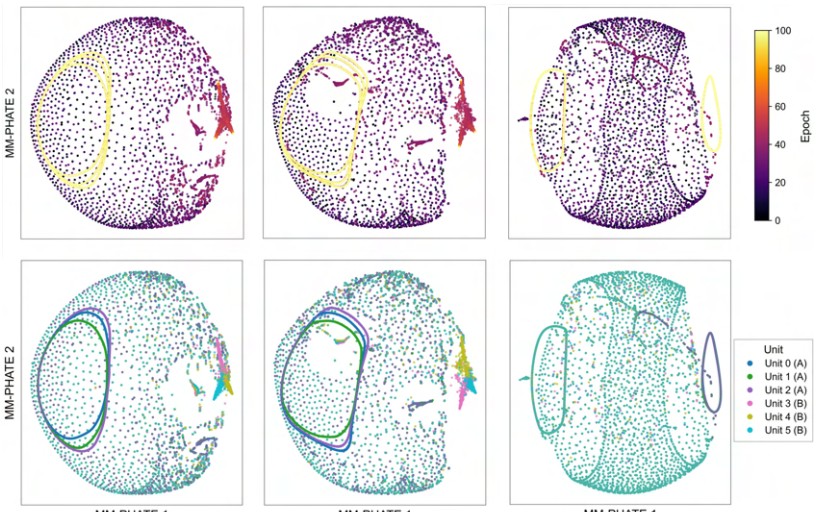

Figure 3: **MM-PHATE on controlled synthetic bifurcation–warp scenarios (first two embedding coordinates).** Columns (left to right) show: **(i)** Hopf vs. Pitchfork with matched $\mu$ sweeps and no warp, **(ii)** Hopf vs. Pitchfork with smooth state-space warps applied to both systems, and **(iii)** a warp-only control in which both groups share the same Hopf latent trajectories and $\mu$ schedule and differ only by whether a smooth state-space warp is applied. Top row: points colored by epoch. Bottom row: points colored by unit index (legend indicates unit identity and group membership).

This Hopf benchmark provides controlled support for two claims used later in the paper. First, in this setting, MM-PHATE more faithfully reflects the latent contraction–transition–stabilization progression than the compared standard embeddings. Second, the embedding-space entropy summaries introduced below are interpretable as geometric markers of representational regime changes: in the Hopf experiment, MM-PHATE intra-step and inter-step entropy recover the expected phase transitions more clearly than the baselines (Figs. A.22–A.23). These synthetic validations motivate the use of the same summaries in task-trained RNNs.

### 5.1.1 Dynamics-family differences and smooth warp perturbations

We next test whether MM-PHATE can separate *dynamical-family differences* from *smooth coordinate perturbations* when both are present. To do so, we construct three controlled settings (Appendix A.5.2): (i) Hopf vs. Pitchfork with matched $\mu$ sweeps and no warp, (ii) Hopf vs. Pitchfork with smooth state-space warps applied to both systems, and (iii) a warp-only control in which two groups share the same Hopf latent trajectories and $\mu$ schedule and differ only by whether a smooth state-space warp is applied.

Figure 3 summarizes the three scenarios (see also the per-scenario appendix visualization suites in Figs. A.6–A.17). In the unwarped Hopf-vs.-Pitchfork setting, MM-PHATE separates the two dynamical families while preserving their distinct progression structure across epochs and time-steps (Figs. A.6–A.9). When both systems are smoothly warped, the family-level distinction remains visible, indicating that the embedding is not driven solely by the raw coordinate representation of the latent trajectories in the tested construction (Figs. A.10–A.13).

In the warp-only control, the two groups share the same Hopf latent trajectories and $\mu$ schedule and differ only by the application of a smooth state-space warp. MM-PHATE separates the clean and warped groups, showing that it is not invariant to smooth warps in a strict sense. However, the post-bifurcation limit-cycle structure remains qualitatively preserved in the MM-PHATE embedding (including pre-/post-bifurcation progression), whereas the warp more strongly distorts the geometry recovered by PCA and t-SNE (Figs. A.14–A.17). This behavior is consistent with robustness of the *qualitative geometric progression* to smooth state-

space warps under the tested conditions, while making clear that MM-PHATE does not satisfy strict warp invariance.

Taken together, the synthetic benchmarks establish a coherent scope for the downstream analyses: MM-PHATE can recover meaningful geometric organization in controlled systems, can preserve qualitative progression under smooth perturbations, and can distinguish family-level dynamics from coordinate-warp effects in the tested settings. We now turn to task-trained RNNs to test whether these geometric signals align with learning-relevant changes in real models.

## 5.2 Neural activity

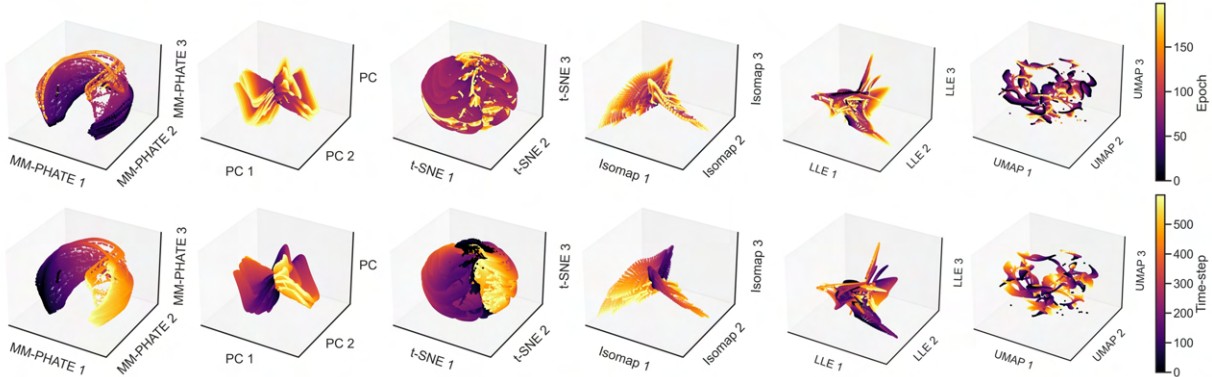

Figure 4: Area2Bump: Visualization of a 20-unit LSTM network trained for 200 epochs. Visualizations are generated using MM-PHATE, PCA, t-SNE, Isomap, Locally Linear Embedding (LLE), and UMAP (left to right). Points are colored by epoch (top row) or time-step (bottom row).

We next apply MM-PHATE to RNNs trained on the Area2Bump dataset, which contains spiking activity recordings from macaques during a reaching task (Chowdhury et al., 2022). We trained a single-layer, 20-unit LSTM to classify reach direction and analyzed how its hidden representations evolve over training. In an example session, the model reached a validation accuracy of 74%. Let $T$ denote the activation tensor across epochs, time-steps, and hidden units (with an activation vector for each unit obtained across trials; see Appendix A.3 for implementation details). Each point in the visualization corresponds to a specific unit–time-step–epoch triple (Fig. 4); in practice, we subsample epochs and time-steps to reduce memory load.

We compare MM-PHATE to five standard embeddings—PCA, t-SNE, Isomap, LLE, and UMAP—and to M-PHATE (Appendix A.4), using the same set of points. For the baselines, we reshape $T$ into a matrix whose rows are unit–time-step–epoch points and whose columns are trial-indexed activations, and then apply each method to this flattened representation.

Qualitatively, MM-PHATE organizes the activity into a single manifold that simultaneously preserves progression across epochs (top row) and continuity across time-steps (bottom row). In particular, MM-PHATE reveals a clear reorganization that becomes pronounced at later time-steps and emerges over mid training (around epoch 40 to 90), roughly coinciding with the main rise and stabilization of validation performance (Fig. 4). By contrast, PCA and Isomap primarily capture a global drift dominated by high-variance directions and do not clearly separate training phases. t-SNE emphasizes local neighborhoods that are strongly driven by time-step variability and often yields discontinuous structure across epochs. LLE and UMAP preserve some within-unit trajectories but still do not clearly expose the mid-training transition visible in MM-PHATE. This motivates the quantitative evaluations below.

### 5.2.1 Neighborhood preservation

To quantitatively compare how well each embedding preserves the structure of the activation tensor, we compute neighborhood preservation between the original activation space and the low-dimensional embedding.

For each point (unit–time-step–epoch triple), we identify its $k$-nearest neighbors in the original space and in the embedding space, and measure the fraction of neighbors that are shared (Appendix A.6). We report the average overlap separately for: (i) *intra-step* neighborhoods (within a fixed epoch and time-step, comparing units), and (ii) *inter-step* neighborhoods (within a fixed epoch and unit, comparing time-steps).

Table 1 summarizes results for the Area2Bump LSTM. MM-PHATE achieves the highest intra-step preservation at informative neighborhood sizes (e.g., $k = 5, 10, 15$), indicating that it best captures within-time-step unit communities. For inter-step neighborhoods, MM-PHATE yields the highest preservation for all but the smallest neighborhood size ($k = 5$), where LLE is marginally higher. We note that intra-step preservation necessarily saturates when $k \geq m$ (here $m = 20$ units), explaining the ceiling at $k = 20$ and $k = 40$; those rows serve mainly as a sanity check.

An additional variant omitting Isomap (due to memory constraints at larger sample sizes) is reported in Appendix Table 3; the relative ranking of methods remains unchanged. Overall, these results support the qualitative impression from Fig. 4: MM-PHATE preserves both *within-time-step* unit neighborhoods and *across-time* trajectories more faithfully than standard alternatives.

Table 1: Neighborhood preservation of visualization methods applied to an RNN trained on Area2Bump.

| k | | MM-PHATE | PCA | t-SNE | Isomap | LLE | UMAP |
|---|---|---|---|---|---|---|---|
| 5 | Intra-step | **0.621** | 0.417 | 0.440 | 0.290 | 0.293 | 0.291 |
| 5 | Inter-step | 0.794 | 0.709 | 0.685 | 0.770 | **0.812** | 0.749 |
| 10 | Intra-step | **0.757** | 0.659 | 0.669 | 0.549 | 0.550 | 0.546 |
| 10 | Inter-step | **0.820** | 0.680 | 0.684 | 0.778 | 0.798 | 0.796 |
| 15 | Intra-step | **0.852** | 0.835 | 0.834 | 0.802 | 0.806 | 0.806 |
| 15 | Inter-step | **0.834** | 0.639 | 0.657 | 0.758 | 0.769 | 0.800 |
| 20 | Intra-step | 1.000 | 1.000 | 1.000 | 1.000 | 1.000 | 1.000 |
| 20 | Inter-step | **0.853** | 0.613 | 0.640 | 0.750 | 0.755 | 0.804 |
| 40 | Intra-step | 1.000 | 1.000 | 1.000 | 1.000 | 1.000 | 1.000 |
| 40 | Inter-step | **0.826** | 0.671 | 0.675 | 0.636 | 0.626 | 0.671 |

### 5.2.2 Intra-step entropy and time-resolved information analysis

To probe how representational geometry changes within an epoch, we measure how dispersed the hidden units are *at each time-step* in the embedding. For a fixed epoch $e$ and time-step $\omega$, we take the $m$ embedded points $\{\mathbf{y}_{e,\omega,i}\}_{i=1}^{m}$ (one per unit) and estimate the differential entropy of this point cloud using a KDE-based estimator. We refer to this quantity as *intra-step entropy*. Importantly, this is an *embedding-space* measure of geometric dispersion (not a direct estimate of the Shannon entropy of the representation distribution in the information-theoretic sense); we use it as a proxy for how strongly units diversify vs. collapse at a given time-step, and we validate its interpretability in a controlled setting. In the Hopf bifurcation experiment (§A.5), MM-PHATE intra-step entropy accurately recovers known contraction/expansion/stabilization regimes of the latent dynamics, whereas variance-dominated baselines do not (Fig. A.22).

On the Area2Bump LSTM, MM-PHATE reveals a structured temporal progression (Fig. 5). Entropy is relatively stable across time-steps until approximately epoch 30. Between epochs 30 and 50, entropy decreases for earlier time-steps while increasing for later ones, coinciding with the initial rise in training and validation accuracy. This pattern is consistent with the network compressing weakly informative early-sequence activity while expanding later representations that more strongly align with the task. From epochs 50 to 90, entropy rises across most time-steps, with later time-steps peaking near epoch 90. Around this point, validation accuracy plateaus and the train–test loss gap widens, indicating the onset of overfitting. Thereafter, entropy for early time-steps continues to increase while entropy for later time-steps declines, consistent with increasing representational variability in the early sequence that does not translate into generalization.

**Linear probes, time-step ablations, and mutual information.** The intra-step entropy trends in Fig. 5 suggest a systematic divergence between early and late time-steps: early states compress around epochs 30–50, later states expand and peak near epoch 90, and after overfitting onset the early sequence

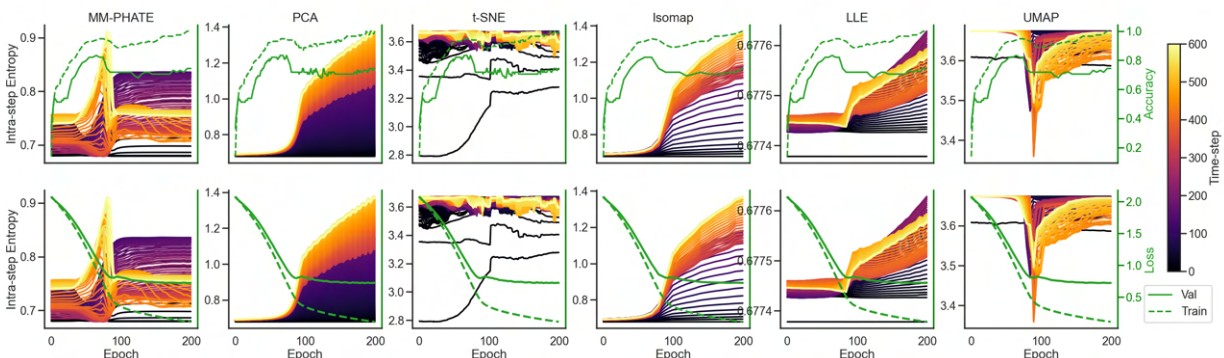

Figure 5: Intra-step entropy of hidden units in embedding space at each time-step and epoch, compared to training/validation accuracy (top) and losses (bottom), for MM-PHATE and baseline embeddings.

becomes increasingly high-entropy while late-step entropy declines. To test whether this early/late divergence reflects *task-relevant* structure (rather than embedding drift or variance scaling), we directly quantify label information in the recurrent state $h_t$ using three complementary diagnostics (Fig. 6; Appendix A.7–A.9): (i) time-resolved linear probes, (ii) time-step ablations of the trained classifier, and (iii) time-resolved mutual information (MI) between labels and $h_t$.

*Linear probes* measure how linearly decodable the label is from $h_t$ at each time-step throughout training. Probe accuracy changes little before epoch 30, consistent with the largely stable intra-step entropy over the same interval. Over epochs 30–50, probe performance begins to separate by time-step: later time-steps improve steadily and reach a higher, more stable test plateau, while early time-steps exhibit only a transient increase (peaking around epochs 50–60) before degrading as the train–test gap widens. Notably, the *phase boundaries* in probe behavior align with the intra-step entropy phases: after approximately epoch 90, the onset of pronounced early time-step overfitting coincides with a sharp rise in early-step entropy, whereas the late time-step test plateau occurs in the same interval in which late-step entropy begins to decline.

*Time-step ablations* test which parts of the sequence the *trained* decision rule depends on. Masking late inputs (early-only) yields an early rise and a mid-training peak near the onset of overfitting, followed by a decline, consistent with reliance on brittle early-sequence shortcuts that do not generalize. In contrast, masking early inputs (late-only) improves gradually but remains below intact performance. After roughly epoch 80, early-only and late-only accuracies converge to a similar moderate level while the intact model remains substantially higher, indicating that strong final performance depends on the *full recurrent trajectory*: evidence accumulated over the sequence is expressed most reliably in later hidden states, but cannot be recovered from either segment alone.

*Time-resolved MI* corroborates this picture by measuring total label information present in $h_t$, independent of the readout. MI increases over training for both early and late segments, but remains consistently higher for later time-steps and reaches a larger final value. Crucially, the post-overfitting regime shows a decoupling for early time-steps: intra-step entropy continues to rise while probe accuracy and MI do not, indicating that the growing early-step dispersion reflects label-irrelevant variability rather than useful task information.

Taken together, probes, ablations, and MI validate the interpretation of MM-PHATE intra-step entropy as a meaningful geometric proxy: later time-steps progressively concentrate stable, task-relevant information, whereas early time-steps ultimately accumulate high-entropy variability that does not support generalization.

**Comparison with baseline embeddings.** Repeating the intra-step entropy analysis using PCA, t-SNE, Isomap, LLE, and UMAP (Fig. 5) yields substantially weaker alignment with probes, ablations, and MI. In particular, the baselines do not clearly reproduce the early compression / late expansion regime or the later divergence between early vs. late time-steps; their entropy changes are delayed and/or dominated by global geometric effects. This mirrors the controlled Hopf bifurcation experiment, where standard embeddings fail

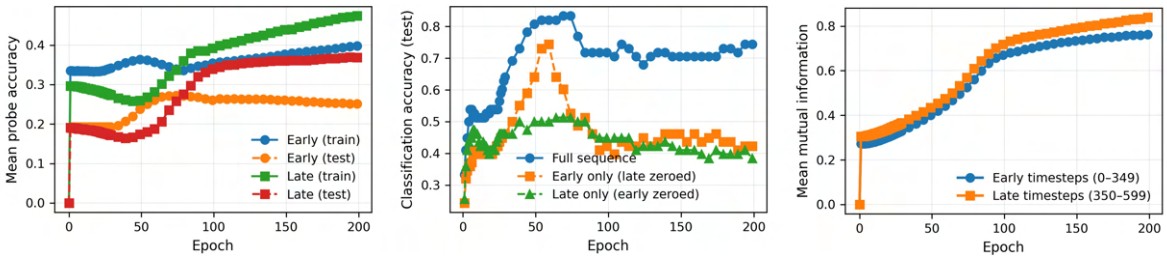

Figure 6: Combined analysis of time-resolved representations. **Left:** Linear probe accuracy for early and late time-steps on train and test sets. **Middle:** Time-step ablation accuracy for intact sequences, early-only sequences, and late-only sequences. **Right:** Time-resolved mutual information between labels and hidden states. All three diagnostics support the intra-step entropy interpretation: task-relevant information migrates toward later time-steps over training, while rising early-step dispersion after overfitting reflects label-irrelevant variability rather than improved generalization.

to recover the known entropy transitions. In contrast, MM-PHATE provides a time-resolved view in which entropy trends, separability in the embedding, probe performance, ablation performance, and MI jointly track how different segments of the input sequence contribute to learning and overfitting.

### 5.2.3 Inter-step entropy

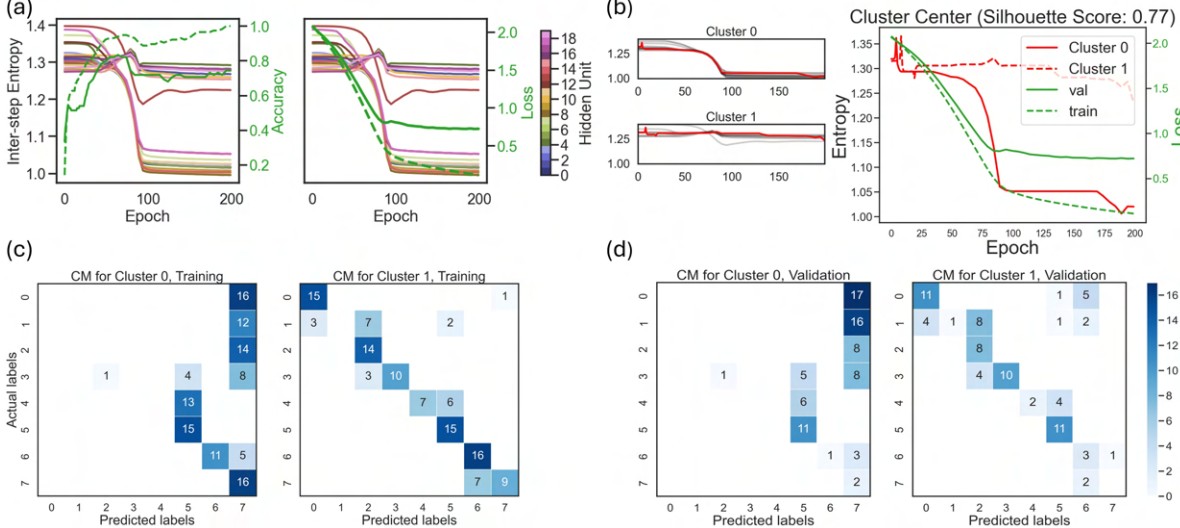

Figure 7: a) Inter-step entropy of each hidden unit across epochs, alongside model accuracy (left) and loss (right). b) Clusters of hidden units from inter-step entropy trajectories across epochs. Left: trajectories of units in cluster 0 (12 units) and cluster 1 (8 units). Right: cluster center trajectories across epochs. c–d) Confusion matrices of sub-networks constructed from each cluster on training and validation data.

The analyses above characterize how representations evolve *across units* at a fixed time-step. We now ask a complementary question: *how temporally selective is each unit across the sequence?* Whereas intra-step entropy summarizes diversity across units at a given $(\tau, \omega)$, inter-step entropy measures (for each unit) how strongly its representation differentiates among time-steps within an epoch, revealing temporal sensitivity.

Inter-step entropy reveals two robust groups of units (Fig. 7a). One subset maintains higher inter-step entropy throughout training, peaking around epoch 80, indicating sustained temporal differentiation. The other subset shows a pronounced decline in inter-step entropy later in training, indicating a collapse in temporal selectivity. To formalize this distinction, we apply time-series $k$-means with Dynamic Time Warping

(DTW) distance (Bringmann et al., 2023) to the inter-step entropy trajectories (Fig. 7b), so that units are grouped by their *temporal pattern* of entropy changes even when the precise epochs of increase or decrease are slightly shifted. We emphasize that this DTW-based clustering is used descriptively to summarize the heterogeneity already visible in Fig. 7a, and our main conclusions do not depend on a specific choice of cluster number or assignments. Sub-networks constructed from these clusters exhibit large performance differences (Fig. 7c–d): the model using only the high-temporal-selectivity cluster achieve substantially higher training and validation accuracy than the model using only the low-temporal-selectivity cluster, confirming that the temporally differentiating units carry the most task-relevant signal.

This unit-level decomposition complements the intra-step entropy, probe, ablation, and MI results. Together they indicate that the example learning is accompanied by (i) increased task-relevant structure at later time-steps and (ii) an emerging specialization of hidden units, with a subset sustaining temporal selectivity while another subset becomes less informative later in training. Among baseline dimensionality reduction techniques, PCA, t-SNE, Isomap, LLE, and UMAP fail to recover this functional separation or the associated training-phase transitions (Fig. A.43). In contrast, MM-PHATE preserves temporal continuity while exposing distinct unit-level roles, yielding a coherent view of representational reorganization during learning.

Results for additional architectures (GRU, Vanilla RNN) and LSTM sizes (10–50 units) are provided in the appendix and show similar qualitative trends.

### 5.3 Analysis with Human Activity Recognition Model

To test whether the training-phase structure observed in Area2Bump generalizes beyond neural spiking data, we trained a 30-unit LSTM for kinematics-based Human Activity Recognition (HAR) (Reyes-Ortiz et al., 2012). The model was trained for 1000 epochs and reached a final validation accuracy of 84%. We applied MM-PHATE to the full hidden-state activation tensor and repeated the intra-step and inter-step analyses from Section 5.2.

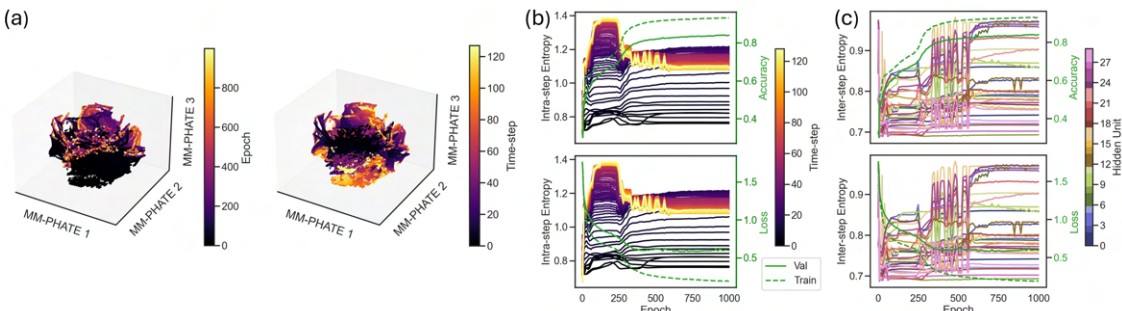

Figure 8: a) MM-PHATE visualization of a 30-unit LSTM network trained on HAR data, colored by epoch (left) and time-step (right). b) Intra-step entropy, c) inter-step entropy.

**MM-PHATE resolves distinct training phases in within-sequence geometry.** The HAR MM-PHATE embedding exhibits an ordered training trajectory (Fig. 8a, left): epochs progress smoothly along a coherent manifold rather than drifting arbitrarily. When colored by time-step (Fig. 8a, right), the within-sequence flow is not stationary across training, suggesting that the mapping from time-step progression to motion on the representation manifold changes with learning. To quantify this effect, we compute two epoch-wise metrics on the 3D embedding (Appendix A.14): `signed flow alignment centroid` (direction of within-sequence progression) and `epoch-to-epoch change magnitude` (magnitude of temporal-geometric reorganization between consecutive sampled epochs). Both metrics identify an early transition: the signed flow changes sign (negative to positive) around epoch $\sim 7$, coinciding with a spike in the change magnitude and a sharp increase in training/validation accuracy (Figs. A.47– A.49). This alignment supports that MM-PHATE captures a learning-relevant reorganization of within-sequence dynamics rather than a purely geometric artifact.

**Intra-step entropy shows a plateau–drop regime aligned with performance gains.** We next measure intra-step entropy to summarize how dispersed hidden units are within each time-step and epoch in embedding space (Fig. 8b), in which we observed a transition around epoch 7 corresponding to the reorganization of within-sequence dynamics (Fig. A.50). Across training, later time-steps exhibit a characteristic *plateau followed by a drop* in entropy, while earlier time-steps tend to increase. Importantly, the entropy drop aligns with an increase in validation performance, suggesting that the late-step contraction reflects a *useful compression* of representations into a more task-aligned, stable geometry. This interpretation is consistent with our controlled Hopf benchmark, where MM-PHATE intra-step entropy tracks known contraction/expansion/stabilization regimes (Section A.5), providing a calibrated link between entropy transitions and underlying dynamical organization.

**Inter-step entropy indicates increasing temporal specialization during the same regime.** Inter-step entropy (Fig. 8c) provides a complementary unit-level view by measuring how strongly each unit differentiates among time-steps within an epoch. During the interval in which late-step intra-step entropy drops and accuracy improves, many units show increased inter-step entropy, indicating stronger temporal selectivity. Together, the plateau–drop pattern in intra-step entropy and the concurrent rise in inter-step entropy support a coherent picture: as learning progresses, the network both (i) consolidates later-step representations into a more compact, task-relevant geometry and (ii) increases the temporal specialization of a subset of units, yielding a clearer division of roles across the sequence.

**Relation to expansion–compression dynamics and comparison to Area2Bump.** The HAR results provide a time-resolved instantiation of the expansion–compression view of representation learning in trained RNNs (Farrell et al., 2022). Here, MM-PHATE makes the phases explicit: an early geometric reorientation in within-sequence flow (captured by embedding-defined metrics) is followed by an entropy regime in which later time-steps increase, exhibit a plateau, and then compress, and this compression aligns with improved generalization and increased temporal specialization.

This behavior differs qualitatively from Area2Bump (Section 5.2). Area2Bump does not show a late-step plateau–drop in intra-step entropy that co-occurs with improved validation performance; instead, late training is dominated by rising early-step entropy that is decoupled from label information (probes/MI), consistent with label-irrelevant variability associated with overfitting rather than consolidation. A plausible contributor is intrinsic data complexity relative to model capacity: HAR is substantially lower-dimensional than Area2Bump (6 vs. 35 PCs to explain 95% variance; Fig. A.2), making it easier for a fixed-size LSTM to concentrate decision-relevant structure into a compact representation. Accordingly, the same MM-PHATE-based diagnostics highlight different regimes: HAR exhibits a performance-aligned late-step entropy drop together with increased inter-step entropy for many units, whereas Area2Bump shows weaker evidence of such consolidation and a late rise in early-step dispersion that does not translate into generalization.

Overall, these results extend our findings from controlled dynamics (Hopf) and neural spiking data (Area2Bump) to a distinct kinematic domain, showing that MM-PHATE can resolve multiple learning-relevant phases of within-sequence organization and that these phases can be quantified directly from the embedding.

## 6 Conclusion

We introduced MM-PHATE, a visualization framework for RNNs that embeds hidden-state dynamics jointly across time-steps and training epochs via a multiway multislice graph. This construction preserves within-time-step unit communities and across-time trajectories in a single coherent manifold in the settings studied here, addressing common failure modes of standard embeddings that collapse or distort one or more of these axes.

Across controlled synthetic benchmarks and real RNN applications, MM-PHATE produced geometry that aligned with known or independently measured structure. In synthetic bifurcation systems, MM-PHATE recovered the Hopf contraction/expansion/stabilization progression more faithfully than the compared standard embeddings and remained informative under smooth state-space warp perturbations, while preserving

visible distinctions between dynamical families (Hopf vs. Pitchfork) when both dynamics-type and warp differences were present. These controlled results provide empirical support for the geometric interpretations used later in the paper, but do not constitute a formal invariance guarantee.

On task-trained RNNs, MM-PHATE achieved the strongest neighborhood preservation on Area2Bump and revealed time-resolved training-phase changes that aligned with probes, ablations, and mutual information, while inter-step entropy identified functionally distinct unit groups with different temporal selectivity. On HAR, MM-PHATE generalized these observations and exposed a pronounced training-phase transition (including a reversal of within-sequence flow direction) with coordinated intra-/inter-step entropy changes that coincided with performance improvements, suggesting task-dependent consolidation consistent with dataset complexity differences.

**Limitations and future directions.** MM-PHATE encodes continuity across time and epochs; highly non-smooth training dynamics (e.g., abrupt restarts or extreme learning rates) may reduce interpretability. While our synthetic benchmarks cover multiple bifurcation families and smooth state-space warps, extending these evaluations and developing theoretical guarantees (e.g., under classes of smooth equivalences or perturbations) remains an important direction. The $nsm \times nsm$ kernel can also limit scale, but PHATE is robust to subsampling and scalable approximations (e.g., subsampling, coarsening, partitioning) are practical routes forward. Extending the multiway multislice construction to richer architectures (e.g., deeper recurrent stacks or attention-based models) is an important direction for future work (Katharopoulos et al.).

### Acknowledgments

N.M. was funded as a fellow by the Kavli NeurData Discovery Institute. G.M. was supported by NSF grant numbers CCF 2217058 and 2403452. A.S.C. was supported in part by NSF CAREER award number 2340338.

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

# A  Appendix

## A.1  Datasets

We employed two datasets (Area2Bump as a neuroscience-motivated benchmark of population dynamics; HAR as a real-world sequential classification dataset).

1. Area2Bump is a part of the well-known "Neural Latents Benchmark" Challenge (Pei et al., 2021) in the neuroscience community. The Area2Bump dataset (Chowdhury et al., 2022) consists of neural activity recorded from Brodmann's area 2 of the somatosensory cortex while macaque monkeys performed a slightly modified version of a standard center-out reaching task. Dataset license: CC0 1.0. According to the authors, data was collected consistently "with the guide for the care and use of laboratory animals and approved by the institutional animal care and use committee of

Northwestern University under protocol #IS00000367". This dataset, collected from monkeys thus does not contain personally identifiable information. Since it is neural data, we do not consider it to be offensive content.

Data Statistics: The dataset includes 193 samples, split into 115 training samples and 78 testing samples. Each sample consists of 600 time-steps with 65 features per time-step. For the training set, the mean values across features range from 0.0001 to 0.03, with standard deviations between 0.001 and 0.02, and maximum values ranging from 0.003 to 0.12. In the test set, the mean values range from 0.00008 to 0.03, standard deviations from 0.0007 to 0.018, and maximum values from 0.0007 to 0.13.

2. The HAR dataset was originally introduced in Anguita et al. (2013) which has been cited 3k times, and is also part of the UCI ML repository (where it has been viewed 196k times). The Human Activity Recognition (HAR) Using Smartphones dataset (Reyes-Ortiz et al., 2012), kinematic recordings of 30 subjects performing daily living activities with a smartphone embedded with an inertial measurement unit. Dataset license: CC BY 4.0. Information relating to participant consent was not found relating to this dataset. This dataset does not reveal participant name or identifiable information. The kinematics contained in the dataset are not considered offensive content.

Data Details: The dataset includes six activity classes (e.g., Walking, Sitting) based on accelerometer and gyroscope data sampled at 50 Hz. The sensor data has been pre-processed with noise filtering and separated into gravitational and body motion components. Data is windowed into 2.56-second segments (128 data points) with 50% overlap, resulting in 561-dimensional feature vectors per window. The dataset is split into 70% training (21 subjects, 7352 samples) and 30% testing (9 subjects, 2947 samples).

Data Statistics: For the training set, mean values across features range from -0.0008 to 0.8, with standard deviations between 0.1 and 0.41, and values spanning from -5.97 to 5.75. For the test set, mean values range from -0.013 to 0.8, with standard deviations between 0.095 and 0.41, and values spanning from -3.43 to 3.47.

## A.2 Model Training

We used TensorFlow's Keras API for all model training and validation.

The network in Section 5.2 was trained as follows. The Area2Bump dataset was randomly split into training and validation subsets containing an equal number of samples for each input class with an 8 to 2 ratio. Additional samples that would have made the training data uneven were added back to the validation subset to utilize all available samples. The network consists of an LSTM layer with 20 units. This is followed by a Flatten layer that converts the LSTM's output into a one-dimensional vector. Finally, a Dense layer with 8 units and a softmax activation function produces the output for the 8-class classification tasks. The network was trained with a batch size of 64. We used an Adam optimizer with a learning rate of $1e^{-4}$. During the training process, we recorded the activations from the LSTM layer into the activation tensor $\boldsymbol{T}$ for visualization.

The network in Section 5.3 was trained as follows. The HAR dataset was preprocessed and split by the authors into training and validation subsets according to the subjects with a 7 to 3 ratio. The network consists of an LSTM layer with 30 units and a Dense layer with 6 units and a softmax activation function produces the output for the 6-class classification tasks. The network was trained with a batch size of 32. We used an Adam optimizer with a learning rate of $2e^{-5}$. During the training process, we recorded the activations from the LSTM layer into the activation tensor $\boldsymbol{T}$ for visualization.

## A.3 Implementation of visualization methods

**Common tensor and reshaping.** For each trained network or dynamical system, we record hidden activations into a tensor with axes (epoch $\tau$) × (unit $i$) × (time-step $\omega$) × (trial/sample index). For baseline embeddings that operate on a matrix, we reshape the sampled tensor into

$$\boldsymbol{T}_{\mathrm{mat}} \in \mathbb{R}^{N_{\mathrm{pts}} \times p},$$

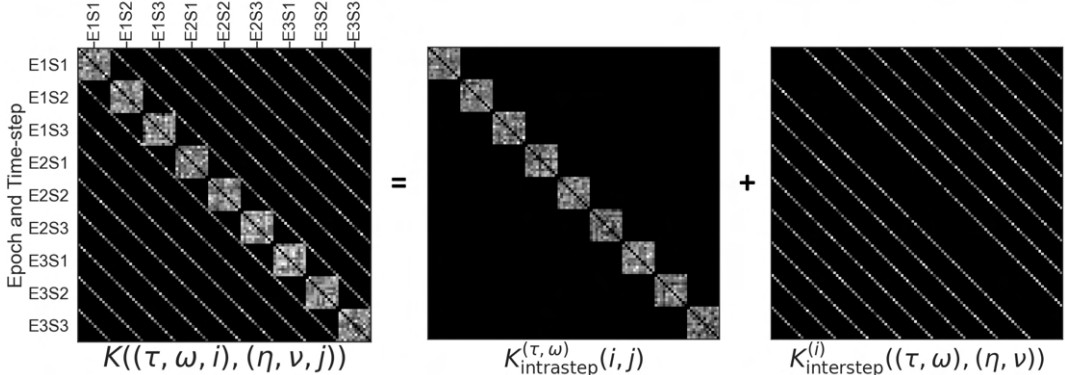

Figure A.1: Example schematic of the multiway multislice kernel used in MM-PHATE for RNNs. The intra-step kernels represent the similarities between the graph nodes at the same time-steps. The inter-step kernels represent the similarities between the nodes and themselves at different time-steps and epochs.

where each row corresponds to one unit–time-step–epoch triple and the $p$ columns index trial-wise activations, as in the main text.

**Hopf bifurcation dataset.** For the Hopf experiment, the main activation tensor is

$$\texttt{trace\_data} \in \mathbb{R}^{(ET) \times H \times S},$$

where $E$ is the number of epochs, $T$ the number of intrinsic time-steps per epoch, $H$ the number of hidden units, and $S$ the number of samples (trials) per unit. Unless otherwise noted, Hopf embeddings use *all* epochs and time-steps; we do not subsample $\tau$ or $\omega$ for MM-PHATE, PCA, t-SNE, or UMAP. LLE and Isomap operate on a random subset of points, but still draw from the full $(\tau, \omega, i)$ grid.

**Area2Bump subsampling used for MM-PHATE and most baselines.** Due to memory constraints, for Area2Bump we compute MM-PHATE on a subsampled tensor $T_{\mathrm{MM}}$ defined by: (i) epochs $\tau$ sampled as the first 29 epochs followed by every 5th epoch thereafter (up to 200 total epochs), and (ii) time-steps $\omega$ sampled uniformly over the 600-step sequence using 100 evenly spaced indices. Unless stated otherwise below, t-SNE, LLE, UMAP are computed on this same sampled tensor $T_{\mathrm{MM}}$.

**HAR subsampling used for MM-PHATE.** For HAR, we sample epochs as the first 29 epochs followed by every 10th epoch thereafter (up to 1000 epochs) and compute MM-PHATE on the resulting tensor. For baseline embeddings we reshape the corresponding sampled tensor into $T_{\mathrm{mat}}$ in the same way.

### A.3.1 MM-PHATE

For Area2Bump and HAR, MM-PHATE is applied to the sampled tensors described above (Area2Bump: $T_{\mathrm{MM}}$; HAR: the sampled HAR tensor). We construct the multiway multislice kernel over units, time-steps, and epochs and then run PHATE via the M-PHATE package with `n_components = 3`.

For the Hopf bifurcation dataset, we apply MM-PHATE directly to `trace_data` of shape $(ET, H, S)$ (no epoch or time-step subsampling), again with `n_components = 3`.

### A.3.2 PCA

Due to PCA's low computational demand, we did not subsample any data before applied PCA using `sklearn.decomposition.PCA`.

Figure A.2 reports how many principal components are required to explain 95% of the variance for Area2Bump and HAR.

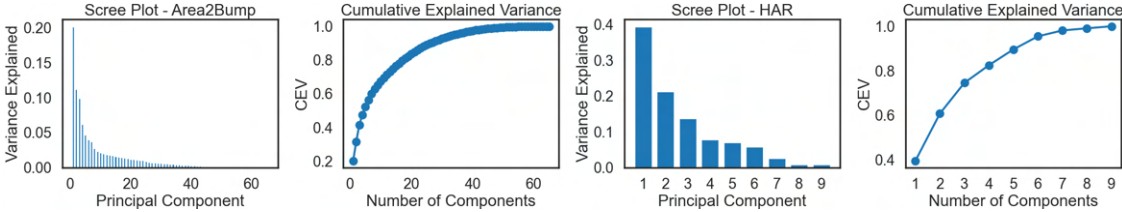

Figure A.2: PCA on the two datasets. Area2Bump requires 35 PCs (left) to explain 95% of the variance, whereas HAR requires 6 PCs (right).

For the Hopf dataset, we flatten `trace_data` to

$$X \in \mathbb{R}^{(ETH) \times S}$$

by stacking all $(\tau, \omega, i)$ combinations and treating the $S$ samples as features. We mean-center $X$ across samples and fit PCA with `n_components = 3` to obtain a 3D embedding used in the Hopf figures (epoch-, time-, unit-, and group-colored views).

### A.3.3   t-SNE

For Area2Bump, we first reduce $T_{\mathrm{mat}}$ to 15 principal components (explaining 99.93% of the variance) and then run Barnes–Hut t-SNE with `n_components = 3`, `perplexity = 50`, and `n_iter = 2000` using `sklearn.manifold.TSNE`.

For Hopf, we run the same t-SNE configuration on the centered matrix $X \in \mathbb{R}^{(ETH) \times S}$ described above (no epoch or time-step subsampling). Because $ETH$ is on the order of $8 \times 10^4$ in our configuration, this run is heavy but tractable on our cluster.

### A.3.4   Isomap

Due to scalability constraints, Isomap is only applied on subsampled data.

For Area2Bump, we build a more aggressively subsampled tensor $T_{\mathrm{ISO}}$ by sampling (i) epochs as the first 29 epochs followed by every 10th epoch thereafter, and (ii) 50 evenly spaced time-steps from the 600-step sequence. We then reduce $T_{\mathrm{ISO}}$ to 15 PCs and apply `sklearn.manifold.Isomap` with `n_neighbors = 30` and `n_components = 3`.

For Hopf, we do not subsample epochs or time-steps, but we do randomly subsample points across the flattened matrix $X \in \mathbb{R}^{(ETH) \times S}$ to form

$$X_{\mathrm{sub}} \in \mathbb{R}^{N_{\mathrm{sub}} \times S},$$

where $N_{\mathrm{sub}} = \min(10{,}000, ETH)$ (with a fixed random seed). We then run `Isomap(n_neighbors = 30, n_components = 3)` on $X_{\mathrm{sub}}$. The label vectors for epoch, time-step, unit, and group are restricted to the same subsample to produce unit-, epoch-, and group-colored Hopf panels.

### A.3.5   Locally Linear Embedding (LLE)

For Area2Bump, LLE is computed on the same sampled tensor $T_{\mathrm{MM}}$ used for MM-PHATE. We reshape to $T_{\mathrm{mat}}$ and apply `LocallyLinearEmbedding(n_neighbors = 100, n_components = 3, method = "modified")`.

For Hopf, as with Isomap, we use the random subsample $X_{\mathrm{sub}}$ of size $N_{\mathrm{sub}}$ drawn from the full flattened matrix $X$. We then run `LocallyLinearEmbedding(n_neighbors = 20, n_components = 3, method = "modified")` to obtain a 3D embedding on the subsampled points, and use the corresponding epoch/time/unit/group labels to generate the Hopf LLE panels.

### A.3.6 Uniform Manifold Approximation and Projection (UMAP)

For Area2Bump, UMAP is applied to $T_{\mathrm{mat}}$ formed from $T_{\mathrm{MM}}$. We use `umap.UMAP` with `n_neighbors = 50`, `min_dist = 0.1`, `n_components = 3`, Euclidean metric, and `random_state = 24`.

For Hopf, UMAP is run on the full centered matrix $X \in \mathbb{R}^{(ETH) \times S}$ with `n_neighbors = 30`, `min_dist = 0.1`, `n_components = 3`, and Euclidean metric. No epoch or time-step subsampling is used here; each $(\tau, \omega, i)$ point is included.

### A.4 M-PHATE as a Qualitative Baseline on a single time-step

To provide a qualitative point of reference, we also apply M-PHATE to the Hopf bifurcation experiment and the Area2Bump LSTM. Importantly, M-PHATE operates on a *single snapshot* of network activity: it embeds units based solely on their activations at one time-step per epoch (typically the final step). We select the final intrinsic time-step $\omega = T - 1$ for all epochs, yielding a tensor

$$\texttt{trace\_last} \in \mathbb{R}^{E \times H \times S}$$

and flatten it to

$$X_{\mathrm{last}} \in \mathbb{R}^{(EH) \times S}.$$

After mean-centering across samples, we run `phate.PHATE(n_components = 3, knn = 30)` to obtain a 3D embedding of epoch–unit pairs at the final time-step. This differs fundamentally from MM-PHATE, which embeds the full multislice tensor (units × time × epoch) using a diffusion process that jointly couples temporal and epoch-wise transitions.

**Why M-PHATE is not an appropriate comparison.** Because M-PHATE ignores all but a single snapshot of the time-steps, it discards the within-epoch temporal dynamics that are essential for understanding recurrent computation. RNNs process input sequences through evolving trajectories, and many of their representational transitions—including contraction, expansion, or bifurcation-like behavior—occur *within* an epoch's trajectory rather than only at its final state. Consequently:

- M-PHATE cannot represent how individual hidden units transition across time-steps.

- It cannot reveal dynamical geometry such as pre-bifurcation collapse, transition structure near $\mu \approx 0$, or post-bifurcation limit-cycle organization.

- It embeds a different mathematical object (units × epochs), whereas MM-PHATE embeds units × time × epochs; the two inhabit incompatible geometric spaces, making direct quantitative comparison inappropriate.

Thus M-PHATE serves only as a *qualitative* baseline showing how standard diffusion-based embeddings behave when restricted to a single time-slice, rather than an algorithm targeting dynamical systems.

**Hopf control experiment.** Figure A.3 shows M-PHATE applied to the last time-step of the Hopf dataset across epochs and units. While it captures a smooth trajectory as $\mu$ increases, it misses the characteristic change in *within-epoch* flow geometry that differentiates pre- and post-bifurcation dynamics (e.g., noise-dominated collapse vs. growing limit cycles). This demonstrates that restricting to the final state obscures the dynamical phenomena MM-PHATE is designed to uncover (Fig. A.18).

**Area2Bump LSTM.** Figure A.4 shows the same M-PHATE procedure applied to the 20-unit LSTM trained on the Area2Bump dataset. The structure is smooth across epochs, and hidden units display modest differentiation; however, the embedding lacks any representation of the sequential processing dynamics that occur within each trial. As a result, M-PHATE provides a useful snapshot-level visualization, but it cannot address questions related to temporal geometry, representational flow, or information propagation that MM-PHATE is explicitly designed to capture.

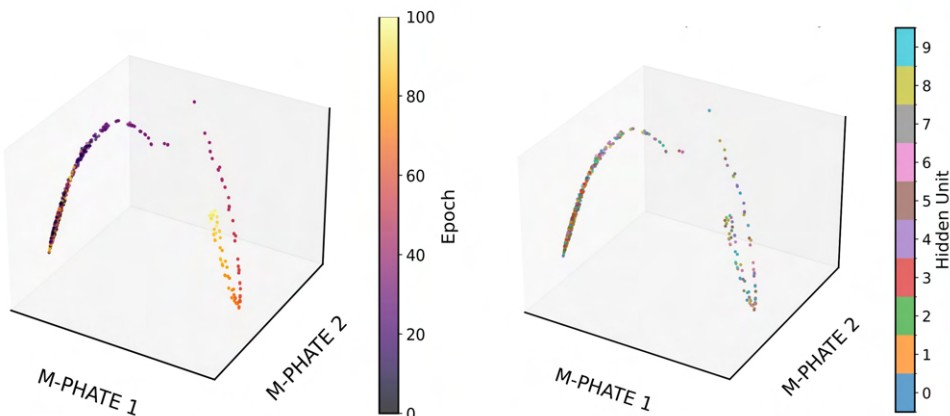

Figure A.3: M-PHATE applied to the Hopf bifurcation experiment (last time-step only). Each point corresponds to a hidden unit at the final time-step of an epoch. The embedding captures a smooth progression across epochs but fails to reveal the underlying bifurcation geometry, which manifests primarily in the *temporal* (within-epoch) dynamics that M-PHATE does not incorporate.

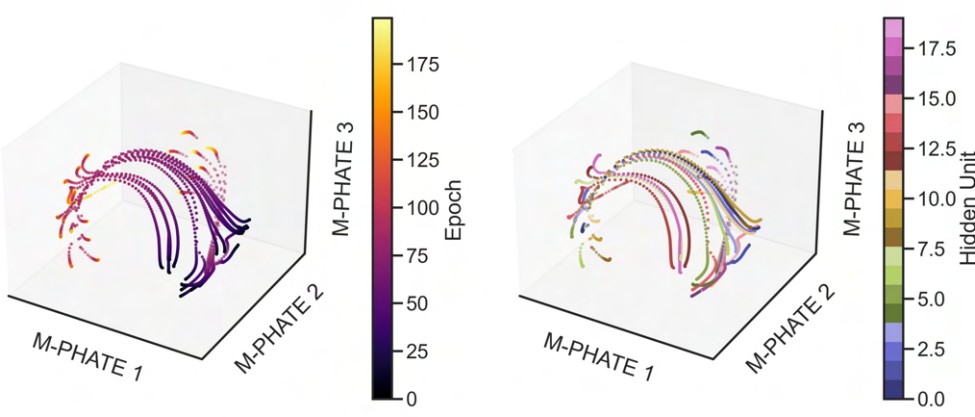

Figure A.4: M-PHATE applied to the Area2Bump LSTM (last time-step across epochs). Points represent hidden units at the final time-step of each epoch, colored by epoch (left) and by unit identity (right). The visualization captures smooth epoch progression but omits the temporal dynamics that are central to recurrent computation.

## A.5    Controlled Synthetic Bifurcation Benchmarks with Smooth Warps

**Scope and motivation.** This appendix documents the controlled synthetic benchmark suite used to address reviewer concerns about (i) multiple bifurcation families, (ii) smooth state-space perturbations (warps), and (iii) settings in which dynamics-family differences and smooth warps are both present simultaneously. These experiments are intended as *empirical stress tests* of MM-PHATE's geometric behavior in controlled settings, not as a formal guarantee of invariance or equivalence under smooth transformations. In particular, our state-warp experiments are motivated by recent work on dynamical comparisons under smooth transformations and alignment/equivalence viewpoints (Sagodi & Park, 2025; Chen et al., 2024; Friedman et al., 2025), but our evaluation remains qualitative/empirical and limited to the tested benchmark conditions.

### A.5.1 Data Generation

We use controlled low-dimensional synthetic dynamical systems to test whether MM-PHATE recovers geometric organization associated with bifurcation structure, and whether these patterns remain interpretable under smooth perturbations of the latent coordinates. The benchmark suite is generated from canonical two-dimensional bifurcation systems (Hopf and Pitchfork), with optional smooth state-space warps before the latent-to-hidden readout.

The benchmark design probes three complementary questions: (i) whether embeddings distinguish qualitatively different bifurcation families (Hopf vs. Pitchfork), (ii) whether this distinction remains visible under smooth state-space warps, and (iii) whether the method preserves the qualitative bifurcation progression in a warp-only control (Hopf clean vs. Hopf warped) where the two groups are identical before warping and readout. In the warp-only control, we simulate a single latent trajectory once, copy it to both groups, and apply the warp only to one group, so the pre-warp latent trajectories are exactly identical by construction. This removes latent-initialization and trajectory-generation confounds when assessing sensitivity to coordinate changes.

Each synthetic dataset contains two latent groups (A and B), each governed by a two-dimensional ODE with control parameter $\mu$ varying across training epochs. For epoch $\tau \in \{0, \ldots, n-1\}$, the control parameter for group $g \in \{A, B\}$ follows a linear schedule

$$\mu_g(\tau) = \mu_{g,\text{start}} + \frac{\tau}{n-1}\big(\mu_{g,\text{end}} - \mu_{g,\text{start}}\big), \tag{3}$$

so that different scenarios can implement matched sweeps (e.g., Hopf vs. Pitchfork with $\mu : -1 \to 2$), within-family parameter shifts (e.g., $-1 \to 2$ vs. $-1 \to 3$), or identical schedules for warp-only controls.

**Latent dynamics.** We use two canonical supercritical bifurcation systems. For the Hopf benchmark, we use the supercritical Hopf normal form

$$\dot{x} = \mu x - \omega y - (x^2 + y^2)x, \tag{4}$$
$$\dot{y} = \omega x + \mu y - (x^2 + y^2)y, \tag{5}$$

where $\omega$ is the angular frequency (always 1 here for simplicity). For $\mu < 0$, trajectories contract to the fixed point at the origin; for $\mu > 0$, trajectories converge to a stable limit cycle.

For the Pitchfork benchmark, we use a supercritical pitchfork in $x$ with a linearly stable auxiliary dimension $y$:

$$\dot{x} = \mu x - x^3, \tag{6}$$
$$\dot{y} = -\lambda y, \tag{7}$$

with $\lambda > 0$. This produces qualitatively different post-bifurcation organization from Hopf (branching in $x$ rather than rotational limit-cycle dynamics).

**Epochs and time-steps.** Within each epoch $\tau$, the latent system is integrated for $s$ time-steps at fixed $\mu_g(\tau)$ using a fourth-order Runge–Kutta method with step size $\Delta t$. Thus each epoch contributes a short trajectory segment under a constant control parameter, and the full dataset traces a progression across epochs as $\mu$ sweeps through the bifurcation regime.

**Optional smooth warps.** To test robustness to smooth perturbations of the latent coordinates, we apply an optional smooth state-space warp (a near-identity diffeomorphic transform in $\mathbb{R}^2$) to each epoch trajectory before readout. In the implementation used here, the state warp is

$$\Phi_\kappa(x, y) = \big(x + \kappa xy, \; y + \kappa x^2\big), \tag{8}$$

with small $\kappa$ (default $\kappa = 0.15$), yielding a smooth coordinate deformation that preserves local continuity while changing the raw coordinate representation.

**Readout mapping (latent → hidden-unit activation).** At each time-step $\omega$ in epoch $\tau$, the latent state $(x, y)$ is mapped to a scalar activation by combining a shared linear projection and a radial component:

$$h_{\tau,w} = \alpha \, \boldsymbol{w}^\top [x_{\tau,w}, y_{\tau,w}] \; + \; \gamma \, r_{\tau,w}, \qquad r_{\tau,w} = \sqrt{x_{\tau,w}^2 + y_{\tau,w}^2}, \tag{9}$$

where $\boldsymbol{w} \in \mathbb{R}^2$ is a shared random projection direction (sampled once per dataset and shared across both groups), and $\alpha, \gamma$ control the relative contributions of signed projection and radius. Optionally, the scalar readout is passed through $\tanh(\cdot)$ to mimic an RNN nonlinearity.

**Hidden units and group structure.** Each group contributes a fixed number of hidden units (typically 3 for group A and 3 for group B in the minimal benchmark suite). Within a group, units are generated by replicating the same group-level scalar readout and adding small i.i.d. noise, with optional per-unit gain and bias jitter. This produces tight but non-identical unit trajectories while preserving the underlying group-level latent dynamics.

**Samples.** We generate $p$ independent samples (trials), each with its own initial condition and process noise realization. All samples within a scenario share the same ODE specification, parameter schedule(s), and warp settings, but produce distinct trajectories. In the warp-only control, the two groups share the same simulated latent trajectories before warping by construction, and differ only through the applied warp and downstream readout replication.

**Tensor structure and default configuration.** The final activation tensor matches the notation of Table 4:

$$T \in \mathbb{R}^{(n\,s) \times m \times p},$$

where the first dimension indexes flattened epoch–time-step pairs $(\tau, w)$, $m$ is the total number of hidden units across both groups, and $p$ is the number of samples. In the benchmark configuration used throughout the synthetic suite unless otherwise stated, we use

$$n = 101, \qquad s = 80, \qquad m = 6 \; (= 3 + 3), \qquad p = 10,$$

with $\Delta t = 0.1$ and a linear sweep in $\mu$ across epochs (e.g., $\mu : -1 \to 2$). Figure A.5 illustrates the latent and readout construction for a Hopf-based example.

The Hopf anchor benchmark (static vs. dynamic) used for entropy validation (Section A.5.4) uses a different unit split (6 dynamic + 4 static) and is described separately there.

### A.5.2 Scenario suite used in the main text

Table 2 summarizes the three core synthetic scenarios used in the main-text comparison (Section 5.1.1). These scenarios isolate (a) dynamical-family differences, (b) smooth state-warp perturbations, and (c) the combination of both factors. Unless otherwise stated, all scenarios use matched epoch/time-step settings, the same readout construction, and matched $\mu$ schedules across groups.

### A.5.3 Synthetic-suite results: bifurcation families and smooth state-space warps

Figure 3 (main text) summarizes the three core synthetic scenarios listed in Table 2. In this appendix subsection, we provide the corresponding per-scenario visualization suites used to support that summary. For readability, we include the full 2-D/3-D embedding views colored by epoch, timestep, group identity, and unit index, together with latent phase portraits and readout traces.

Across these settings, MM-PHATE exhibits three qualitative behaviors relevant to the reviewer concern: (i) sensitivity to differences in latent dynamical family, (ii) retention of a family-level distinction under smooth state-space warps, and (iii) preservation of progression structure even when the same latent dynamics are observed through different coordinates. In every composite embedding panel below, the three columns are MM-PHATE, PCA, and t-SNE (left-to-right).

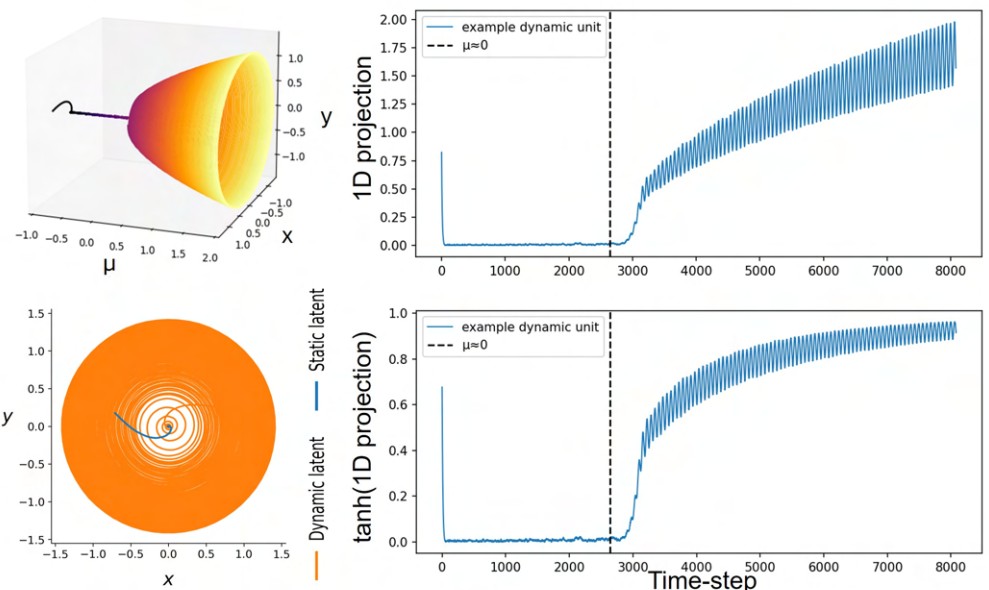

Figure A.5: Latent and readout dynamics of the Hopf bifurcation experiment. (Top left) Latent $(x, y)$ trajectories of the Hopf system as $\mu$ moves from negative (stable focus) to positive (stable limit-cycle), illustrating the bifurcation. (Bottom left) Example hidden-unit-generating trajectories in the 2D latent plane, illustrating how latent dynamics across epochs are converted into group-wise readout trajectories. (Top right) The corresponding one-dimensional scalar readout obtained from $h = \alpha(\boldsymbol{w}^\top[x, y]) + \gamma r$ before optional nonlinearity. (Bottom right) The scalar readout after applying a tanh nonlinearity, which compresses large-amplitude oscillations and highlights saturation in the post-bifurcation regime.

Table 2: Controlled synthetic scenarios used to test dynamics-family differences and smooth state-space warps. Main-text results focus on state-warp settings (no time warp).

| Scenario | Group A | Group B | State warp | What it tests |
|---|---|---|---|---|
| Hopf vs. Pitchfork (clean) | Hopf | Pitchfork | none / none | Family-level dynamical differences without coordinate perturbation |
| Hopf vs. Pitchfork (warped) | Hopf | Pitchfork | yes / yes | Whether family-level differences remain visible when both systems are smoothly warped |
| Hopf clean vs. Hopf warped (warp-only control) | Hopf | Hopf (same latent trajectories before warp) | no / yes | Sensitivity to smooth coordinate perturbation when latent dynamics are otherwise identical |

**(1) Dynamics-family differences without warps.** In the unwarped Hopf-vs.-Pitchfork setting, **Group A is the Hopf bifurcation system** and **Group B is the Pitchfork bifurcation system** (consistent with the scenario name `hopf_bif__pitchfork_bif__no_warp`). MM-PHATE separates the two groups while preserving distinct progression structure across epochs and time-steps (Figs. A.6–A.9). The latent phase portrait confirms that the two groups differ at the level of generating dynamics (rotational organization for Hopf versus branching organization for Pitchfork in this construction), while the readout traces help verify that the observed separation is not explained by a trivial monotonic scaling alone.

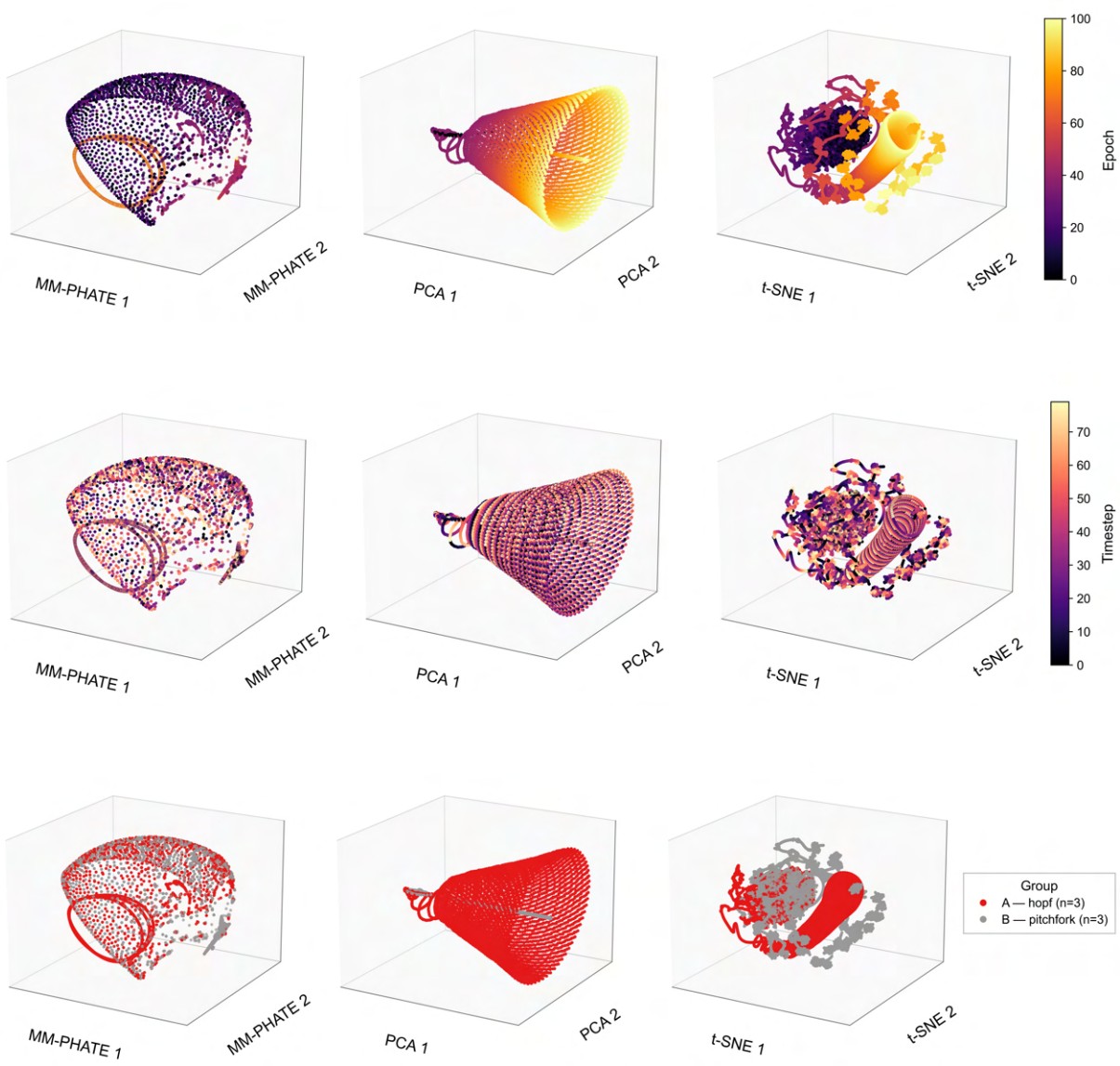

Figure A.6: **Hopf vs. Pitchfork without warps: 3-D embedding views (epoch, timestep, and group identity).** From top to bottom: colorings by training epoch, within-epoch timestep, and group identity (A = Hopf bifurcation, B = Pitchfork bifurcation). Each image is a three-column composite (MM-PHATE, PCA, t-SNE). These views assess progression organization and family-level separation in 3-D.

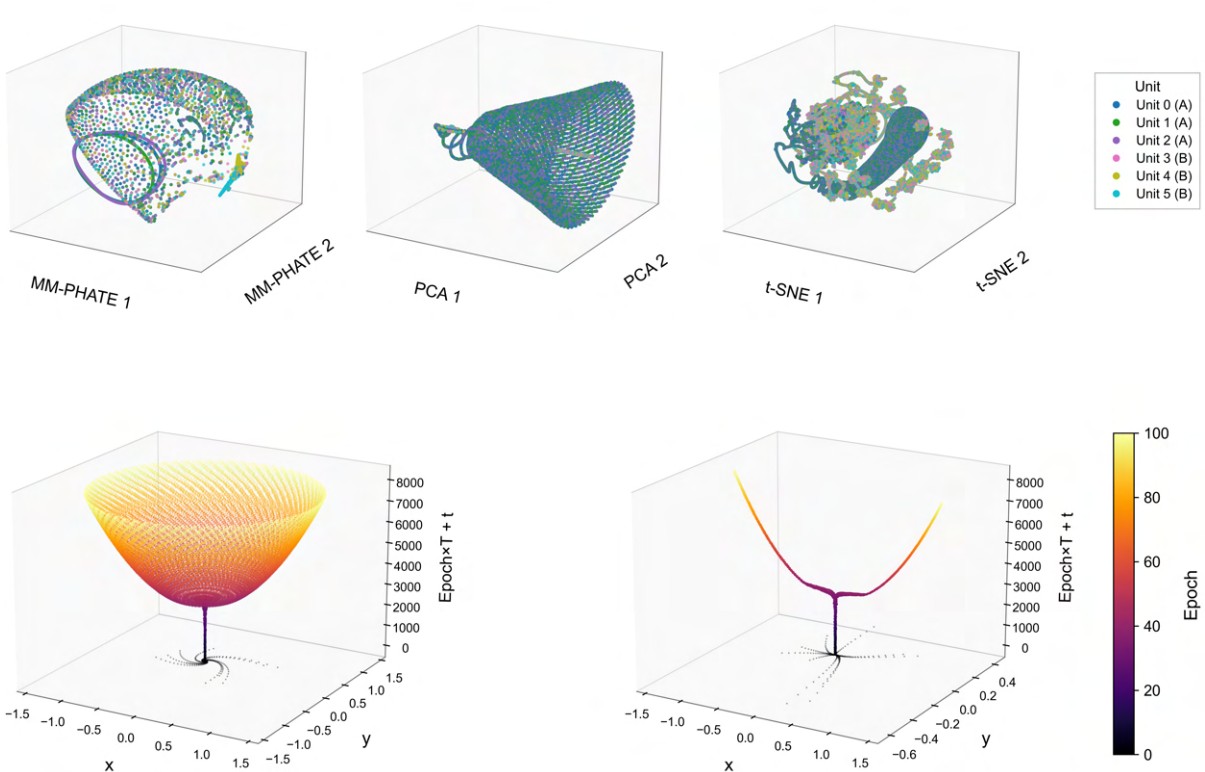

Figure A.7: **Hopf vs. Pitchfork without warps: 3-D unit-level organization and latent phase portraits.** Top: embeddings colored by unit index. Bottom: latent phase portraits for Group A (Hopf bifurcation; left panel) and Group B (Pitchfork bifurcation; right panel), i.e., the ground-truth latent dynamics used to generate the observations.

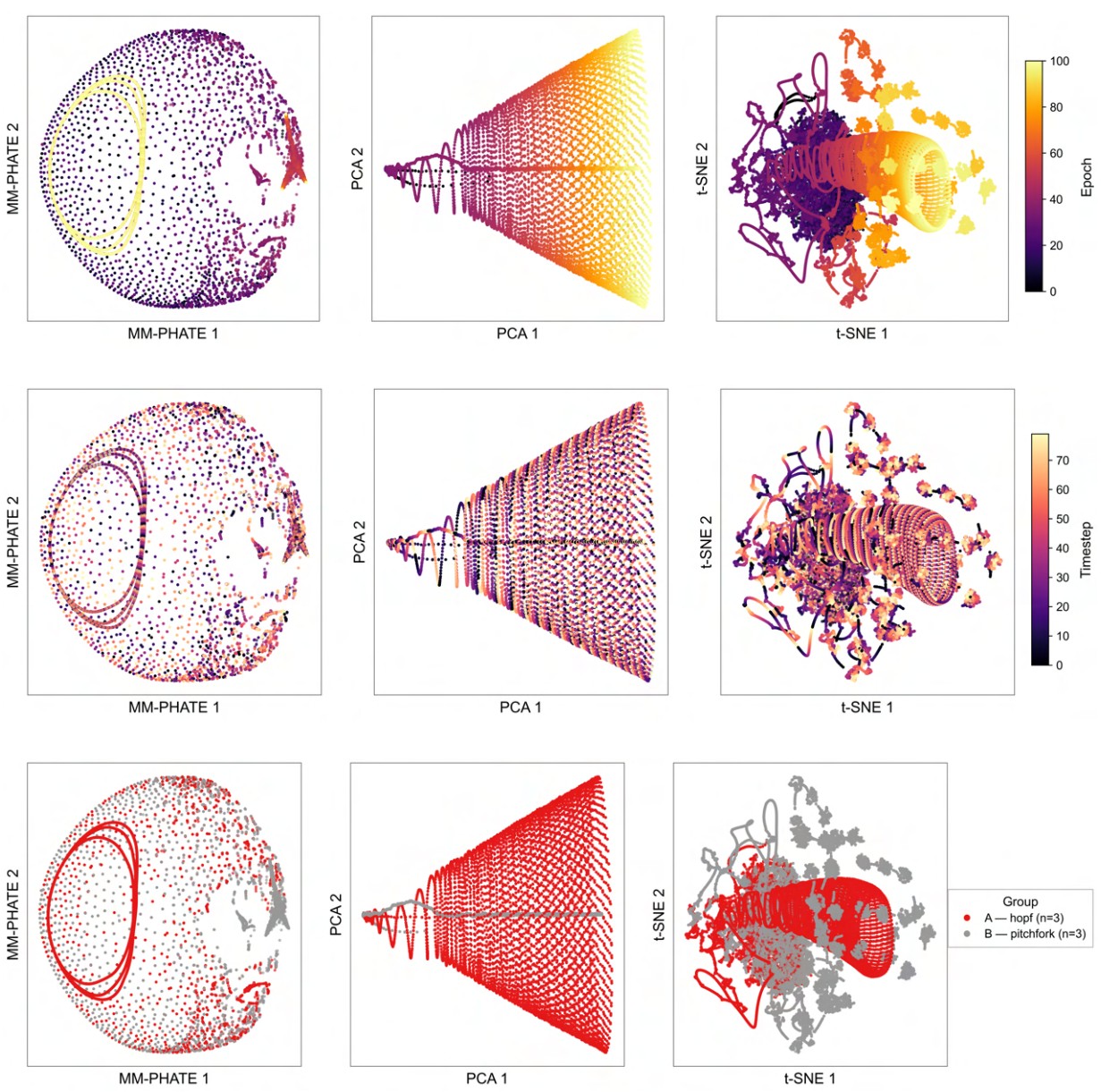

Figure A.8: **Hopf vs. Pitchfork without warps: 2-D embedding views (epoch, timestep, and group identity).** These 2-D projections are the compact counterparts of Fig. A.6 and verify that the same qualitative progression and family-level separation patterns remain visible after projection to two dimensions.

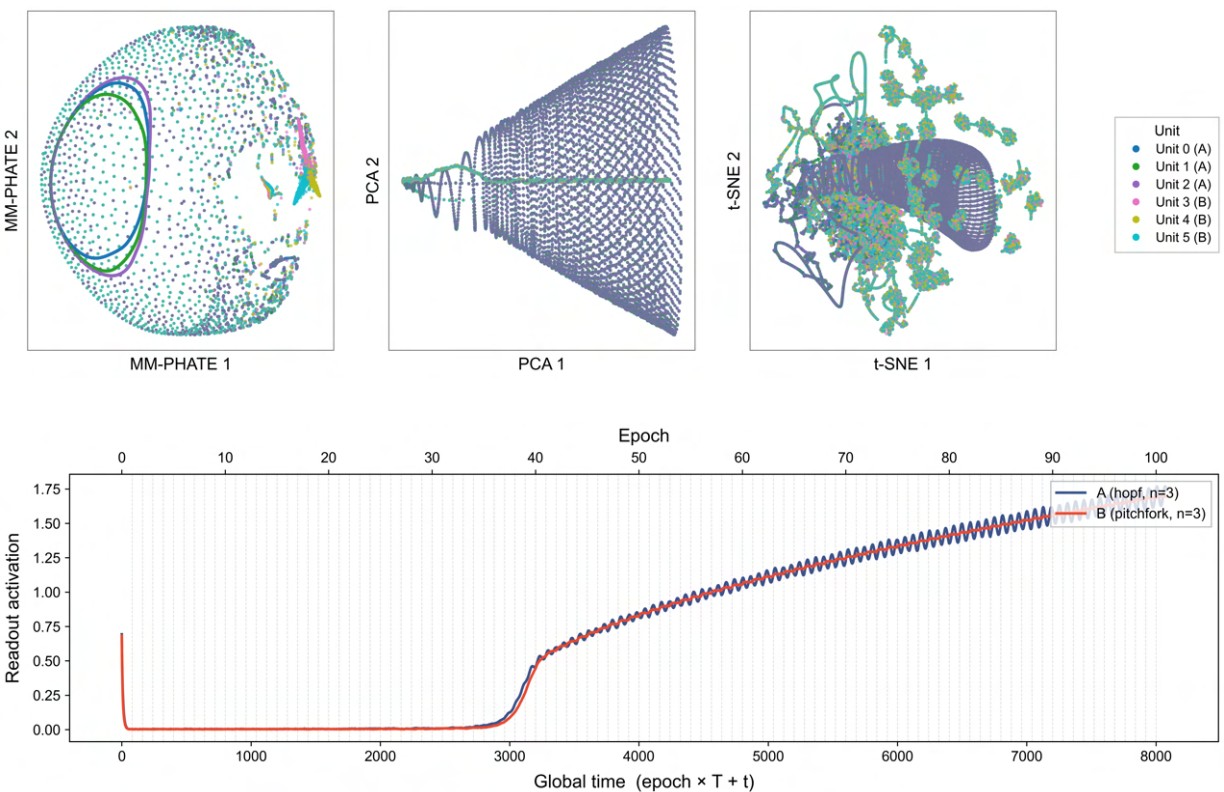

Figure A.9: **Hopf vs. Pitchfork without warps: 2-D unit-level organization and average readout traces.** Top: embeddings colored by unit index. Bottom: average readout traces for groups A and B across global training time. Together with the latent phase portraits in Fig. A.7, these panels support the interpretation that MM-PHATE is sensitive to qualitative dynamics-family differences rather than only generic amplitude growth.

**(2) Dynamics-family differences with smooth state-space warps applied to both groups.** When both systems are subjected to smooth state-space warps prior to readout, **Group A remains the warped Hopf bifurcation system** and **Group B remains the warped Pitchfork bifurcation system** (consistent with `hopf_bif__pitchfork_bif__warp`). MM-PHATE still preserves a visible family-level distinction while retaining epoch/time-step progression structure (Figs. A.10–A.13). This indicates that, in the tested conditions, the embedding is not simply memorizing raw latent coordinates and remains informative under smooth coordinate perturbations. The latent phase portraits and readout traces again provide the mechanistic context for interpreting the embedding geometry.

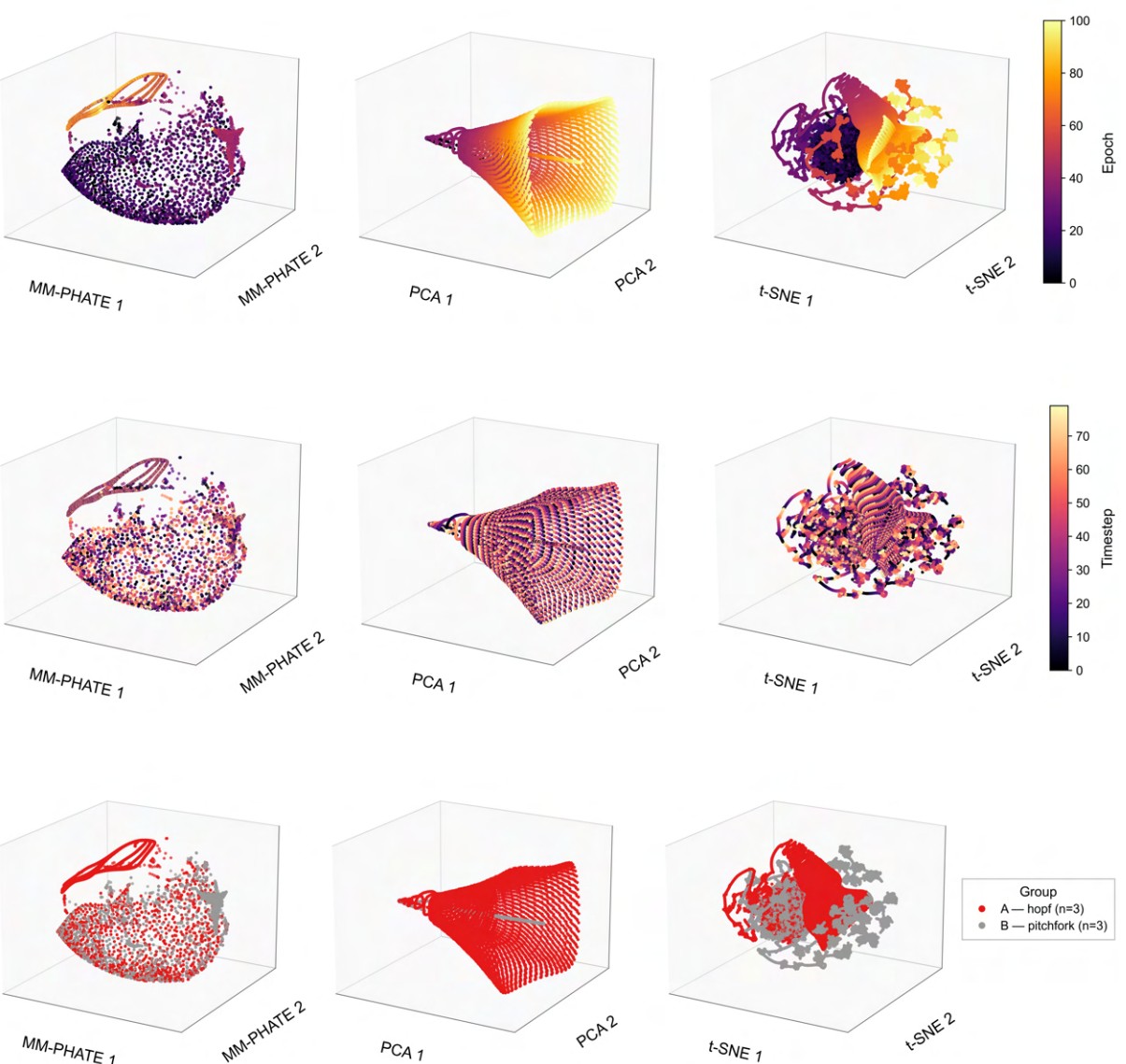

Figure A.10: **Hopf vs. Pitchfork with smooth state-space warps: 3-D embedding views (epoch, timestep, and group identity).** From top to bottom: colorings by training epoch, within-epoch timestep, and group identity (A = warped Hopf bifurcation, B = warped Pitchfork bifurcation). Each image is a three-column composite (MM-PHATE, PCA, t-SNE). Despite coordinate warps, MM-PHATE continues to display organized progression and a visible family-level distinction.

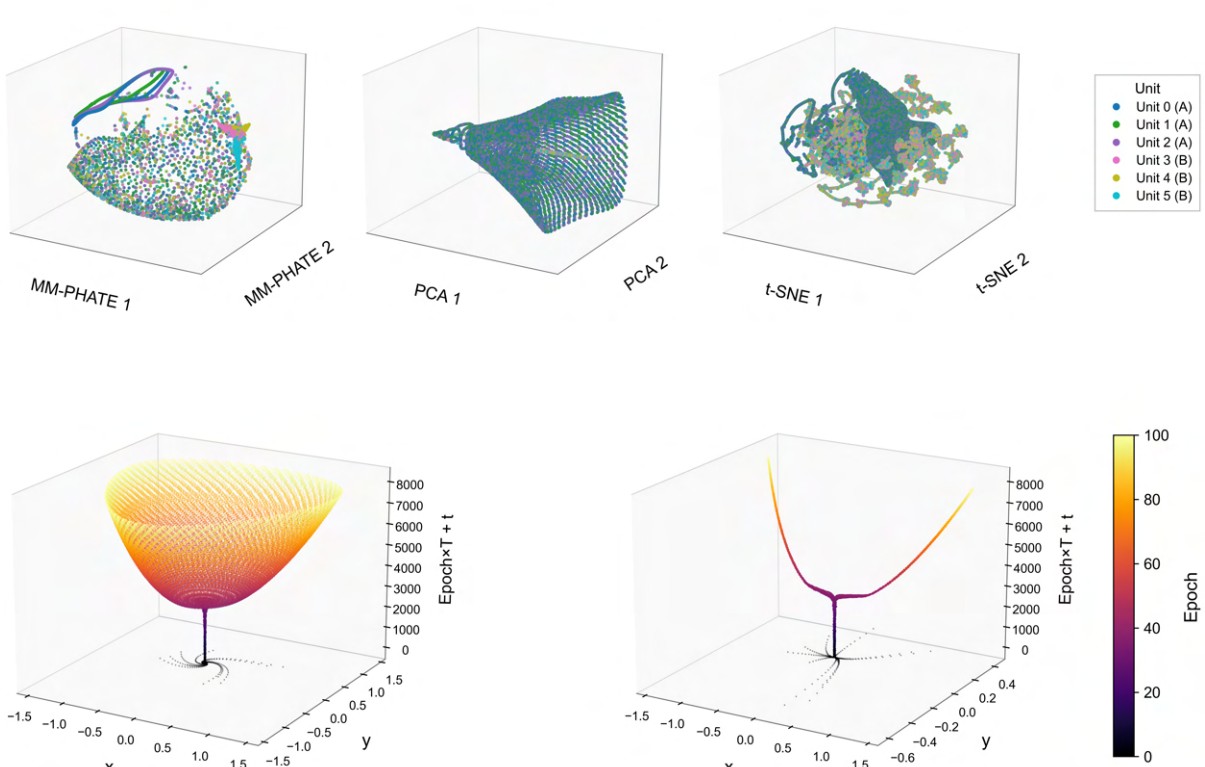

Figure A.11: **Hopf vs. Pitchfork with smooth state-space warps: 3-D unit-level organization and latent phase portraits.** Top: embeddings colored by unit index. Bottom: latent phase portraits for Group A (warped Hopf bifurcation; left panel) and Group B (warped Pitchfork bifurcation; right panel). These panels help verify that the observed family-level separation is not driven by only a small subset of units and remains interpretable relative to the underlying latent trajectories.

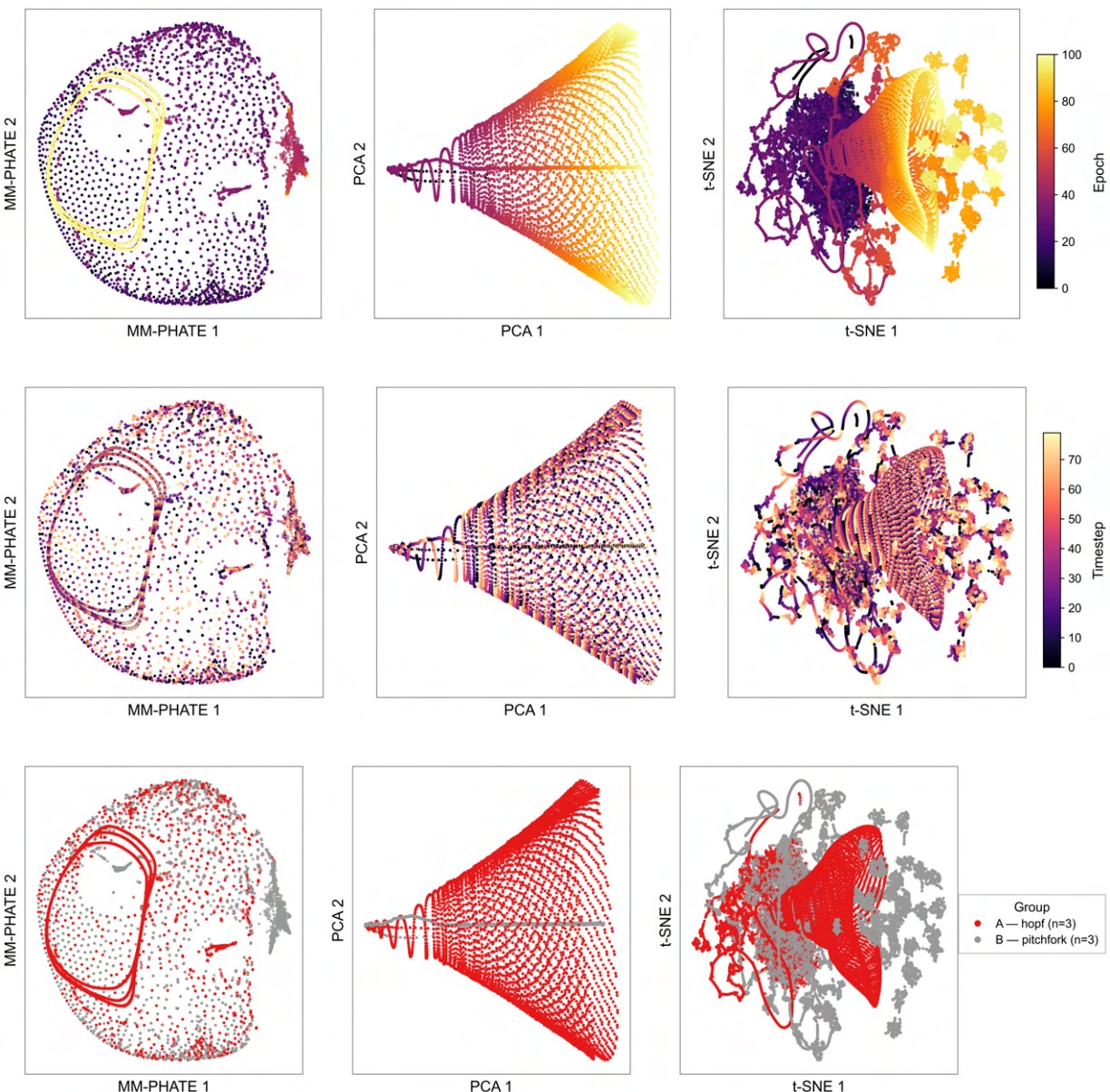

Figure A.12: **Hopf vs. Pitchfork with smooth state-space warps: 2-D embedding views (epoch, timestep, and group identity).** The 2-D projections confirm that the family-level distinction and progression organization remain visible in a compact representation, rather than only in a favorable 3-D view.

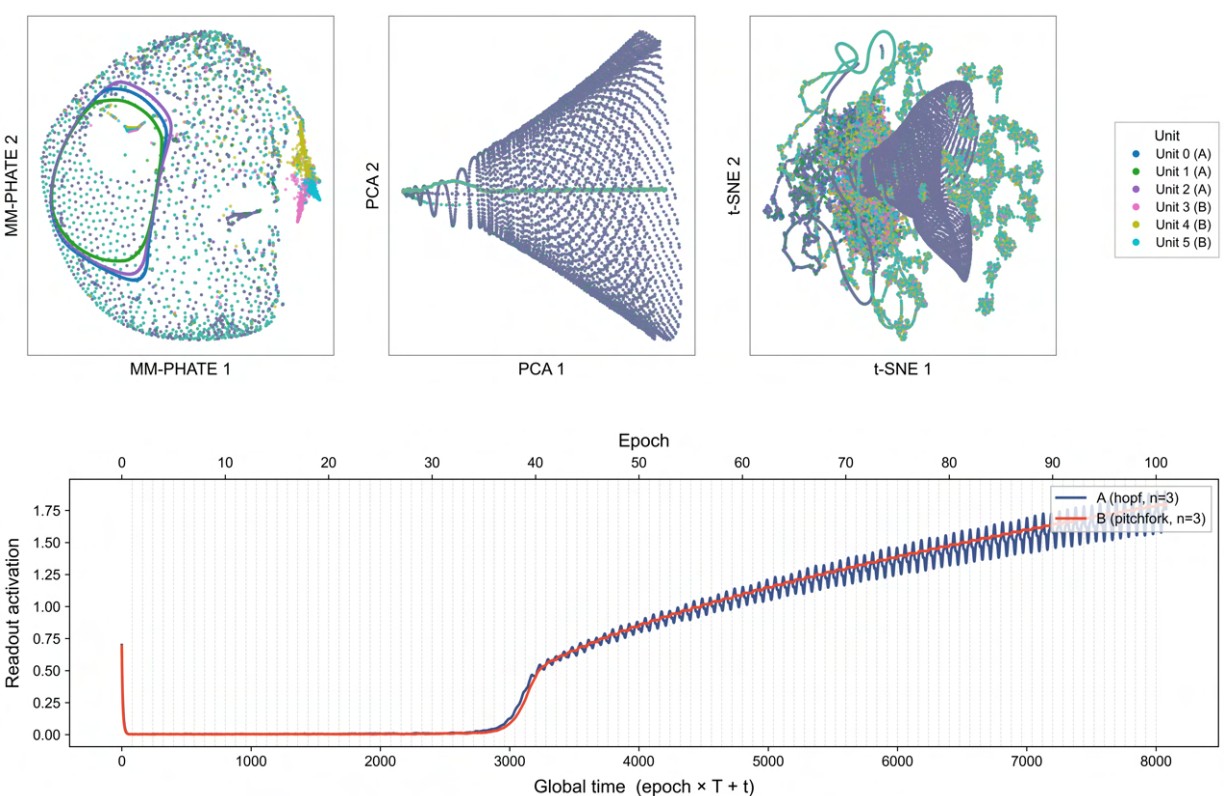

Figure A.13: **Hopf vs. Pitchfork with smooth state-space warps: 2-D unit-level organization and average readout traces.** Top: embeddings colored by unit index. Bottom: average readout traces for groups A and B across global training time. These panels complement the 3-D views and support the claim that MM-PHATE remains informative under smooth coordinate warps in the tested synthetic suite.

**(3) Warp-only control (same latent dynamics, different coordinates).** In the Hopf clean-vs.-Hopf warped control, **Group A is the clean Hopf system** and **Group B is the warped Hopf system** (consistent with `hopf_big_clean_vs_warped`); both groups share the same latent trajectories and $\mu$ schedule before warping by construction. MM-PHATE still separates the clean and warped groups, showing that it is *not* invariant to smooth warps in a strict sense; however, it preserves the qualitative bifurcation progression (including pre-/post-bifurcation organization), and in particular the post-bifurcation limit-cycle structure remains visibly preserved in MM-PHATE across the compared views (Figs. A.14–A.17). By contrast, PCA and t-SNE show stronger warp-induced geometric distortion, which makes the shared Hopf post-bifurcation organization less clear. This is the key qualitative observation motivating the downstream entropy analyses: under smooth coordinate warping, MM-PHATE preserves *qualitative dynamical geometry* (notably the post-bifurcation limit-cycle organization and progression structure) despite lacking strict invariance, whereas PCA and t-SNE more strongly distort these features.

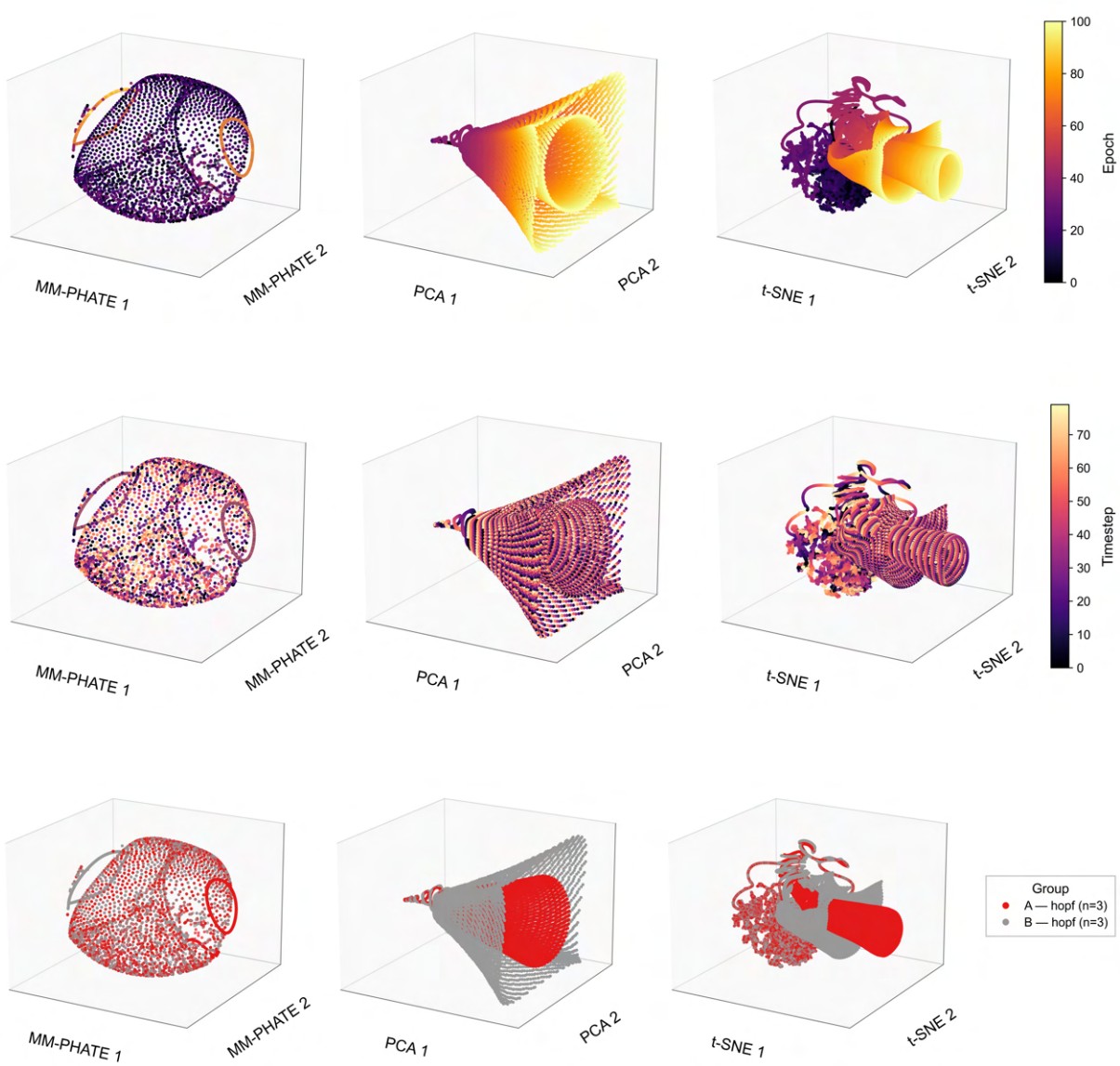

Figure A.14: **Hopf clean vs. Hopf warped (warp-only control): 3-D embedding views (epoch, timestep, and group identity).** From top to bottom: colorings by training epoch, within-epoch timestep, and group identity (A = clean Hopf, B = warped Hopf). Because both groups share the same underlying latent dynamics before warping, the group separation here reflects sensitivity to coordinate warp rather than a change in dynamical family. However, despite separating the clean and warped groups, MM-PHATE still preserves the qualitative post-bifurcation limit-cycle organization, whereas PCA and t-SNE show stronger warp-induced geometric distortion that obscures this shared Hopf structure.

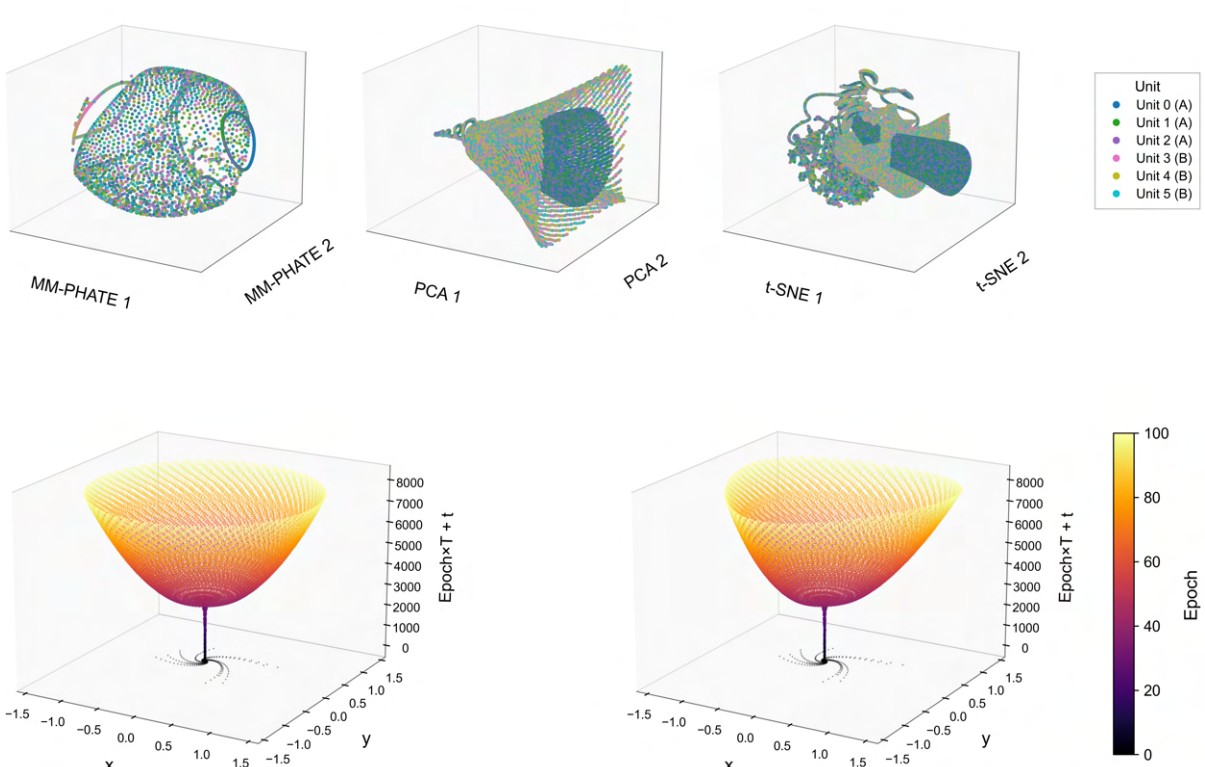

Figure A.15: **Hopf clean vs. Hopf warped (warp-only control): 3-D unit-level organization and latent phase portraits.** Top: embeddings colored by unit index. Bottom: latent phase portraits for Group A (clean Hopf; left panel) and Group B (warped Hopf; right panel). The latent panel verifies that the underlying bifurcation progression is matched by construction prior to warping, making this a direct test of warp sensitivity versus progression preservation.

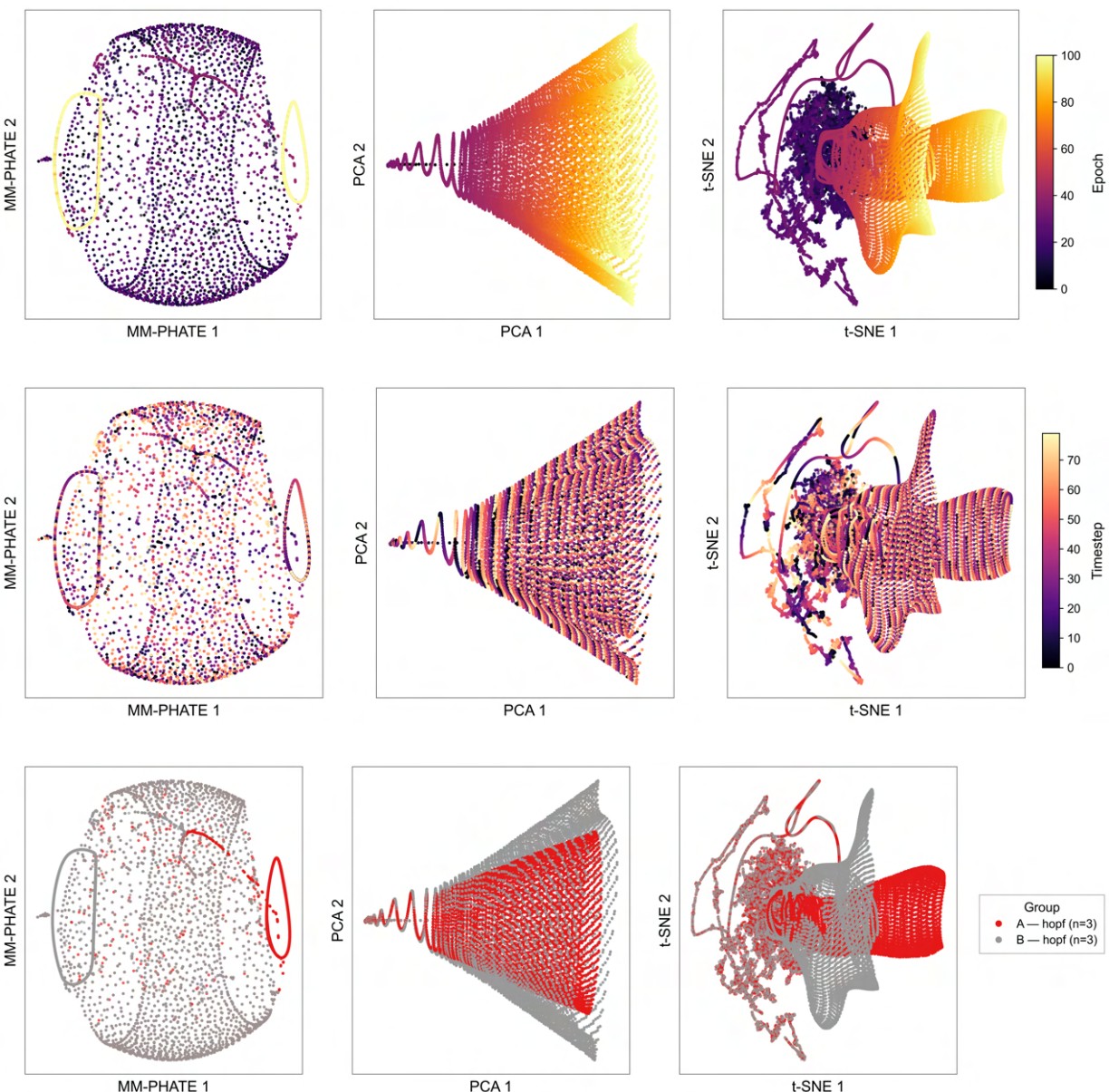

Figure A.16: **Hopf clean vs. Hopf warped (warp-only control): 2-D embedding views (epoch, timestep, and group identity).** The same qualitative interpretation persists in 2-D: MM-PHATE distinguishes the coordinate-warped observation spaces while preserving the temporal/progression organization relevant to the synthetic benchmark analysis.

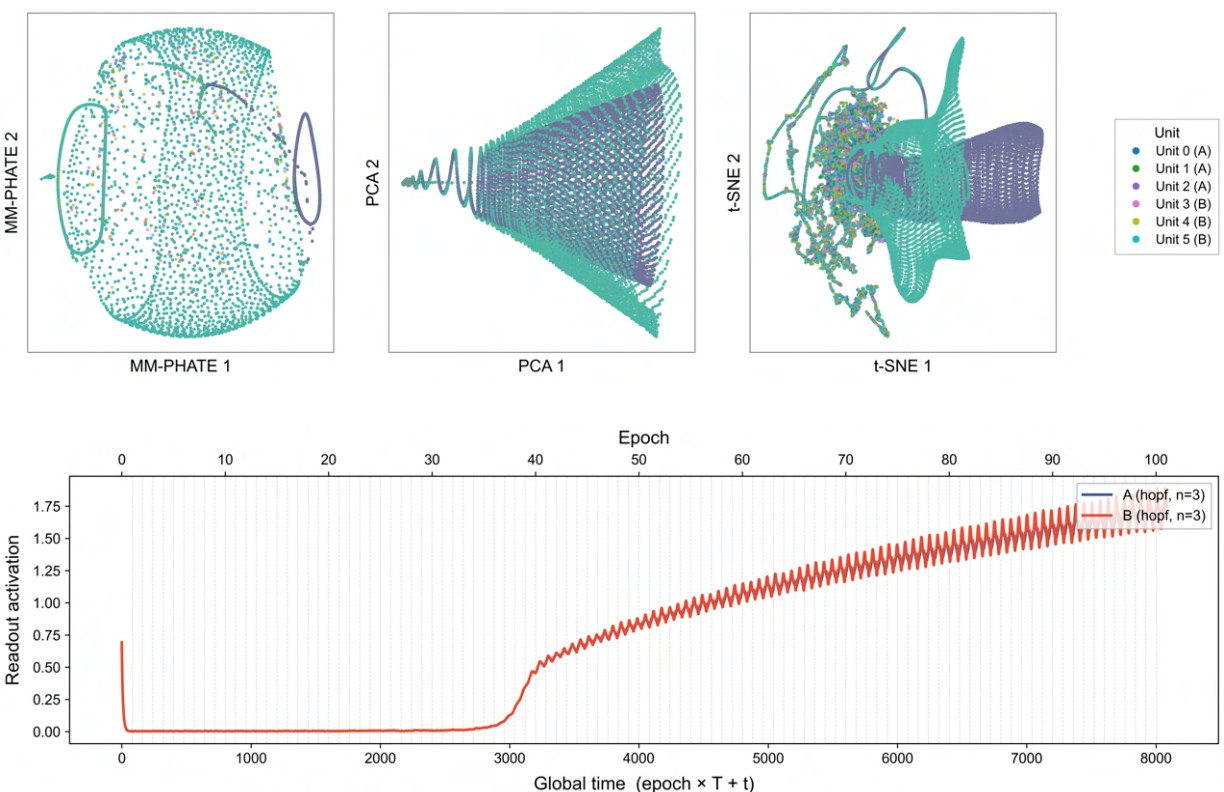

Figure A.17: **Hopf clean vs. Hopf warped (warp-only control): 2-D unit-level organization and average readout traces.** Top: embeddings colored by unit index. Bottom: average readout traces for the clean and warped Hopf groups across global training time. These panels complete the warp-only control and support the claim that MM-PHATE is not strictly warp-invariant, but still retains the qualitative progression structure used in the downstream entropy analyses.

Taken together, these three synthetic scenarios provide controlled qualitative evidence that MM-PHATE can (i) distinguish latent dynamics-family differences, (ii) retain an interpretable family-level distinction under smooth state-space warps, and (iii) preserve progression structure even when the same latent dynamics are observed through different coordinates. We emphasize again that these are qualitative synthetic stress tests and do not constitute a formal invariance guarantee under arbitrary smooth equivalences.

### A.5.4 Hopf anchor benchmark and entropy validation

**Hopf anchor construction (dynamic vs. static Hopf groups).** The Hopf anchor benchmark used for the main-text geometric and entropy validation is distinct from the symmetric 3+3 scenario suite in Section A.5.2. Here, all units are generated from the same supercritical Hopf system, but with two different $\mu$-schedules: a dynamic (swept-$\mu$) group with 6 units, for which $\mu$ is linearly swept from $-1$ to $2$ across epochs, and a static (fixed-$\mu$) control group with 4 units, for which $\mu \equiv -1$ at all epochs. Thus, the dynamic group crosses the bifurcation and develops post-bifurcation limit-cycle dynamics, whereas the static group remains in the pre-bifurcation contracting regime throughout training. Both groups use the same latent ODE family and the same readout construction (up to unit-level noise/jitter), so the key difference is whether the bifurcation is traversed.

The Hopf bifurcation experiment provides a controlled setting for evaluating how embedding methods capture latent geometry, readout distortions, and bifurcation structure.

**Epoch-colored embeddings (Figs. A.18, A.24).** As the dynamic group crosses the Hopf bifurcation, the oscillatory radius increases substantially. PCA, t-SNE, Isomap, LLE, and UMAP primarily encode this amplitude growth, inflating post-bifurcation trajectories and obscuring the bifurcation geometry. MM-PHATE instead collapses post-bifurcation epochs onto a single coherent orbit, revealing the latent limit-cycle structure, rather than its raw magnitude. With a `tanh` readout, conventional methods additionally show saturation effects, while MM-PHATE remains qualitatively unchanged. (Static units are limited for Isomap/LLE due to scalability constraints.)

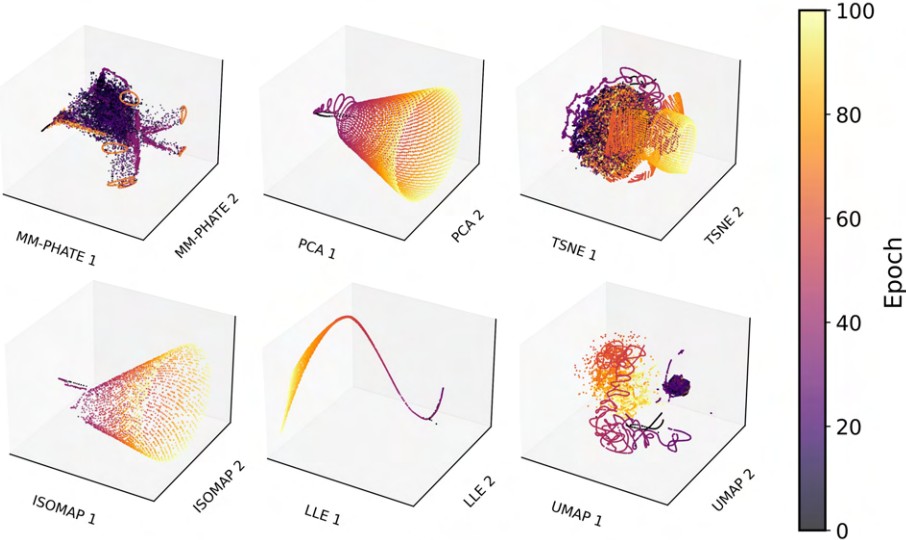

Figure A.18: Embeddings of the Hopf RNN traces colored by epoch using MM-PHATE, PCA, t-SNE, Isomap, LLE, and UMAP (left to right, top to bottom). *Note: For Isomap and LLE, we were unable to include sufficiently many static units due to poor scalability and subsampling constraints.*

**Unit-colored embeddings (Figs. A.19, A.25).** All units read out the same underlying Hopf dynamics but acquire subtle unit-specific variations after the bifurcation. MM-PHATE more clearly resolves these differences, separating individual units along the shared manifold. PCA, t-SNE, UMAP, Isomap, and LLE instead collapse unit trajectories together, both with and without the `tanh` nonlinearity. They show only minor deviations from injected noise and fail to capture the structured unit-level dynamical variation. (Isomap/LLE omit some static units as above.)

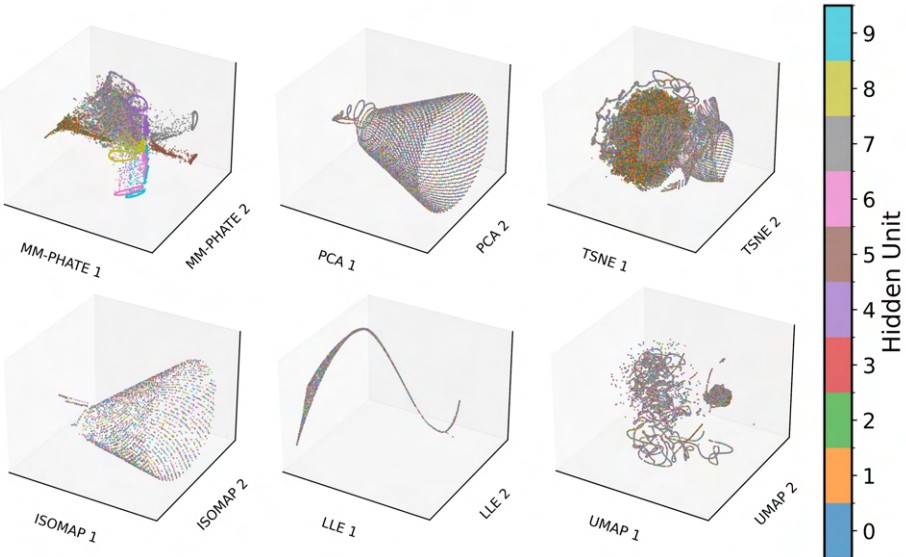

Figure A.19: Embeddings of the Hopf RNN traces colored by hidden-unit identity across six methods. *Note: For Isomap and LLE, we were unable to include sufficiently many static units due to poor scalability and subsampling constraints.*

**Single–dynamic-unit zoom (Fig. A.20).** MM-PHATE organizes the evolution of a single dynamic unit across the entire sweep: contraction for $\mu < 0$, unfolding geometry near $\mu \approx 0$, and a clean limit cycle for $\mu > 0$, with the magnified view showing a smooth circular orbit.

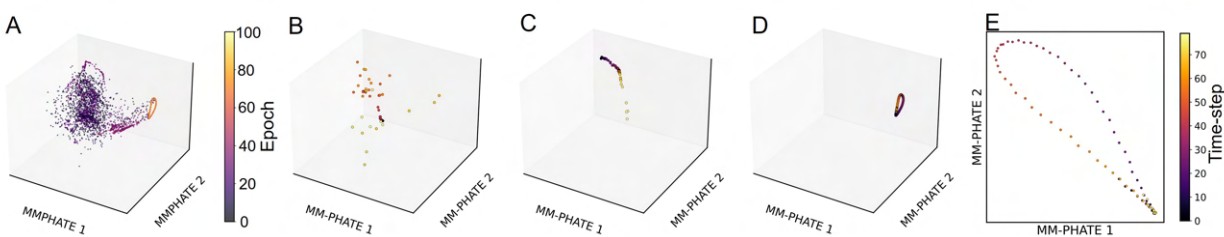

Figure A.20: Zoomed-in views of the MM-PHATE embedding for a single dynamic hidden unit across the Hopf bifurcation. (1) Full trajectory across all epochs, showing the global transition from the pre-bifurcation contraction to the post-bifurcation limit cycle (colored by epochs). (2) An early epoch with $\mu < 0$, where the trajectory collapses toward the fixed point with minimal rotational structure (colored by time-steps). (3) An epoch near the bifurcation point ($\mu \approx 0$), where the geometry begins to unfold and organization emerges (colored by time-steps). (4) A late epoch with $\mu > 0$, where the dynamics form a clear and stable limit cycle in the embedding (colored by time-steps). (5) A magnified view of a single post-bifurcation limit cycle, illustrating the smooth and consistent circular structure recovered by MM-PHATE (colored by time-steps).

**Group-colored embeddings (Fig. A.21).**  Before bifurcation, dynamic units form a diffuse cloud reflecting heterogeneous transients en route to the emerging limit cycle, whereas static units (constant $\mu < 0$) repeatedly collapse to the same fixed point and form a compact cluster. MM-PHATE cleanly differentiates the two groups and reveals their distinct trajectory geometries across the Hopf bifurcation. Before bifurcation ($\mu < 0$), the dynamic units form a broad cloud, reflecting noise-dominated dynamics near the collapsing fixed point and the fact that these units are progressing toward the emergent limit cycle. In contrast, the static units (whose $\mu$ remains fixed $< 0$ across all epochs) accumulate into a denser, more compact structure, since their temporal evolution is repeatedly drawn toward the same contracting fixed point. Post-bifurcation, MM-PHATE continues to separate individual dynamic-unit trajectories, demonstrating that the method captures the full temporal progression of each unit rather than collapsing them into a single manifold. In contrast, the other methods are dominated by amplitude changes and small stochastic differences between units: they produce concentric, trumpet-shaped loops in which trajectories from different units largely overlap, with only subtle separations that primarily reflect noise rather than genuinely distinct dynamical orbits.

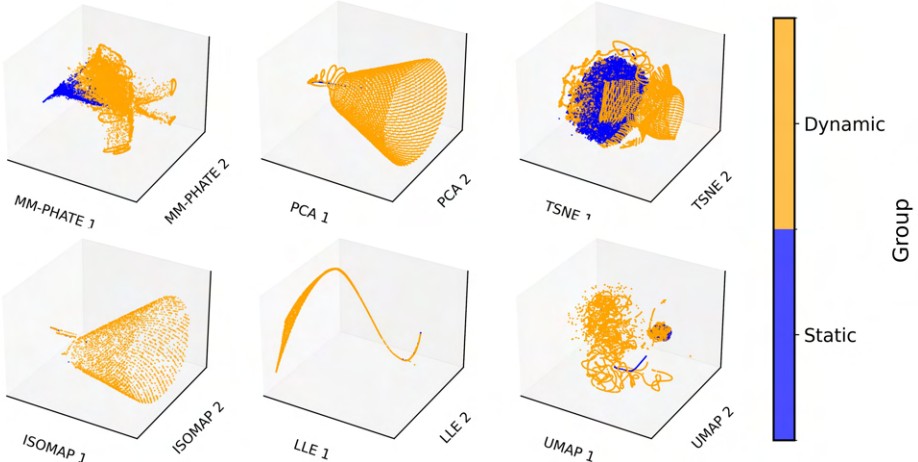

Figure A.21: Embeddings of the Hopf RNN colored by group identity (dynamic vs. static hidden units). *Note: For Isomap and LLE, we were unable to include sufficiently many static units due to poor scalability and subsampling constraints.*

**Intra-step entropy (Fig. A.22).**  Latent intra-step entropy (computed across samples) shows low entropy for static samples and a sharp rise for dynamic samples near the bifurcation. The readout inverts this pattern due to radius-dominated projection. Among the evaluated embeddings in this benchmark, MM-PHATE is the only one that qualitatively reproduces the tested latent entropy structure; PCA, t-SNE, and UMAP instead mirror the readout, indicating their sensitivity to activation magnitudes rather than dynamical organization.

**Inter-step entropy (Fig. A.23).**  Latent inter-step entropy exhibits consistently low entropy for static samples and a clear low-to-high transition for dynamic samples as $\mu$ crosses zero. The readout again shows the opposite pattern. MM-PHATE recovers the latent trend: static units remain low entropy, and dynamic units shift from high entropy in the noise-dominated regime ($\mu < 0$) to low entropy in the limit-cycle regime ($\mu > 0$). t-SNE captures part of this transition but spreads it broadly across epochs, while PCA and UMAP mostly follow the readout's magnitude-driven structure.

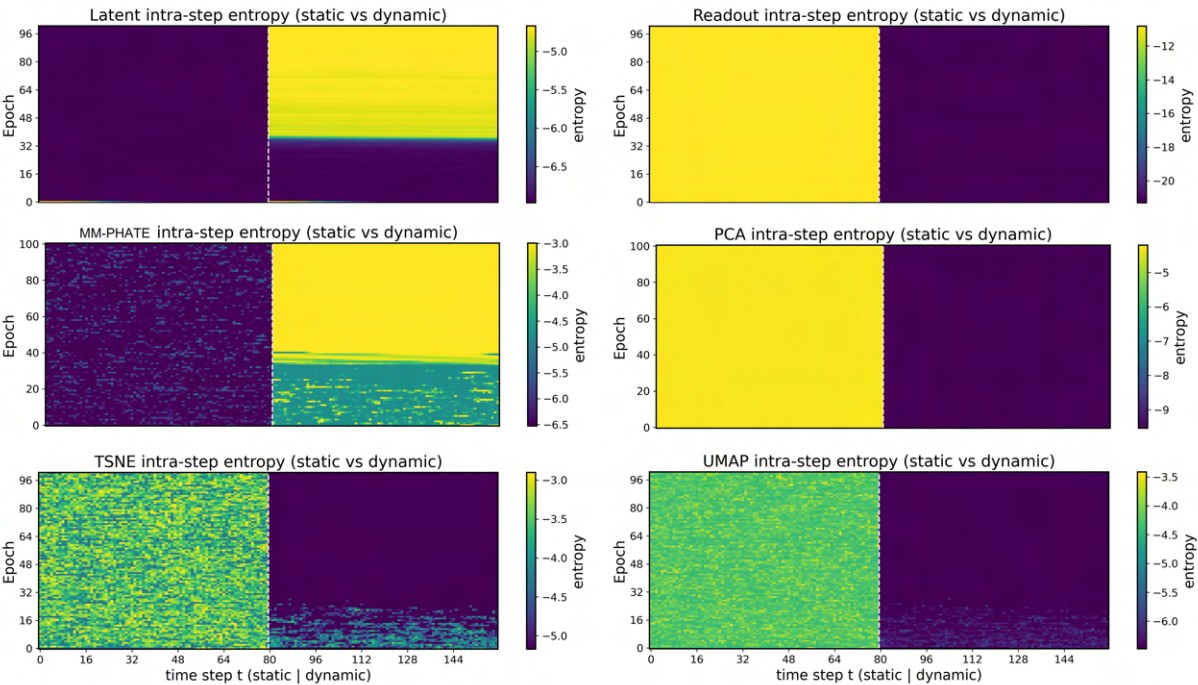

Figure A.22: Intra-step entropy analysis across the Hopf bifurcation. The first column shows ground-truth quantities: (top-left) latent intra-step entropy computed across samples (the latent space has no notion of hidden units), and (top-right) readout intra-step entropy computed across hidden units. The remaining columns show intra-step entropy computed from the MM-PHATE, PCA, t-SNE, and UMAP embeddings. MM-PHATE most closely reproduces the latent-space entropy dynamics: it preserves the consistently low entropy of the static group and captures the sharp transition in the dynamic group from low to high entropy as $\mu$ approaches the bifurcation point. By contrast, PCA, t-SNE, and UMAP fail to reflect the organization present in the latent dynamics, yielding high entropy for the static units and persistently low entropy for the dynamic units—mirroring the readout-level distortions rather than the underlying system dynamics.

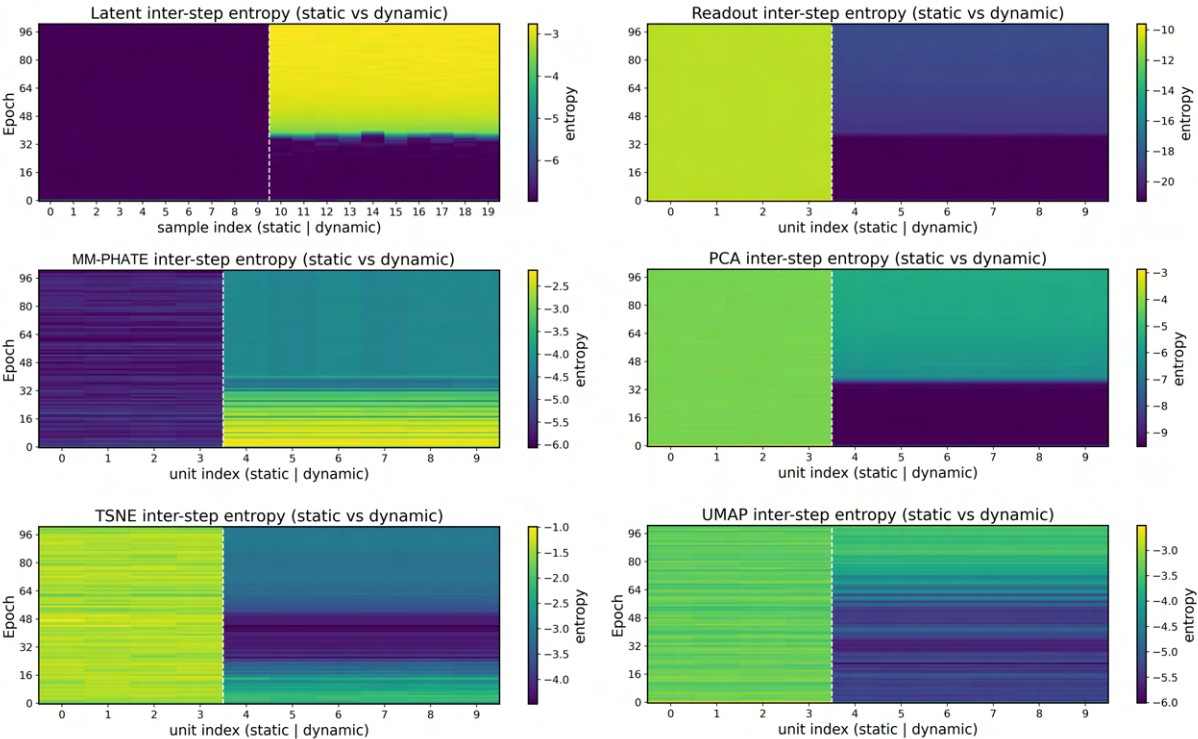

Figure A.23: Inter-step entropy across latent space (top-left), readout space (top-right), and four embedding methods (MM-PHATE, PCA, t-SNE, UMAP; columns 2–3). In the latent ground truth (top-left), the $x$-axis indexes samples, whereas in the readout and all embedding methods the $x$-axis indexes hidden units. Latent trajectories exhibit consistently low entropy for the static group, and a clear transition in the dynamic group from low entropy for $\mu < 0$ to high entropy for $\mu > 0$. The readout shows the opposite static–dynamic pattern, with high entropy for static units and a low-to-high transition for dynamic units, driven by the radius-dominated projection. MM-PHATE is the only embedding that qualitatively recovers the latent structure: static units remain low-entropy across epochs, and dynamic units transition from high entropy in the noise-dominated regime ($\mu < 0$) to low entropy in the limit-cycle regime ($\mu > 0$), indicating that MM-PHATE embeds units according to their full temporal trajectories rather than instantaneous magnitude. PCA and UMAP largely mirror the readout's magnitude-based structure, maintaining high entropy for static units and a low-to-high transition for dynamic units. t-SNE partially captures a high-to-low transition in the dynamic group but spreads this transition across many epochs. In contrast, MM-PHATE produces a sharp and well-localized transition around $\mu \approx 0$ and preserves the distinction between static and dynamic units, faithfully reflecting the underlying bifurcation geometry.

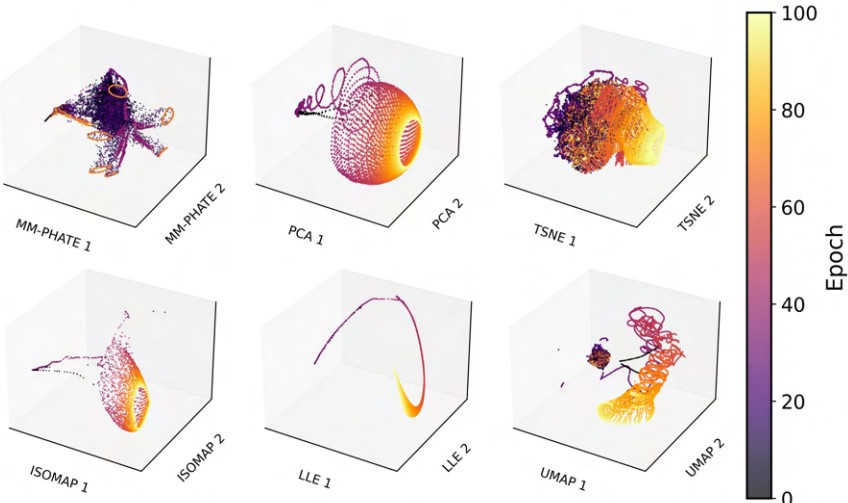

Figure A.24: Embeddings of the Hopf RNN traces with a `tanh` nonlinearity applied to the readout, colored by epoch using MM-PHATE, PCA, t-SNE, Isomap, LLE, and UMAP. The saturation induced by the `tanh` compresses the amplitudes of late-epoch trajectories, causing PCA, t-SNE, Isomap, LLE, and UMAP to exhibit visibly capped or flattened embeddings that primarily reflect this value saturation rather than the underlying dynamical progression. MM-PHATE, however, remains qualitatively unchanged: it continues to collapse post-bifurcation trajectories onto a single coherent orbit and preserves the bifurcation geometry despite the nonlinear distortion of the readout. Although the global appearance of all embeddings remains similar to the non-`tanh` case, these results highlight that conventional methods are sensitive to the absolute scale of activations, whereas MM-PHATE retains robustness to such transformations. *Note: Isomap and LLE cannot include sufficient static units due to poor scalability and subsampling constraints.*

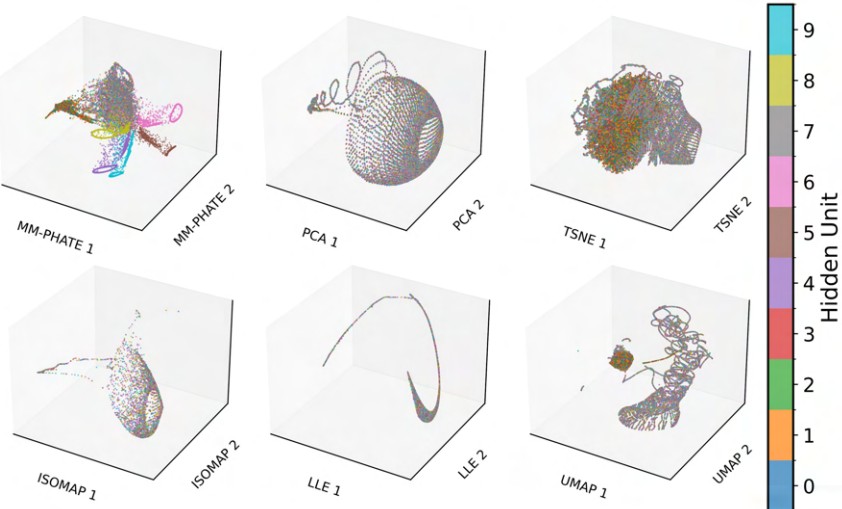

Figure A.25: Embeddings of the Hopf RNN traces with a `tanh` readout nonlinearity, colored by hidden-unit identity. Despite the saturation effects introduced by the `tanh`, MM-PHATE continues to cleanly separate the trajectories of individual dynamic units after the bifurcation, reflecting its ability to organize samples according to their underlying dynamical evolution rather than their raw activation magnitudes. In contrast, PCA, t-SNE, UMAP, Isomap, and LLE mix the unit trajectories together, with embeddings largely shaped by value saturation rather than dynamical structure. *As in the epoch-colored plots, Isomap and LLE omit some static units due to scalability and subsampling constraints.*

### A.6   Neighborhood Preservation Analysis

We assess how faithfully each embedding preserves the local geometric relationships present in the activation tensor $T \in \mathbb{R}^{(ns) \times m \times p}$ by computing *intra-step* and *inter-step* neighborhood preservation. Each time–epoch pair $(\tau, \omega)$ corresponds to an intra-step slice $T_{\tau,\omega} \in \mathbb{R}^{m \times p}$ containing the activations of all $m$ hidden units across $p$ samples.

**Common evaluation grid.**   Because different methods operate on different subsampled sets of $(\tau, \omega)$ pairs (e.g., UMAP and LLE require subsampling, Isomap requires even more aggressive subsampling), we evaluate all methods on the *intersection* of available slices. Isomap is excluded from the main comparison due to its extremely small intersection set under feasible subsampling.

**Intra-step neighborhood preservation.**   For each slice $(\tau, \omega)$ and hidden unit $i$, we compute its $k$ nearest neighbors among $\{t_{\tau,\omega,j}\}_{j=1}^{m}$ in the z-scored activation space and among $\{y_{\tau,\omega,j}\}_{j=1}^{m}$ in the embedding. The intra-step preservation score is the average fraction of overlapping neighbors across all slices and units:

$$\mathrm{NP}_{\text{intra-step}} = \mathbb{E}_{\tau,\omega,i} \left[ \frac{|\mathcal{N}_k^{\text{orig}}(\tau, \omega, i) \cap \mathcal{N}_k^{\text{emb}}(\tau, \omega, i)|}{k} \right].$$

**Inter-step neighborhood preservation.**   For each hidden unit $i$, we consider its trajectory across time and epochs,

$$\{t_i(\tau, \omega)\}_{(\tau, \omega)},$$

and compute the $k$ nearest neighbors of each slice in both the original and embedded spaces (excluding self-neighbors). The inter-step score is the average neighbor overlap across units and slices:

$$\mathrm{NP}_{\text{inter-step}} = \mathbb{E}_{i,\tau,\omega} \left[ \frac{|\mathcal{N}_k^{\text{orig}}(i; \tau, \omega) \cap \mathcal{N}_k^{\text{emb}}(i; \tau, \omega)|}{k} \right].$$

**Reported results.**   For each method and $k \in \{5, 10, 15, 20, 40\}$, we report

$$(\mathrm{NP}_{\text{intra-step}}, \mathrm{NP}_{\text{inter-step}})$$

computed on the exact same intersection of $(\tau, \omega)$ pairs, ensuring that differences reflect the embedding quality rather than differences in sampling or data coverage.

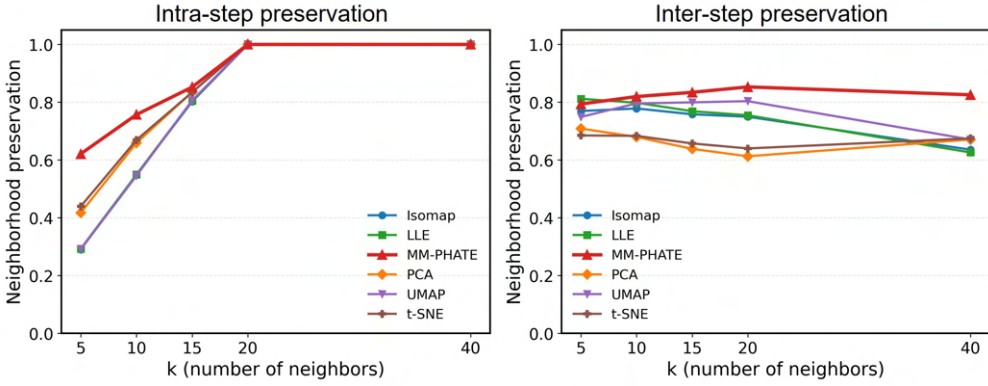

Figure A.26: Area2Bump: Neighborhood preservation of visualization methods including Isomap.

Table 3: Neighborhood preservation of visualization methods (Without Isomap).

| k | | MM-PHATE | PCA | t-SNE | UMAP | LLE |
|---|---|---|---|---|---|---|
| 5 | Intra-step | **0.649** | 0.475 | 0.515 | 0.300 | 0.299 |
| 5 | Inter-step | 0.671 | 0.519 | 0.574 | 0.657 | **0.703** |
| 10 | Intra-step | **0.789** | 0.717 | 0.741 | 0.546 | 0.546 |
| 10 | Inter-step | **0.672** | 0.473 | 0.544 | 0.659 | 0.647 |
| 15 | Intra-step | **0.859** | 0.860 | 0.863 | 0.814 | 0.812 |
| 15 | Inter-step | **0.667** | 0.466 | 0.528 | 0.630 | 0.599 |
| 20 | Intra-step | 1.000 | 1.000 | 1.000 | 1.000 | 1.000 |
| 20 | Inter-step | **0.660** | 0.471 | 0.517 | 0.594 | 0.560 |
| 40 | Intra-step | 1.000 | 1.000 | 1.000 | 1.000 | 1.000 |
| 40 | Inter-step | **0.634** | 0.511 | 0.534 | 0.476 | 0.426 |

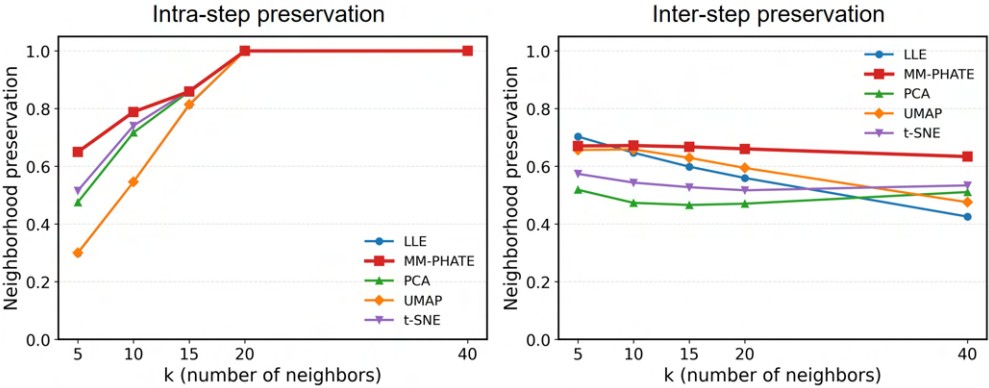

Figure A.27: Area2Bump: Neighborhood preservation without Isomap

## A.7 Time-resolved linear probing analysis

To quantify how much task-relevant information is *linearly decodable* from recurrent representations at different positions in the sequence, we performed a time-resolved linear probe analysis on the hidden states $h_t$ of the recurrent network $R$. Let $X$ denote the supervised dataset with labels $L$, and let $s$ be the total number of time-steps in the sequence. At a collection of training epochs $\tau \in \{1, \ldots, n\}$ (subsampled for efficiency), we evaluated the trained network on the fixed training and test splits and extracted the sequence of hidden states $\{h_t\}_{t=1}^s$ for each input example.

**Per-time-step linear probes.** For each probed epoch $\tau$ and each time-step $\omega \in \{1, \ldots, s\}$, we formed a feature matrix by collecting the hidden state at that time-step across examples, yielding $H^{(\tau,\omega)} \in \mathbb{R}^{N \times m}$, where $m$ is the number of hidden units and $N$ is the number of evaluated examples. We then fit a multinomial logistic regression (linear classifier) to predict the class labels $L$ from $H^{(\tau,\omega)}$ using the training split, and evaluated the resulting probe on both the training and test splits. This yields time-resolved probe accuracies $\mathrm{Acc}^{\mathrm{train}}(\tau, \omega)$ and $\mathrm{Acc}^{\mathrm{test}}(\tau, \omega)$, which quantify the extent to which label information is linearly accessible from the representation at a specific time-step and epoch.

**Heatmap summaries over epoch and time.** We visualize $\mathrm{Acc}^{\mathrm{test}}(\tau, \omega)$ and $\mathrm{Acc}^{\mathrm{train}}(\tau, \omega)$ as heatmaps (Fig. A.28) with axes (epoch $\tau$) $\times$ (time-step $\omega$). These summaries reveal where in the sequence and when during training label information becomes linearly separable in the hidden state. In our experiments, later time-steps exhibit a steady increase in test decodability over training and reach a higher, more stable plateau, whereas early time-steps show more transient test decodability that peaks mid-training and later degrades.

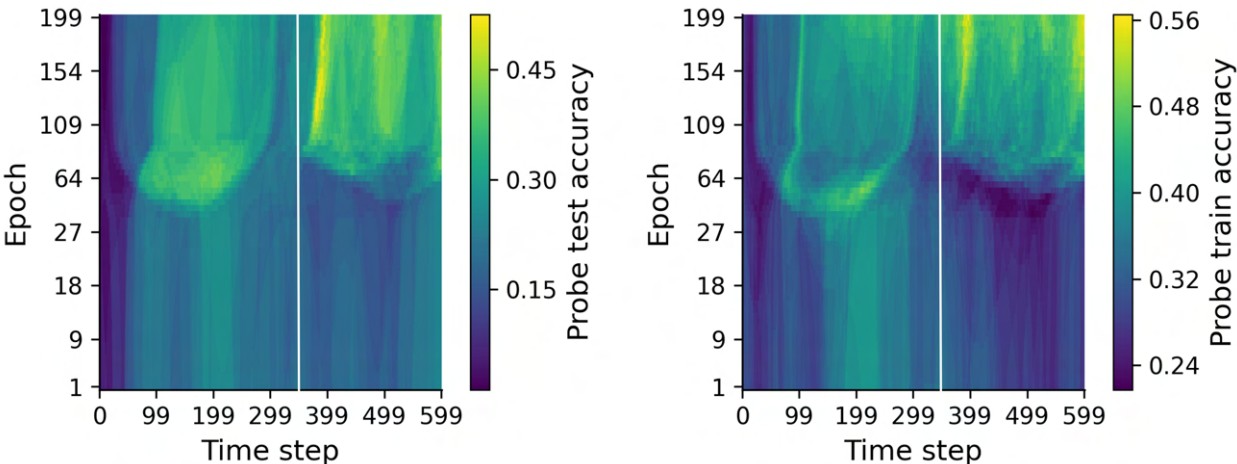

Figure A.28: Heatmaps showing linear probe accuracies across training epochs and time-steps. Left: Probe test accuracy, illustrating the model's performance on the test set for each time-step at different epochs. Right: Probe train accuracy, showing the model's performance on the training set across epochs and time-steps. Both heatmaps visualize the evolution of probe accuracy over time, with the color intensity representing accuracy values.

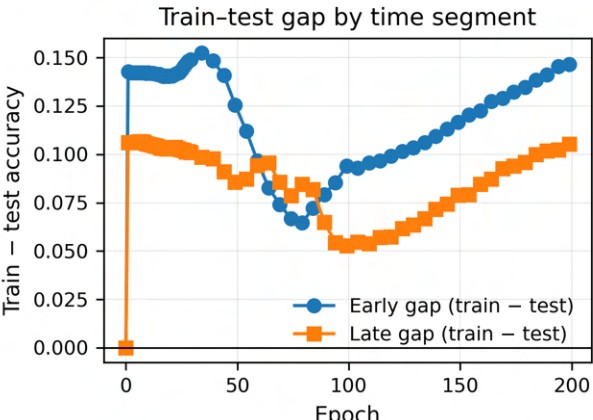

Figure A.29: Train-test accuracy gap over training epochs, separated into early and late time segments. The plot compares the difference between training and testing accuracies for early and late time steps, illustrating how the gap evolves throughout the training process. Early time steps are represented by circles, while late time steps are shown with squares. A larger gap typically indicates overfitting, while a smaller gap suggests more generalization.

**Early/late aggregation and train–test gap.** To align the probe results with the early-vs-late comparisons used elsewhere in the paper, we further aggregate probe accuracies over an "early" and "late" segment of the sequence (defined by a fixed boundary time-step = 350 based on divergence of intra-step entropy trends). For each epoch $\tau$, we compute mean probe accuracies within each segment for both training and test splits, and report the corresponding generalization gaps (Fig. A.29):

$$\text{Gap}_{\text{early}}(\tau) = \overline{\text{Acc}}_{\text{early}}^{\text{train}}(\tau) - \overline{\text{Acc}}_{\text{early}}^{\text{test}}(\tau), \qquad \text{Gap}_{\text{late}}(\tau) = \overline{\text{Acc}}_{\text{late}}^{\text{train}}(\tau) - \overline{\text{Acc}}_{\text{late}}^{\text{test}}(\tau).$$

We visualize these quantities in Fig. A.29 to highlight when and where overfitting emerges in the sequence. Consistent with the main-text findings, the early-segment gap increases substantially after mid-training,

coinciding with the deterioration of early time-step test probe accuracy, whereas the late segment maintains higher test decodability with a smaller train–test gap.

Overall, the time-resolved probe analysis provides a complementary validation that the representational patterns observed in the MM-PHATE entropy analyses track task-relevant information: label signal becomes increasingly decodable at later time-steps, while early time-steps eventually exhibit higher variability that does not translate into improved generalization.

### A.8    Time-step ablation analysis

To assess how the *trained* recurrent classifier $R$ uses information across the sequence, we performed a time-step ablation analysis in which portions of the input sequence were masked at evaluation time while keeping the network weights $W, b$ fixed. Let $X$ denote the evaluation dataset with labels $L$, and let $s$ be the total number of time steps. For a set of training epochs $\tau$ (subsampled consistently with the entropy and probing analyses), we evaluated classification accuracy under three input conditions:

1. **Full sequence (intact):** the original input sequence is provided to the network.

2. **Early-only:** time steps $\omega > \omega_0$ are zeroed (late segment ablated), preserving only the early portion of the sequence.

3. **Late-only:** time steps $\omega \leq \omega_0$ are zeroed (early segment ablated), preserving only the late portion of the sequence.

Here $\omega_0$ (equals to 350 in the example) is a fixed boundary time step defining the early/late split (the same split used to summarize probe results). For each epoch $\tau$, we load the corresponding trained parameters and compute accuracy on both the training and test splits under each masking condition. Because masking is applied only at evaluation time, differences in accuracy directly reflect the extent to which the *existing trained decision rule* relies on early versus late parts of the sequence, rather than changes in training dynamics.

Figure A.30 reports the training-set accuracy across epochs for the intact model and the two ablated settings. Consistent with the main-text interpretation, early-only performance rises during early training but does not sustain the intact model's final performance, while late-only performance improves gradually over training yet remains below the intact sequence. These results not only indicate that final performance depends on integrating information across the full sequence but also provide insight into how the network's reliance on different sequence segments evolves during training. Early time-steps provide shortcuts that are initially leveraged by the classifier, but as training progresses, the reliance shifts toward later time-steps, culminating in a model that integrates information across the entire sequence for robust generalization.

Taken together, these results indicate that although both segments can support nontrivial classification when isolated, the strongest performance depends on coherent integration across the full sequence, and the network's reliance gradually migrates from early to later time-steps as training continues.

### A.9    Time-resolved mutual information analysis

To measure the *total* task-relevant label information present in the recurrent representation, independent of any particular readout, we computed a time-resolved mutual information (MI) between class labels $L$ and the hidden state $h_t$ of the recurrent network $R$. Let $s$ denote the total number of time steps and $m$ the number of hidden units. At a set of probed training epochs $\tau$ (subsampled consistently with the entropy and probing analyses), we extracted the hidden-state tensor on the training set and estimated MI at each time step and hidden unit.

**Per-time-step, per-unit MI.**    For each epoch $\tau$ and time step $\omega \in \{1, \ldots, s\}$, we considered the hidden state vector across examples and treated each unit as a continuous feature. We estimated the mutual information between $L$ and each unit activity at $(\tau, \omega)$ using a nonparametric estimator for continuous features, producing a time–unit MI array. We then summarized the overall label information at each time

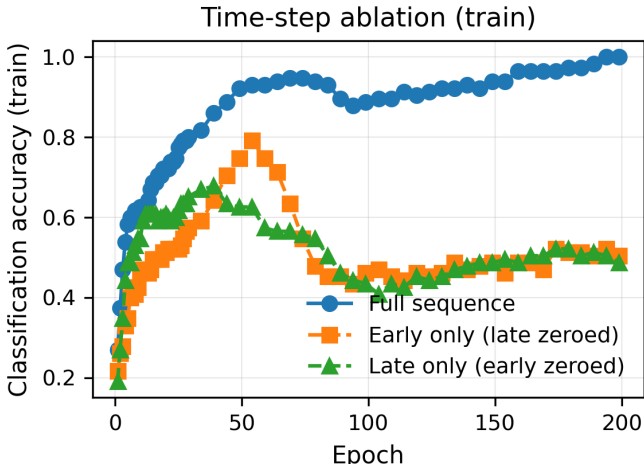

Figure A.30: Time-step ablation on the training split across training epochs. We compare the intact model (full sequence) to *early-only* evaluation (late time steps zeroed) and *late-only* evaluation (early time steps zeroed). The intact model consistently achieves higher accuracy than either ablated condition, indicating that final performance relies on integrating information across the full sequence rather than any single segment alone. The gradual shift from early reliance to full-sequence integration reveals the network's evolving strategy for learning and generalization.

step by averaging MI across units, yielding a time-resolved MI curve $I_\tau(\omega)$ that reflects how much label information is present in $h_\omega$ at epoch $\tau$, regardless of whether the trained classifier exploits it.

**Heatmap summary across epochs and time steps.** Figure A.31 visualizes the mean-over-units MI $I_\tau(\omega)$ as a heatmap with axes (epoch $\tau$) × (time step $\omega$). This representation highlights how label information accumulates and redistributes across the sequence during training. In our experiments, MI increases over training and is consistently higher at later time steps, corroborating the probe and ablation results that later portions of the sequence carry more task-relevant information in the learned hidden dynamics (Fig. 6).

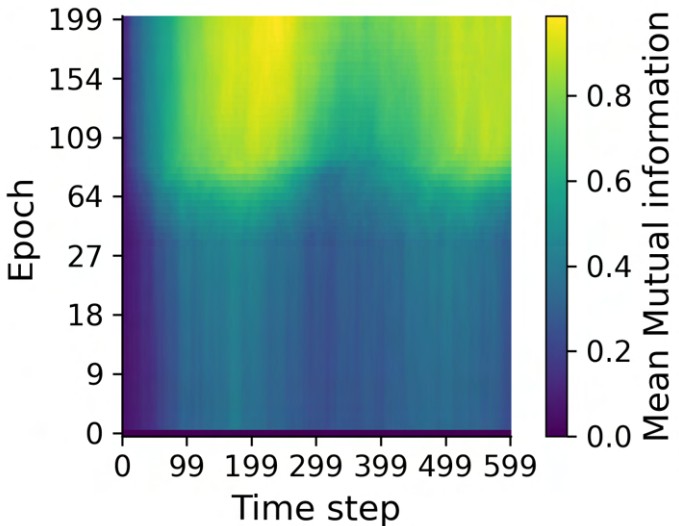

Figure A.31: Time-resolved mutual information between labels $L$ and the recurrent hidden state $h_\omega$, summarized by averaging MI across hidden units $(m)$ at each epoch $\tau$ and time step $\omega$. Warmer colors indicate greater label information present in the representation at that time and training stage. The heatmap shows increasing label information over training, with consistently higher MI at later time steps.

### A.10 Random Label Analysis

#### A.10.1 Experimental setup

To probe how our MM-PHATE and entropy summaries behave in an extreme overfitting regime, we performed a random-label experiment inspired by Fischer (2020). The idea is to progressively destroy the correspondence between inputs and labels and ask whether the resulting representational changes are reflected in MM-PHATE geometry and intra-/inter-step entropy.

We trained multiple 1-layer LSTM models with 100 hidden units on the Area2Bump dataset under different label-shuffling conditions. For each condition, a specified number of classes had their labels randomly reassigned while the remaining classes retained their true labels. For each shuffle level, we drew 10 independent random class-permutation configurations. All models were trained with the same hyperparameters (learning rate $10^{-4}$, batch size and optimizer as in Section 5.2), and most runs reached (near-)zero training loss within 200 epochs, while training on shuffled labels is known to severely degrade test performance.

We focused on the final training epoch and extracted hidden activations across all 600 time-steps. On this epoch we computed: (i) MM-PHATE embeddings of the hidden units across time and epoch, (ii) intra-step and inter-step entropies in the MM-PHATE space (as defined in Section 5.2), and (iii) summary statistics and distributional distances of the intra-step entropy across time-steps, using a clean-label model as a reference.

Specifically, for each shuffle level we summarized the intra-step entropy at the last epoch by: (i) maximum entropy across time-steps, (ii) mean entropy across time-steps, (iii) entropy at the last time-step, and (iv) variance of entropy across time-steps. We then compared the entropy distributions over time-steps between shuffle levels using Jensen–Shannon divergence (JSD) and total variation distance (TVD), both computed relative to a reference model trained on correct labels.

#### A.10.2 Metrics and analysis

**Intra-step entropy.** As in Section 5.2, intra-step entropy is computed at each time-step by applying a KDE-based estimator to the cloud of embedded hidden units at that time-step and epoch. It is therefore a measure of geometric dispersion across units in the MM-PHATE embedding, not a direct estimate of information-theoretic entropy of neural responses. In this experiment, we use its summary statistics as a compact way to track how the spread of unit representations across time-steps changes as label noise increases.

**Inter-step entropy.** Inter-step entropy is computed per unit and epoch, capturing how dispersed a unit's embedded trajectory is across time-steps. Larger inter-step entropy indicates that the unit differentiates more strongly across time within an epoch; smaller values indicate more temporally homogeneous activity. Here we examine how these unit-level temporal profiles change with label shuffling.

**Distributional distances (JSD, TVD).** To quantify how much the overall entropy profile at the last epoch deviates from the clean-label regime, we treat the intra-step entropy values across time-steps as a one-dimensional empirical distribution and compute: (i) the Jensen–Shannon divergence between this distribution and that of the clean-label model, and (ii) the corresponding total variation distance. These distances do not by themselves identify "good" or "bad" representations, but they provide a simple way to measure how strongly the entropy profile shifts as labels are progressively randomized.

#### A.10.3 Results

**MM-PHATE geometry.** With no or minimal label shuffling, MM-PHATE produces embeddings that show structured trajectories across epochs and a relatively organized arrangement of hidden units over time (Fig. A.32). As more classes are shuffled, these structures become progressively less differentiated: trajectories appear more tangled across epochs and units, and the embedding shows weaker separation consistent with the loss of stable, task-aligned structure. This qualitative change is not meant as a fine-grained diagnostic of class structure, but it demonstrates that MM-PHATE geometry is sensitive to the transition from meaningful learning to fitting random targets.

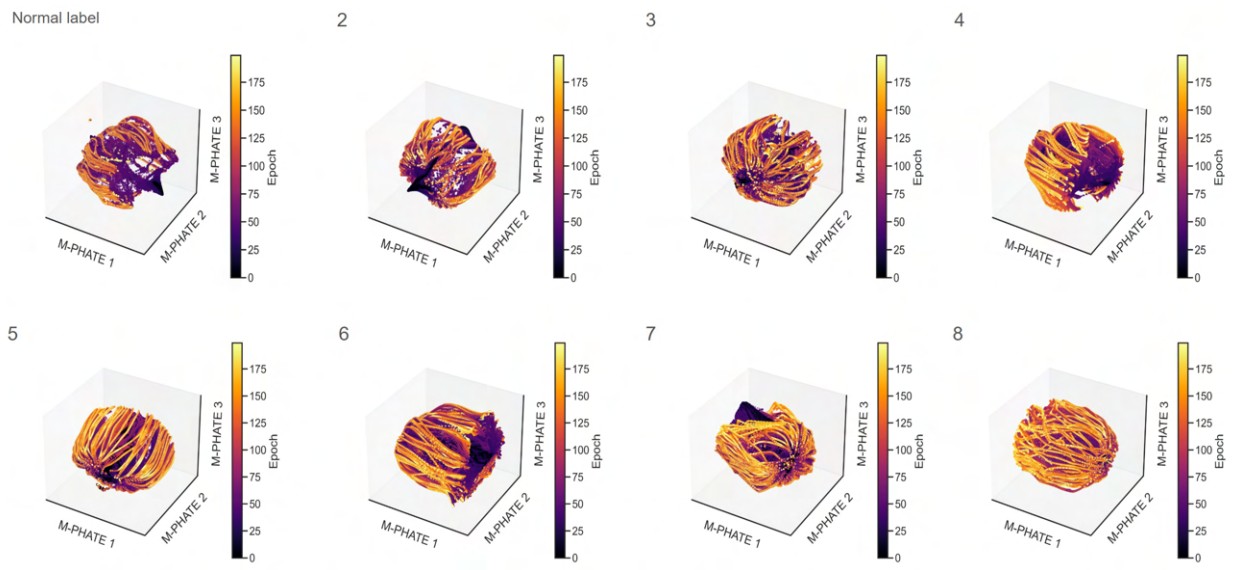

Figure A.32: Random label experiment: MM-PHATE visualization of 100-unit LSTM networks trained for 200 epochs on Area2Bump training data with different numbers of shuffled classes (indicated in the legend). Each point is a hidden unit at a given time-step and epoch. Points are colored by epoch.

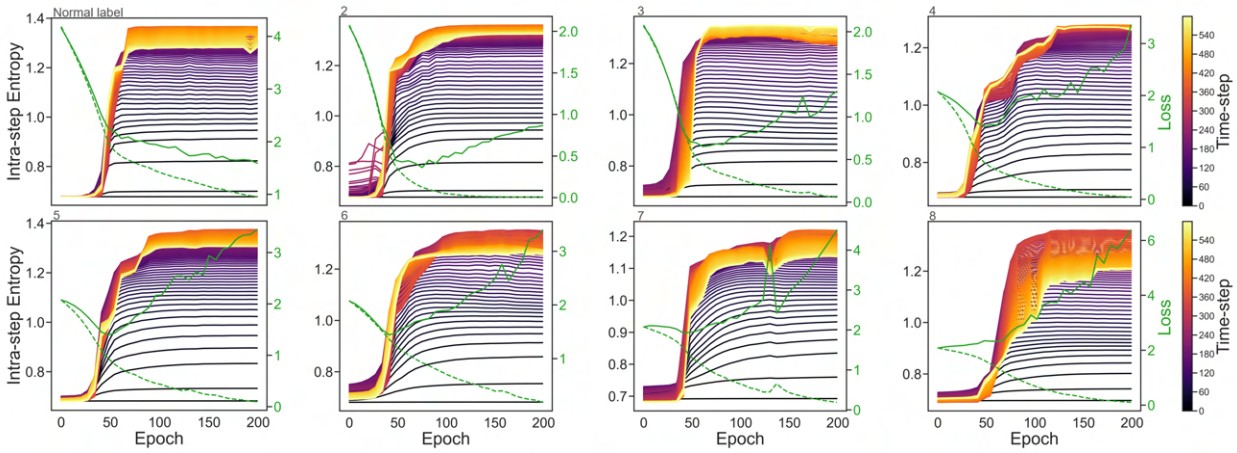

Figure A.33: Random label experiment: intra-step entropy of the MM-PHATE embedding for 100-unit LSTM networks trained for 200 epochs on Area2Bump, with different numbers of shuffled classes. Each curve shows the entropy at a given time-step across training epochs.

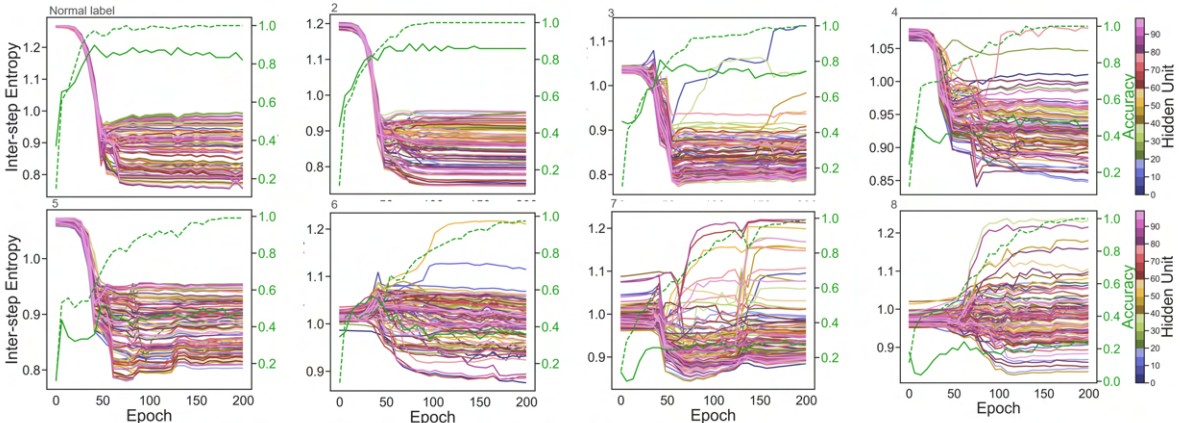

Figure A.34: Random label experiment: inter-step entropy of the MM-PHATE embedding for 100-unit LSTM networks trained for 200 epochs on Area2Bump, with different numbers of shuffled classes. Each curve shows the entropy trajectory of a hidden unit across time-steps at different epochs.

**Intra-step and inter-step entropy trends.** Across shuffle levels, intra-step entropy and inter-step entropy exhibit systematic changes (Figs. A.33–A.34). In the clean-label condition, intra-step entropy over training shows the structured temporal patterns described in Section 5.2. As more classes are shuffled, these patterns are attenuated and the entropy trajectories across epochs and time-steps become more homogeneous. At the final epoch, summary statistics of the intra-step entropy (maximum, mean, last time-step value, and variance) decrease with increasing label shuffling (Fig. A.35), indicating that the embedding-space spread of unit representations across time-steps becomes more concentrated and less temporally differentiated under heavy label noise.

Inter-step entropy traces also change with shuffling. With true labels, units exhibit distinct temporal profiles over time-steps; as shuffling increases, many inter-step entropy trajectories become larger and less structured across epochs (Fig. A.34), consistent with units varying over time in ways that are less clearly aligned with the training phases seen in the clean-label regime. We interpret these trends as evidence that the temporal organization captured by MM-PHATE entropy summaries is disrupted when the network is forced to fit inconsistent labels.

**Distributional distances.** Finally, JSD and TVD between the intra-step entropy distributions (over time-steps) at the last epoch and the corresponding clean-label reference increase as more classes are shuffled (Fig. A.36). A similar trend is observed when computing these distances on the embedded point clouds themselves. These distances capture, in a single scalar per condition, how far the entropy profile has moved away from the structure observed in the clean-label model.

Overall, the random-label experiment serves as a stress test: when we deliberately destroy the input–label relationship, MM-PHATE geometry and the associated entropy summaries change in a systematic way. While we do not claim these measures uniquely quantify "representation quality," their sensitivity to this controlled overfitting regime supports their use as indicators of regime changes and loss of task-aligned structure in the main experiments.

## A.11 Area2Bump with GRU

Here is the same analysis as section 5.2 using GRU (Fig. A.37, A.38, A.39). Other parameters were kept the same.

From these figures, it is evident that PCA and t-SNE present similar visualizations of the hidden dynamics across different network architectures, while MM-PHATE distinctly captures the unique learning behaviors of each model. Consistent with Section 5.2, PCA displays an increasing intra-step entropy even after model

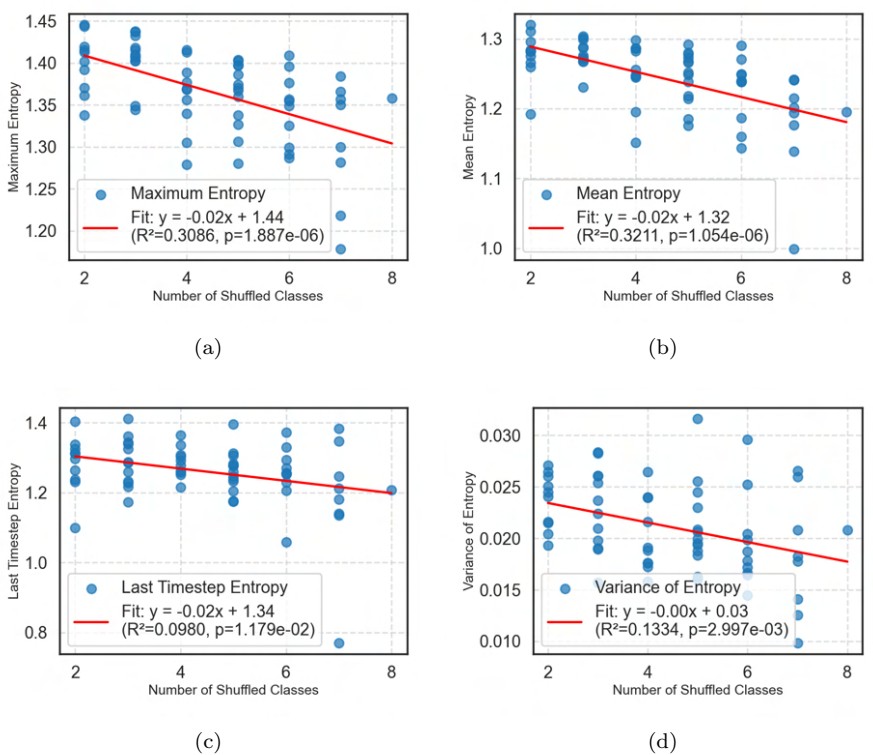

Figure A.35: Random label experiment: (a) maximum, (b) mean, (c) last time-step, and (d) variance of intra-step entropy at the last epoch for models trained with different numbers of shuffled classes.

accuracy has plateaued, and t-SNE produces a noisy visualization. In contrast, MM-PHATE more clearly aligns its transitions well with the learning curve. Notably, the GRU model's representation appears more compact and organized compared to the LSTM model, potentially reflecting its superior performance and reduced overfitting.

### A.12 Area2Bump with Vanilla RNN

Here is the same analysis as section 5.2 using vanilla RNN (Fig. A.40, A.41, A.42). Other parameters were kept the same.

From these figures, it is evident that regardless of the network architectures, PCA exhibits a revolving pattern with overly smooth transitions across epochs, while t-SNE produces a noisy visualization. In contrast, the MM-PHATE visualization reveals that the Vanilla RNN displays a more chaotic pattern compared to the LSTM and GRU models, which is likely associated with its reduced performance and increased overfitting. Furthermore, the intra-step entropies of MM-PHATE show reduced variation across time-steps, indicating that the model struggles to process the input data effectively to generate meaningful representations.

### A.13 Area2Bump with LSTM of Various Sizes

Here we repeat the same MM-PHATE analysis as in section 5.2 using LSTM of various sizes (Fig. A.44, A.45, A.46). Other parameters were kept the same. These results demonstrate that MM-PHATE consistently captures smooth yet distinct transitions across epochs and time-steps, regardless of the LSTM network size. The intra- and inter-step entropy analyses further reveal that these transitions closely correlate with performance changes throughout training. Specifically, we observe a general increase in intra-step entropy as models begin to overfit, suggesting that the networks increasingly memorize input information. In contrast, inter-step entropy shows a significant decline as overfitting progresses, reflecting a loss of sensitivity to input

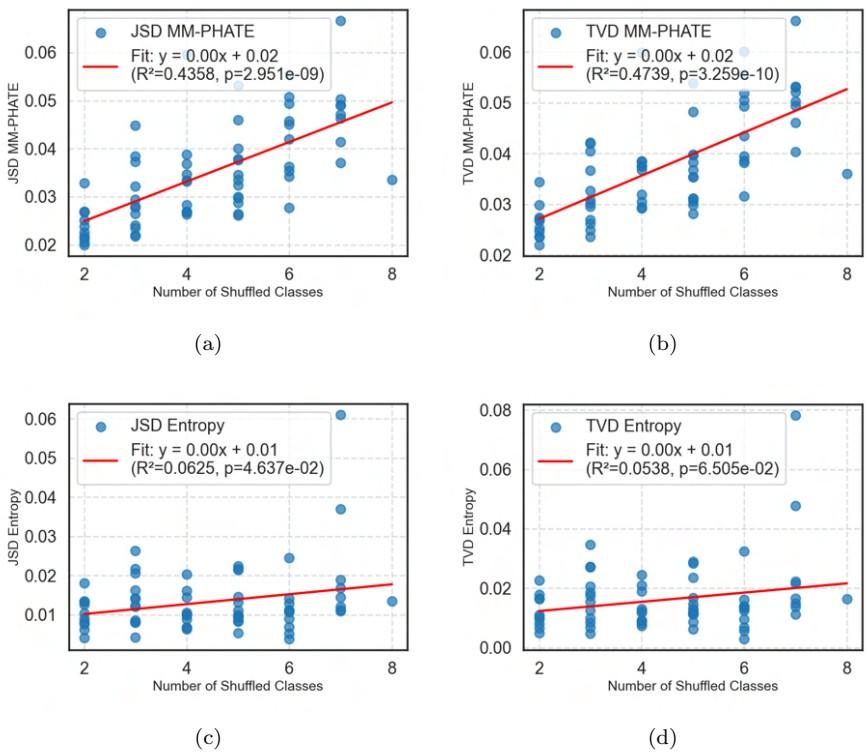

Figure A.36: Random label experiment: Jensen–Shannon divergence (JSD) and total variation distance (TVD) of the MM-PHATE embedding and intra-step entropy distribution at the last epoch, comparing models trained with different numbers of shuffled classes to a clean-label reference model.

changes over time. The loss curve shows worse overfitting as the network size increases, and the networks' inter-step entropy becomes less structured.

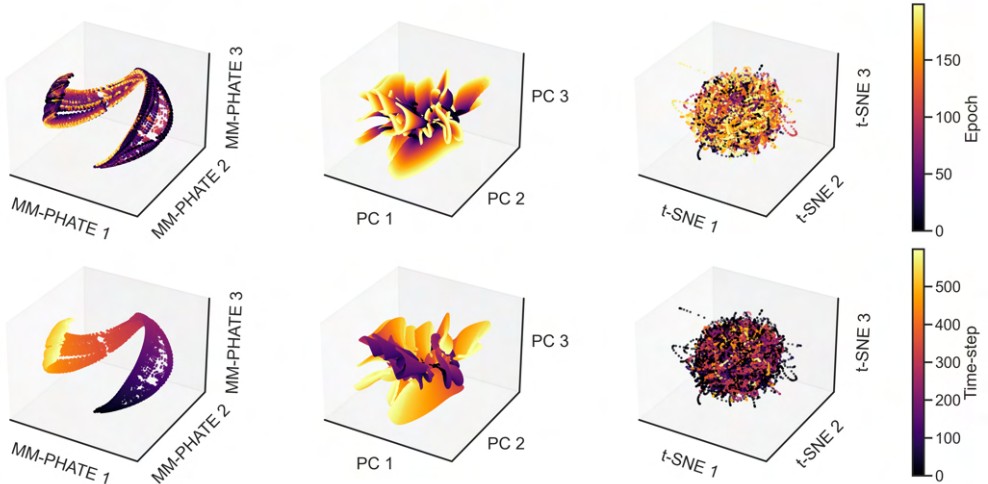

Figure A.37: Area2Bump GRU: Visualization of a 20-unit GRU network trained for 200 epochs. Each point represents a hidden unit at a specific time-step in a given epoch throughout the entire training process. The visualizations are generated using MM-PHATE, PCA, and t-SNE, from left to right, respectively. Points are colored based on epoch (top row) or time-step (bottom row)

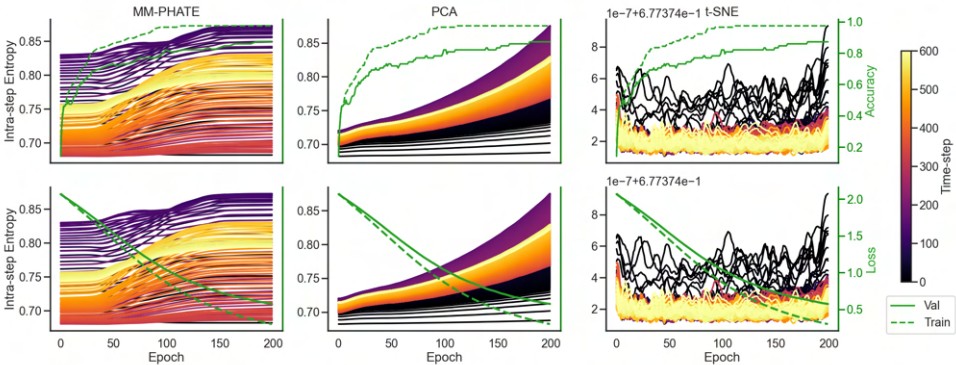

Figure A.38: Area2Bump GRU: Intra-step entropy of all hidden units in embedding space at each time-step in each epoch, compared to training and validation accuracy (top) and losses (bottom), comparing embeddings of MM-PHATE, PCA, and t-SNE.

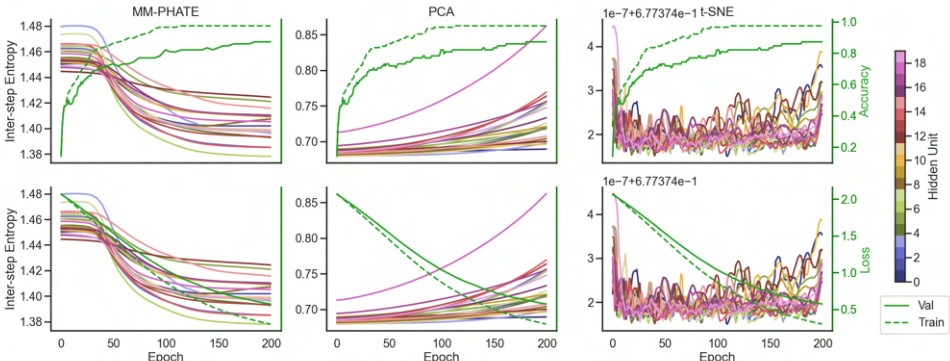

Figure A.39: Area2Bump GRU: Inter-step entropy of all hidden units in embedding space of the Area2Bump model at each time-step in each epoch, compared to training and accuracies (top) and losses (bottom). From left to right, the dimensionality reduction metrics used are MM-PHATE, PCA, and t-SNE.

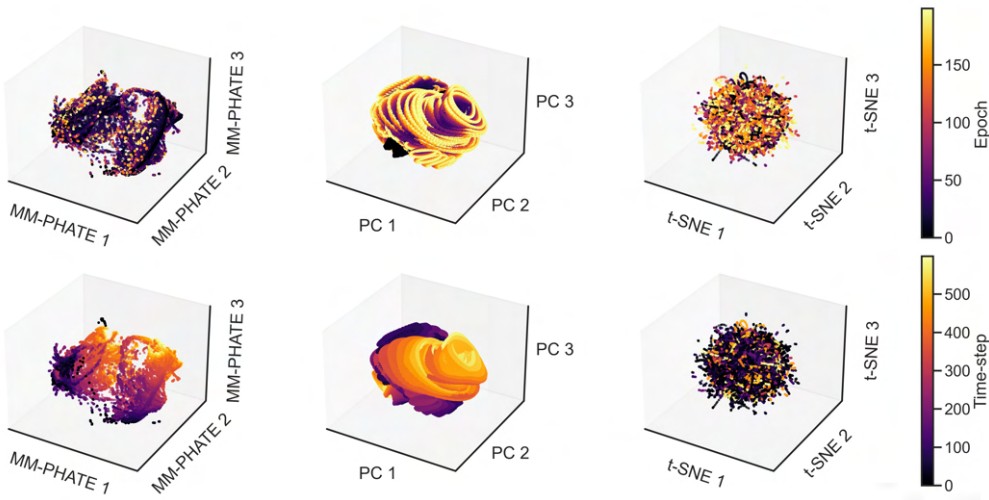

Figure A.40: Area2Bump Vanilla: Visualization of a 20-unit Vanilla RNN trained for 200 epochs. Each point represents a hidden unit at a specific time-step in a given epoch throughout the entire training process. The visualizations are generated using MM-PHATE, PCA, and t-SNE, from left to right, respectively. Points are colored based on epoch (top row) or time-step (bottom-row)

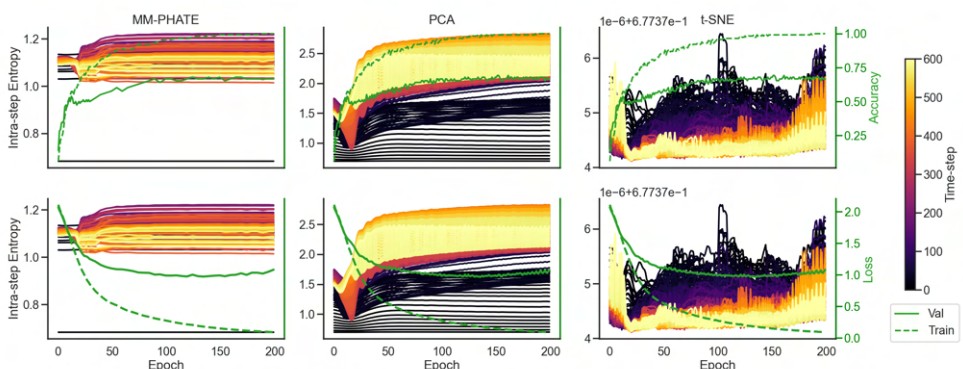

Figure A.41: Area2Bump Vanilla: Intra-step entropy of all hidden units in embedding space at each time-step in each epoch, compared to training and validation accuracy (top) and losses (bottom), comparing embeddings of MM-PHATE, PCA, and t-SNE.

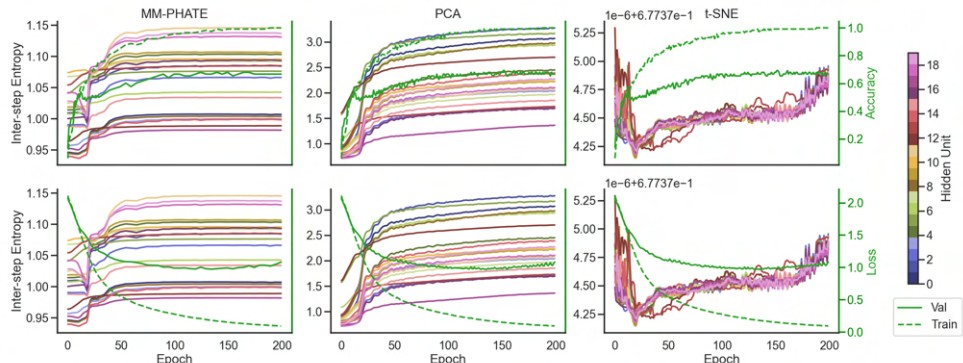

Figure A.42: Area2Bump Vanilla: Inter-step entropy of all hidden units in embedding space of the Area2Bump model at each time-step in each epoch, compared to training and accuracies (top) and losses (bottom). From left to right, the dimensionality reduction metrics used are MM-PHATE, PCA, and t-SNE.

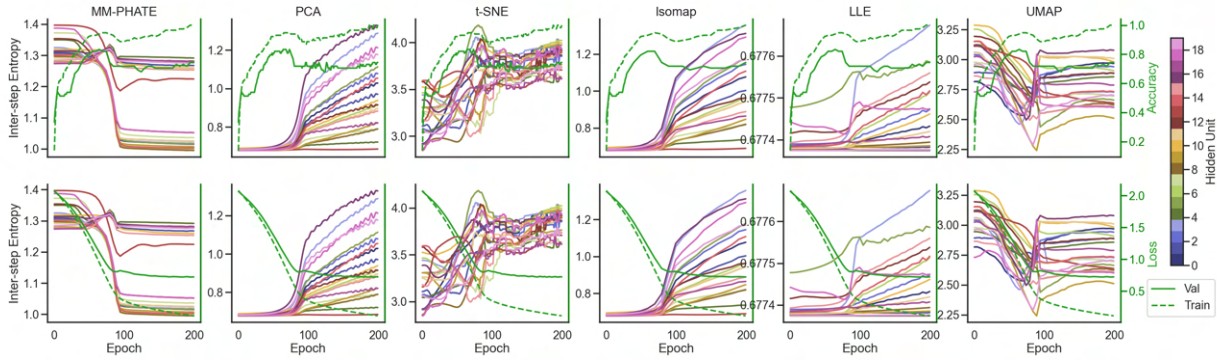

Figure A.43: Area2Bump: Inter-step entropy of all hidden units in embedding space of the Area2Bump model at each time-step in each epoch, compared to training and accuracies (top) and losses (bottom). From left to right, the dimensionality reduction metrics used are MM-PHATE, PCA, t-SNE, Isomap, LLE, and UMAP.

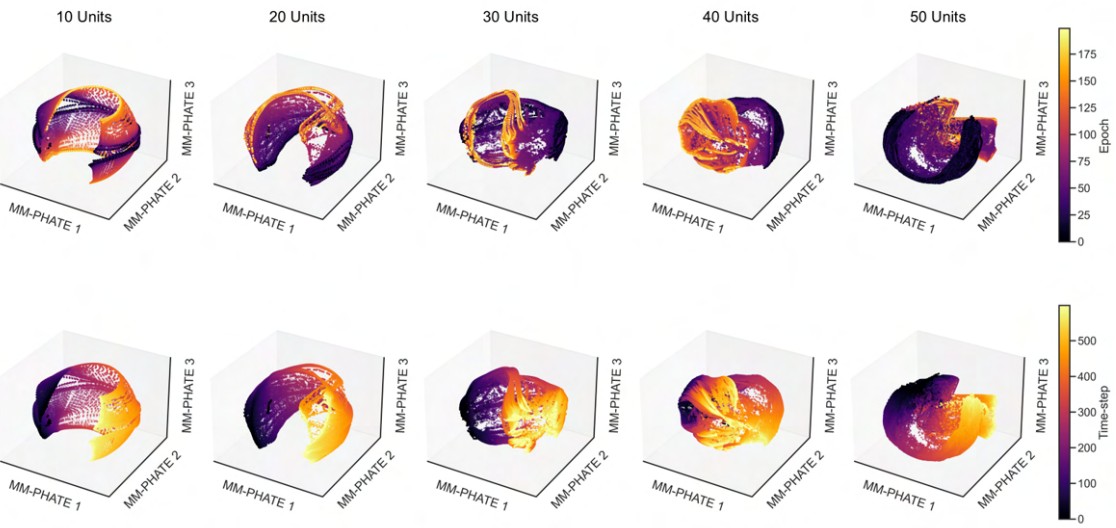

Figure A.44: Area2Bump LSTM: MM-PHATE visualization of networks of size 10 to 50 (left to right). Each point represents a hidden unit at a specific time-step in a given epoch. Points are colored based on epoch (top) or time-step (bottom).

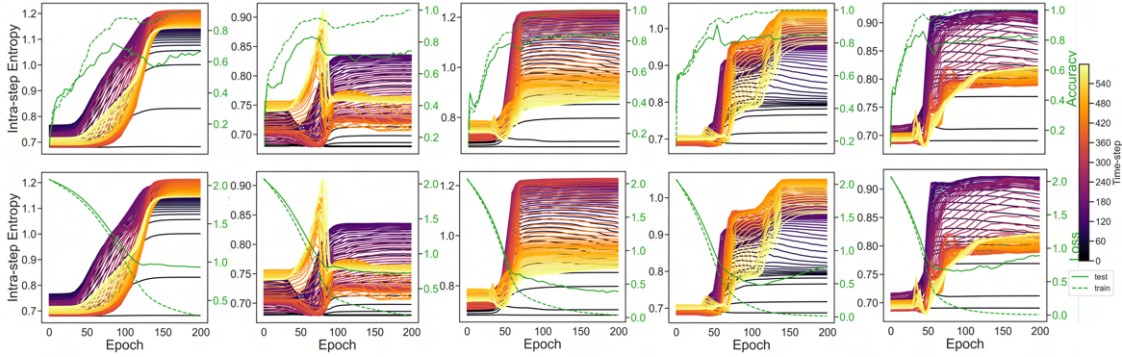

Figure A.45: Area2Bump LSTM: Intra-step entropy of all hidden units in MM-PHATE embedding space at each time-step in each epoch, compared to accuracies (top) and losses (bottom). Network sizes range from 10 to 50 (left to right).

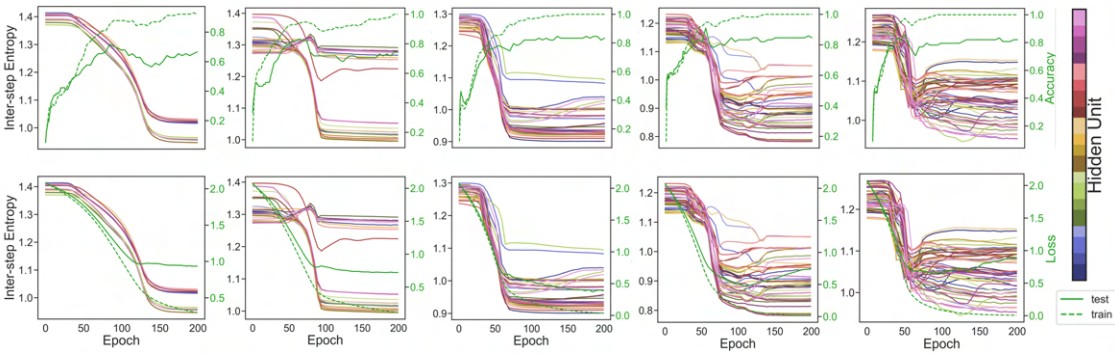

Figure A.46: Area2Bump LSTM: Inter-step entropy of all hidden units in MM-PHATE embedding space of the Area2Bump model at each time-step for each unit in each epoch, compared to accuracies (top) and losses (bottom). Network sizes range from 10 to 50 (left to right).

## A.14 Quantitative temporal-geometric metrics for HAR MM-PHATE embeddings

To complement the qualitative MM-PHATE visualizations in Section 5.3 and reduce reliance on visual inspection alone, we computed two epoch-wise quantitative metrics directly from the 3D MM-PHATE embedding of the HAR LSTM hidden states: (i) `signed_flow_alignment_centroid`, which measures the *direction* of within-sequence progression at each epoch, and (ii) `epoch_to_epoch_change_magnitude`, which measures the *amount of temporal-geometric reorganization* between consecutive sampled epochs.

**Epoch-wise temporal centroid trajectories (unit-balanced).** Let $z_{\tau,\omega,i} \in \mathbb{R}^3$ denote the 3D MM-PHATE point associated with sampled epoch $\tau$, intrinsic time-step $\omega \in \{1,\ldots,s\}$, and hidden unit $i \in \{1,\ldots,m\}$, whenever that tuple is present in the subsampled embedding. In our HAR pipeline, each observed $(\tau,\omega,i)$ tuple contributes at most one MM-PHATE point; subsampling removes some tuples but does not create duplicate tuples.

Because some $(\tau,\omega,i)$ tuples may be missing after subsampling, we define the set of hidden units present at epoch $\tau$ and time-step $\omega$ as

$$\mathcal{U}_{\tau,\omega} \subseteq \{1,\ldots,m\}. \tag{10}$$

We then form a *unit-balanced* centroid for each observed $(\tau,\omega)$ by averaging equally across the hidden units that are present:

$$\bar{z}_{\tau,\omega}^{(\mathrm{ub})} = \frac{1}{|\mathcal{U}_{\tau,\omega}|} \sum_{i \in \mathcal{U}_{\tau,\omega}} z_{\tau,\omega,i}. \tag{11}$$

For a fixed epoch $\tau$, the ordered sequence $\{\bar{z}_{\tau,\omega}^{(\mathrm{ub})}\}_\omega$ defines an epoch-specific trajectory in MM-PHATE space.

**Signed flow alignment centroid (direction of temporal progression).** To quantify whether within-sequence progression points in a consistent global direction, we define a data-driven reference axis $\hat{u} \in \mathbb{R}^3$ from the first principal component of epoch-wise net displacement vectors. For each epoch $\tau$, let $\omega_1 < \omega_2 < \cdots < \omega_{K_\tau}$ denote the observed time-steps (after subsampling), and define the net displacement

$$g_\tau = \bar{z}_{\tau,\omega_{K_\tau}}^{(\mathrm{ub})} - \bar{z}_{\tau,\omega_1}^{(\mathrm{ub})}. \tag{12}$$

The reference axis $\hat{u}$ is estimated from the first principal component of $\{g_\tau\}$ across epochs with sufficient observed time-steps.

Next, define consecutive centroid increments along the within-epoch trajectory:

$$\Delta\bar{z}_{\tau,k} = \bar{z}_{\tau,\omega_{k+1}}^{(\mathrm{ub})} - \bar{z}_{\tau,\omega_k}^{(\mathrm{ub})}, \qquad k = 1,\ldots,K_\tau - 1. \tag{13}$$

We define the `signed_flow_alignment_centroid` as

$$A_\tau = \frac{\sum_{k=1}^{K_\tau-1} \langle \Delta\bar{z}_{\tau,k}, \hat{u} \rangle}{\sum_{k=1}^{K_\tau-1} \|\Delta\bar{z}_{\tau,k}\|_2}, \tag{14}$$

which is bounded in $[-1,1]$ (up to numerical error). Positive values indicate that the within-sequence trajectory predominantly progresses along $\hat{u}$, whereas negative values indicate progression in the opposite direction. This yields an epoch-wise directional summary without pre-specifying training regimes.

**Epoch-to-epoch change magnitude (temporal-geometric reorganization).** A change in temporal dynamics may appear not only as a sign flip in Eq. equation 14, but also as changes in temporal ordering, trajectory extent, or spread. To capture this more broadly, we compute an epoch-wise temporal-geometric signature vector

$$q_\tau = \begin{bmatrix} A_\tau \\ \rho_\tau \\ \ell_\tau \\ \xi_\tau \end{bmatrix}, \tag{15}$$

where:

- $A_\tau$ is the signed flow alignment in Eq. equation 14,

- $\rho_\tau$ is the Spearman correlation between pairwise centroid distances $\left\|\bar{z}_{\tau,\omega}^{(\mathrm{ub})} - \bar{z}_{\tau,\nu}^{(\mathrm{ub})}\right\|_2$ and time-step lags $|\omega - \nu|$ (temporal ordering strength),

- $\ell_\tau = \sum_{k=1}^{K_\tau-1} \left\|\Delta\bar{z}_{\tau,k}\right\|_2$ is the within-epoch centroid trajectory path length (trajectory extent),

- $\xi_\tau$ is the within-epoch centroid spread across time-steps, computed as the mean coordinate-wise variance of $\bar{z}_{\tau,\omega}^{(\mathrm{ub})}$ over observed $\omega$.

Let $\tau_1 < \tau_2 < \cdots < \tau_R$ denote the sampled epochs used in the embedding/metric computation (which may be subsampled and therefore non-consecutive). After z-scoring each component of $\boldsymbol{q}_{\tau_r}$ across $r = 1, \ldots, R$, the `epoch_to_epoch_change_magnitude` is defined as

$$C_r = \left\|\tilde{\boldsymbol{q}}_{\tau_{r+1}} - \tilde{\boldsymbol{q}}_{\tau_r}\right\|_2, \qquad r = 1, \ldots, R-1, \tag{16}$$

and is plotted at the midpoint $(\tau_r + \tau_{r+1})/2$. This formulation avoids imposing continuity assumptions when epoch labels are subsampled.

**Quantitative support for the HAR transition.** These metrics quantitatively support the qualitative MM-PHATE observations reported in Section 5.3. In particular, `signed_flow_alignment_centroid` ($A_\tau$) increases from negative to positive around epoch $\sim 7$ (Fig. A.47, zoomed panel), indicating a sign change in the dominant direction of within-sequence progression in the MM-PHATE embedding. This sign change coincides with a pronounced spike in `epoch_to_epoch_change_magnitude` ($C_r$) and a sharp increase in training/validation accuracy, consistent with an early training-phase reorganization rather than a purely gradual drift. The corresponding shift in temporal dynamic direction is also visible in the original 3D MM-PHATE embedding (Fig. A.49), where the direction of time-step progression changes across early epochs.

**Alignment with training dynamics over the full trajectory.** Across the full training run, large fluctuations in $C_r$ and changes in $A_\tau$ align with changes in model performance (accuracy and loss; Figs. A.47–A.48). This consistent alignment supports the interpretation that MM-PHATE is capturing meaningful temporal-geometric structure in the evolving hidden-state dynamics, rather than arbitrary low-dimensional distortion.

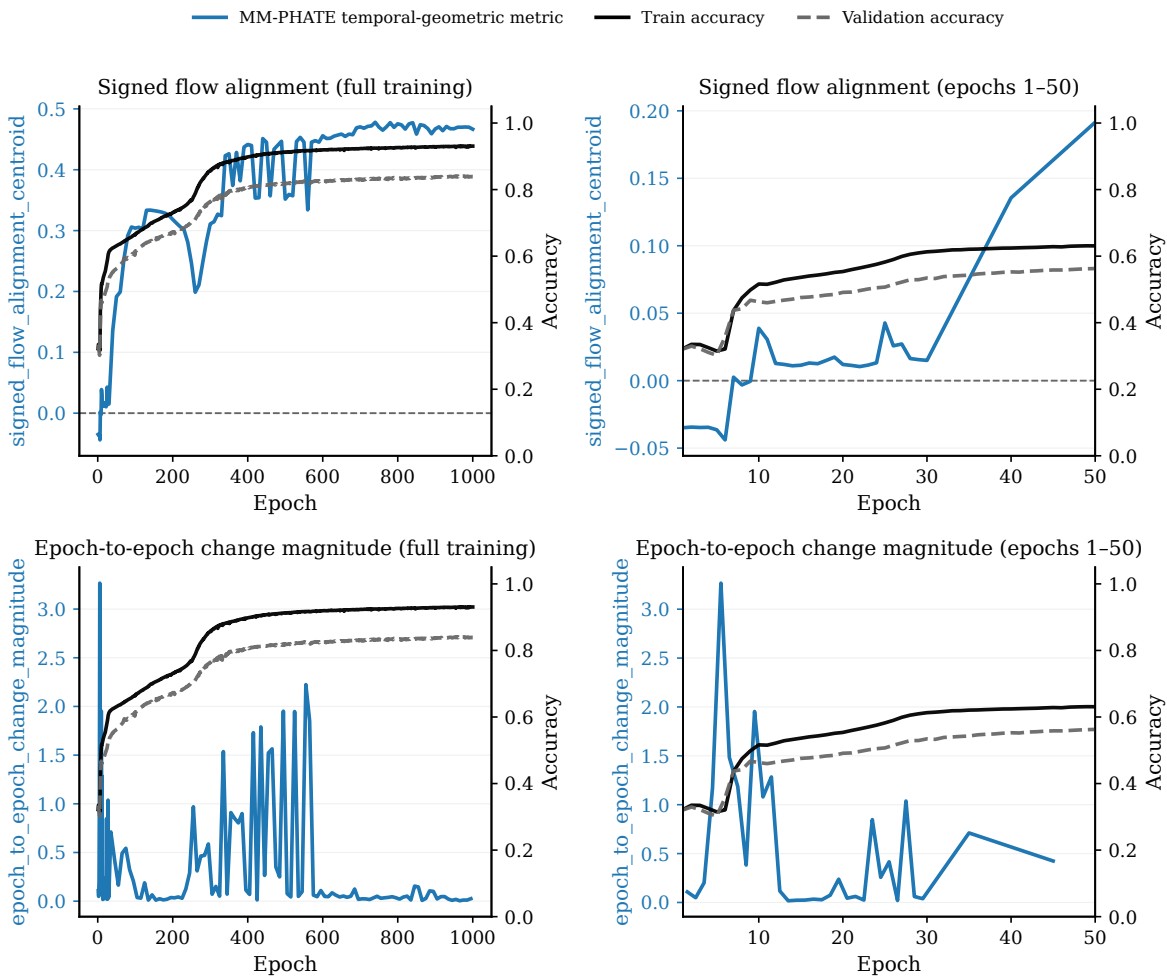

Figure A.47: Quantitative temporal-geometric metrics on HAR MM-PHATE embeddings, compared with training and validation accuracy. **Top:** `signed_flow_alignment_centroid` ($A_\tau$) over training (left: full training; right: zoom to early epochs). **Bottom:** `epoch_to_epoch_change_magnitude` ($C_r$) (left: full training; right: zoom to early epochs). The early sign change in $A_\tau$ (around epoch $\sim 7$) coincides with a spike in $C_r$ and a sharp increase in accuracy.

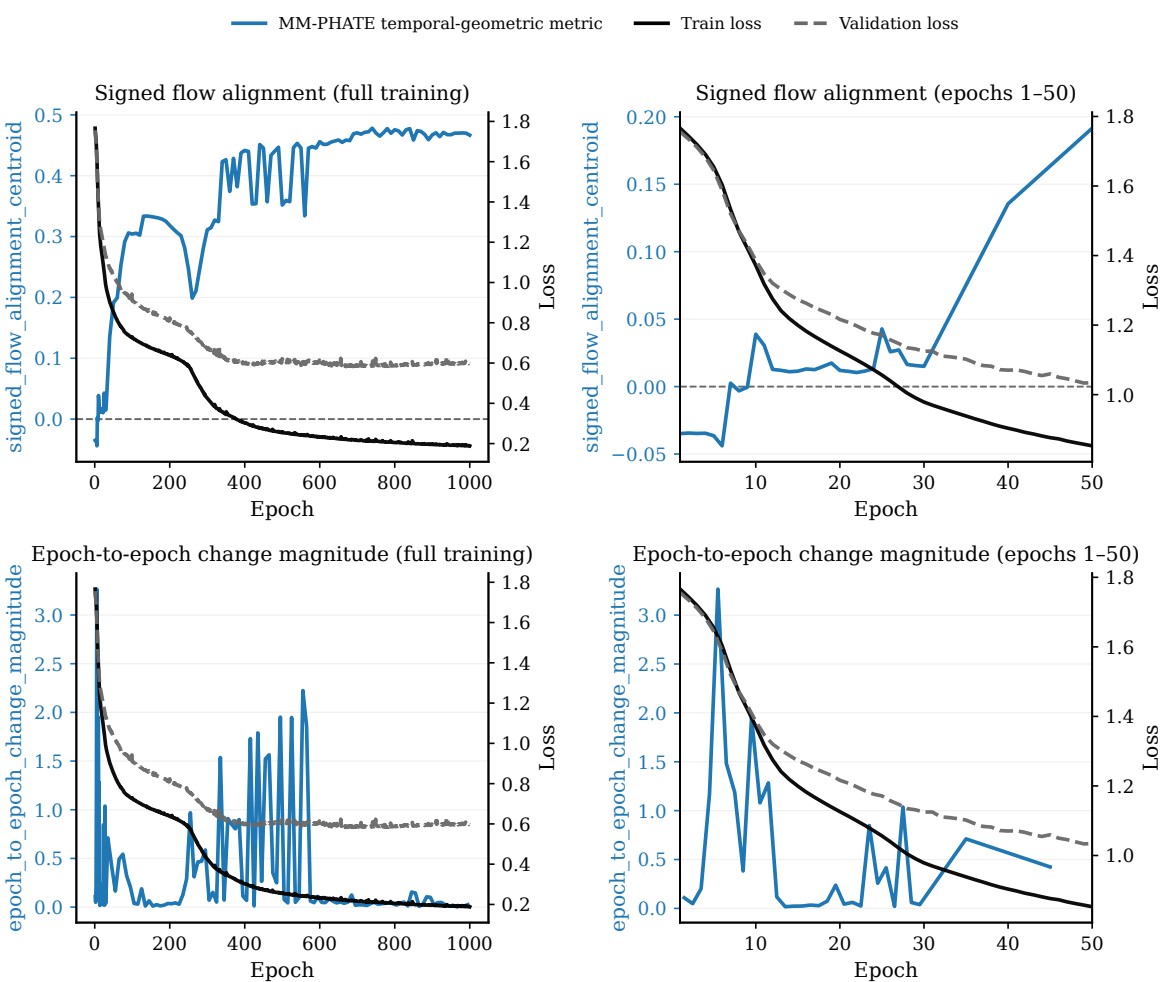

Figure A.48: Same quantitative temporal-geometric metrics as Fig. A.47, but compared with training and validation loss. Large fluctuations in the MM-PHATE temporal-geometric metrics align with changes in loss over training, supporting that the embedding captures training-relevant temporal reorganization in the hidden-state dynamics.

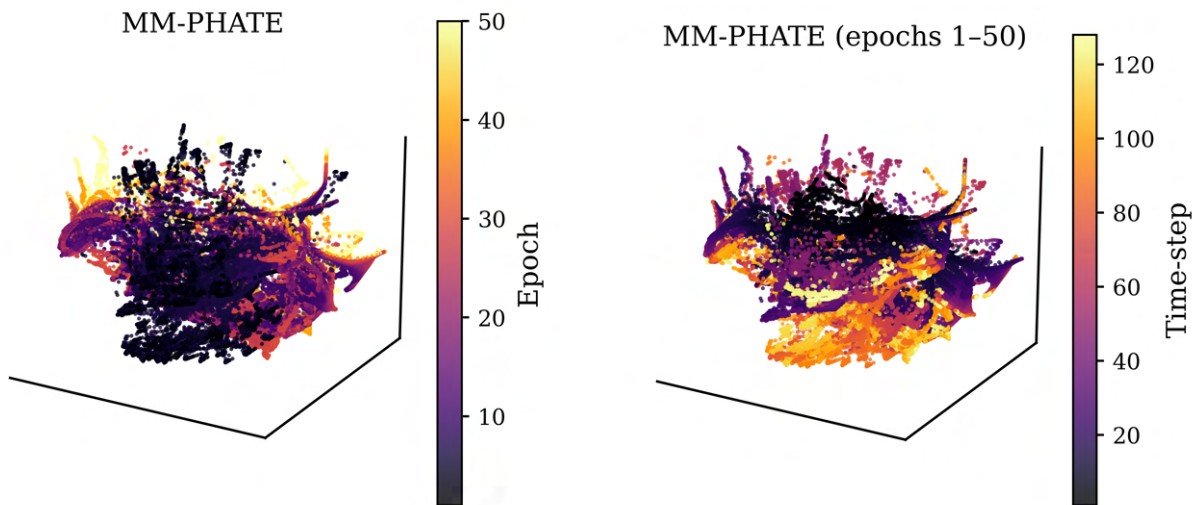

Figure A.49:  Early-epoch HAR MM-PHATE 3D embeddings (first 50 epochs), shown from a fixed viewpoint. **Left:** colored by epoch $\tau$. **Right:** colored by intrinsic time-step $\omega$. The time-step-colored view makes the early shift in temporal progression direction visible, consistent with the sign change in `signed_flow_alignment_centroid` ($A_\tau$).

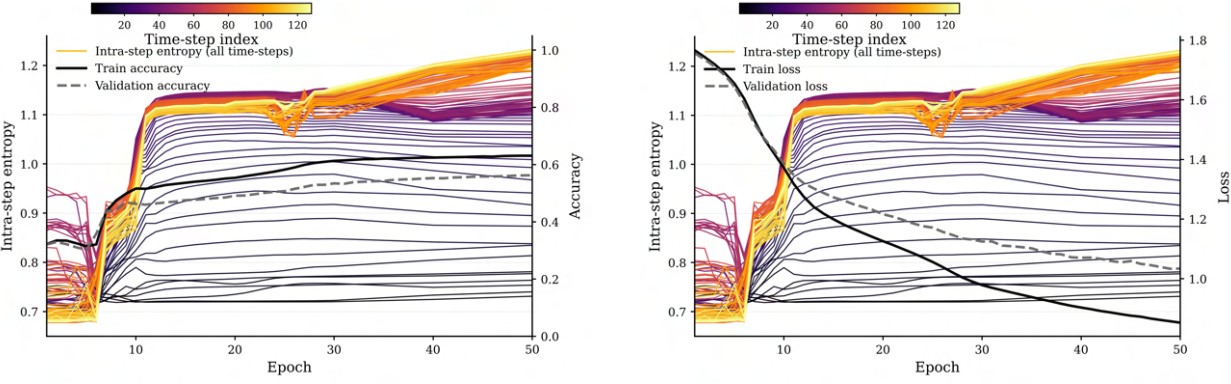

Figure A.50:  Early-epoch HAR MM-PHATE intra-step entropy (first 50 epochs). **Left:** compared to accuracy. **Right:** compared to loss. The intra-step entropy changes drastically around epoch 7, aligning to the sign change in `signed_flow_alignment_centroid` ($A_\tau$) and performance increase.

## B  Computing Infrastructure

All but the t-SNE Area2Bump computation was carried out on a 14-core laptop running Windows 11 Home with a NVIDIA GeForce RTX 3070 Ti Laptop graphics card and 40GB of RAM. The t-SNE Area2Bump visualization was conducted on a single 95-core internal cluster running Ubuntu 18.04.6 LTS with 10 Quadro RTX 5000 graphics cards and 755GB of RAM.

## C  Mathematical Notations

| Notation | Definition |
|---|---|
| $\boldsymbol{x}$ | Point in high dimensional space |
| $\boldsymbol{E}$ | Euclidean distance matrix between all data points $\boldsymbol{x}$ |
| $\boldsymbol{K}$ | Affinity kernel matrix |
| $\epsilon_k(x_i)$ | $k$-nearest-neighbor distance of $x_i$ |
| $\alpha$ | Parameter controlling the decay rate |
| $\boldsymbol{P}$ | Diffusion operator |
| $\boldsymbol{D}$ | Diagonal matrix of row sums of $\boldsymbol{K}$ |
| $\boldsymbol{P}^t$ | Transition probabilities of a diffusion process over $t$ steps |
| $n$ | Total number of epochs the network is trained for |
| $\boldsymbol{F}$ | Feed-forward neural network |
| $m$ | Total number of hidden units in the network |
| $\boldsymbol{X}$ | Training data, subset of $\boldsymbol{\Pi}$ |
| $\boldsymbol{\Pi}$ | Larger dataset |
| $T$ | Activation tensor |
| $\boldsymbol{Y}$ | Input data, subset of $\boldsymbol{X}$ with equal number of samples per class |
| $p$ | Number of samples in $\boldsymbol{Y}$ |
| $\boldsymbol{K}^{(\tau)}_{\text{intraslice}}(i,j)$ | Intraslice affinities between pairs of hidden units within an epoch $\tau$ |
| $\boldsymbol{K}^{(i)}_{\text{interslice}}(\tau,\upsilon)$ | Interslice affinities between a hidden unit $i$ and itself at different epochs |
| $\sigma_{(\tau,i)}$ | Intraslice bandwidth for unit $i$ in epoch $\tau$ |
| $\epsilon$ | Fixed interslice bandwidth |
| $\tau$ | Index for given epoch |
| $i,j$ | Index for given hidden unit |
| $h_t$ | RNN hidden state at time-step $t$ |
| $W$ | RNN weights |
| $b$ | RNN biases |
| $y_t$ | RNN output at time-step $t$ |
| $f$ | RNN activation function |
| $\boldsymbol{R}$ | Recurrent neural network |
| $\omega$ | Index for given time-step |
| $s$ | Total number of time-steps in the RNN |
| $\boldsymbol{K}^{(\tau,\omega)}_{\text{intra-step}}(i,j)$ | Intra-step affinities between hidden units $i$ and $j$ at time-step $\omega$ in epoch $\tau$ |
| $\boldsymbol{K}^{(i)}_{\text{inter-step}}((\tau,\omega),(\eta,\nu))$ | Inter-step affinities between a hidden unit $i$ and itself at different time-steps and epochs |
| $\sigma_{(\tau,\omega,i)}$ | Intra-step bandwidth for unit $i$ at time-step $\omega$ and epoch $\tau$ |
| $k$ | Number of nearest neighbors |
| $\boldsymbol{L}$ | Labels of $\boldsymbol{X}$ |

Table 4: Notations

