# OpenReview forum: "Multiway Multislice PHATE: Visualizing Hidden Dynamics of RNNs through Training"
_TMLR — Accepted by TMLR_

### Review · Reviewer_J7ag · 2025-08-15

**Summary Of Contributions:**

The paper describes a new method for visualizing the hidden dynamics of RNNs during training.
The method is shown to be able to identify changes in network performance, even on validation data.
These insights are demonstrated in two RNN training examples.
The proposed method generally provides more insight into the network dynamics during learning than other visualization methods.
The evaluation of the method is however limited in scope, often relying on a single trained RNN for a parameter/architecture set.

**Audience:**

Yes

**Audience Explanation:**

This problem is very relevant for the ML field to understand and monitor RNN training.

**Claims And Evidence:**

No

**Claims Explanation:**

## Claims with evidence
1. "Our analysis of the MM-PHATE embedding revealed a general increase in entropy (Fig. 3) around epoch 100, where the network begins to overfit."
This is a very interesting and useful property of MM-PHATE with clear evidence for the trained LSTM networks and a good demonstration of how other visualization methods fail to capture this.
However, it is unclear that this is something specific for this example or whether this is a general feature. Including another example of overfitting would make this claim stronger.

1. The method gives consistent results even for different architectures and network sizes, especially for the time-step dimension (e.g. Figure A.15).

1. The clustering analysis provides some interesting insight into the internal workings of the RNN that was investigated.

1. The method generally outperforms other visualization methods considered in the paper, namely PCA, t-SNE, UMAP, ISOMAP and LLE.
For example, in Figure 2, the structure identified by MM-PHATE is much more intuitive and interpretable than the embeddings of the other methods.


## Claims without clear evidence
1. It is claimed that MM-PHATE is a novel method for visualizing the evolution of RNNs’ hidden states.
I do believe that the method is promising, but it is unclear what structures are captured/ignored by the method and how reliable the method is in different settings, see the points below.
1. "fluctuations in validation accuracy before epoch 30 were not reflected in the training loss curve but were detected by MM-PHATE" but it is unclear whether this is concluded from Figure 2 or Figure 3. Having looked at both figures for a while, I am still just guessing how this might be read out from the top left subfigure from Fig 2. Perhaps having a figure in the supplementary with only the first 30 epochs could clarify how the fluctuations in validation accuracy were detected by MM-PHATE.
Looking at Figure 3, I would have concluded that the representation in the first 30 epochs do not change, even though the loss/accuracy are changing.

1. It remains unclear what exactly MM-PHATE can (and cannot) capture.

For example, why does an increase in the training accuracy correspond to the change in intra-step entropy for single units? Does this suggest that these units are responsible for overfitting?

What kind of insight does Figure 5a provide? The figure does not seem to indicate any particular interpretable structure.

1. To demonstrate the effectiveness and reliability of the method the authors should demonstrate (e.g. through correlation) that the claimed evidence of validation loss change is indeed reflected in MM-PHATE.


1.
"Certain units exhibited significantly higher inter-step entropy, indicating greater sensitivity to input changes over time and a potential role in capturing temporal dependencies".
This is indeed an interesting insight.


1. Why was Dynamic Time Warping used? (Citation needed!)
How robust is this clustering algorithm?
These experiments should be repeated in different settings or instantiations to provide certainty about the reliability.

1. Looking at LSTM and vanilla RNNs seems to affect the

Figure A.12 seems to suggest that PCA is a better method to reflect changes in training and validation accuracy and loss than MM-PHATE as its entropy changes over time (while it seems to be constant for MM-PHATE).

1. The number RNN units seems to also affect the insights provided by the method.
Figure A.15 is less easily interpretable compared to Figure 2.
For example, why do later epochs evolve in two different direction along MM-PHATE 3?
How do the authors explain what the method is capturing?

1. Overall, having different instantiations for each RNN parameter setting could clarify how robust the method is.



1. Other methods that have been applied to describing RNN training:
Ostrow, M., Eisen, A., Kozachkov, L. and Fiete, I., 2023. Beyond geometry: Comparing the temporal structure of computation in neural circuits with dynamical similarity analysis. Advances in Neural Information Processing Systems, 36, pp.33824-33837.

Guilhot, Q., Wójcik, M., Achterberg, J. and Costa, R.P., 2024. Dynamical similarity analysis can identify compositional dynamics developing in RNNs. arXiv preprint arXiv:2410.24070.

Redman, W., Bello-Rivas, J., Fonoberova, M., Mohr, R., Kevrekidis, Y. and Mezic, I., 2024. Identifying equivalent training dynamics. Advances in Neural Information Processing Systems, 37, pp.23603-23629.

Ribeiro, A.H., Tiels, K., Aguirre, L.A. and Schön, T., 2020, June. Beyond exploding and vanishing gradients: analysing RNN training using attractors and smoothness. In International conference on artificial intelligence and statistics (pp. 2370-2380). PMLR.

**Requested Changes:**

## Main requested changes
1. The claims about the working of the method should be tested in well-controlled experiments.
The claims would be better supported by well-designed numerical experiments based on a fully-understood toy model. For example, to demonstrate the effectiveness of the method, the authors could investigate a simple, low-dimensional system that goes through a bifurcation (e.g. Hopf) that would function as a simplified model of learning in an RNN.
1. The figure labels should better help the reader to understand the content of the plots and the message overall.
1. The method should be demonstrated on multiple instances of each RNN parameter combination (architecture, size, etc).
1. Outside of the captions, more interpretation should be given to the different figures and how the authors arrive at their conclusions/claims from them.




## Smaller remarks
1. The top and bottom subfigures in Fig. 5b and 5c should be merged, the bottom ones provide no additional information. This is probably also true for Figure 3.
1. Visualizing high-dimensional data is hard, even in 3D. It might clarify some uncertainties about the full cloud points to use transparency for the figures.

---

> ### Author Response · Authors · 2025-12-12
> **Response sec.1**
>
> We sincerely thank the reviewer for the thoughtful and detailed feedback. We appreciate the reviewer’s positive assessment of MM-PHATE’s potential to reveal meaningful training dynamics and the constructive suggestions for strengthening the empirical and conceptual foundations. Below, we address each point raised and outline the revisions we incorporated.
>
> 1. Controlled experiment using a well-understood dynamical system: “Claims should be tested in a well-controlled toy model… e.g., Hopf bifurcation.”
>
> Response: We agree that validating MM-PHATE on a fully understood system helps ground the method. We have conducted an analysis using a Hopf bifurcation system, a canonical dynamical system with a well-known transition from a stable fixed point to a limit cycle. We have included a new Section 5.1 and Appendix A.5, which show that MM-PHATE accurately captures the onset of oscillatory dynamics and the qualitative geometric transition at the bifurcation point. This controlled experiment directly demonstrates that MM-PHATE detects known dynamical transitions and, hopefully, addresses some concerns about the structure it captures, as noted in the following response.
>
> 2. Clarification of what MM-PHATE captures and why: “It remains unclear what exactly MM-PHATE can (and cannot) capture… Why do certain changes correspond to entropy changes? Why two directions in some figures?”
>
> Response: We appreciate this request for conceptual clarity. In the revised manuscript, we have extended the discussion regarding what MM-PHATE captures in the Hopf Bifurcation analysis (Section 5.1) to emphasize the following points:
>
> >MM-PHATE captures the local geometric structure of hidden states across units × time steps × epochs.
>
> >MM-PHATE emphasizes temporal continuity and inter-step correlations while de-emphasizing global distances.
>
> >An increase in entropy reflects increased dispersion and variability in hidden-state trajectories, which can occur even when accuracy remains steady.
>
> >Multi-branch or “two-direction” evolutions (e.g., in larger LSTMs) arise when later-epoch trajectories diverge due to heterogeneous unit roles or emergent specialization (e.g., in the Area2Bump example) or due to changes in latent representation (e.g., in the HAR example).
>
> These additions will clarify the two directions of embedding change observed in Figure 7 (originally Fig.5), as discussed in Section 5.3.
>
> 3. Interpretation of Figures (early epochs, HAR dataset, complex patterns): “Fluctuations before epoch 30 are hard to see.” ;  “Unclear what insight Figure 5a provides.” ; “Later epochs evolve in two directions—needs explanation.”
>
> Response: We appreciate the reviewer’s request for clearer figure interpretation. We have now extended the main text and captions for Figures 4 and 7 to explicitly describe:
>
> >why are early-epoch structural changes in Area2Bump hard to see, even though the accuracy is changing (Section 5.2.2, Linear probes, ablations, and mutual information): “The weak change in probe accuracy before epoch 30 suggests that the hidden states have not yet undergone a substantial reorganization into task-informative subspaces. “
>
> >what structures are present in the HAR embeddings (e.g., separation of cyclic gait phases)： Geometry reveals a training-phase reorganization in Section 5.3.
>
> 4. Robustness across architectures, network sizes, and random initializations: “The method should be demonstrated across multiple instantiations…  LSTM vs GRU vs Vanilla RNN differences need attention…  Larger networks show harder-to-interpret structures.”
>
> Response: We agree with the reviewer that method robustness is important. We note that analyses covering GRU, Vanilla RNN, and multiple hidden unit sizes (10–50) are already included in the Appendix (Sections A.6–A.8), but were not sufficiently emphasized in the main text. In the revision, we:
>
> >Explicitly referenced these appendix sections in Results,
>
> >Added brief summary sentences describing the consistent patterns across architectures and sizes,
>
> These additions substantially strengthen the reliability claims.

---

> ### Author Response · Authors · 2025-12-12
> **Response sec.2**
>
> 5. Clustering: DTW justification and robustness: “Why was Dynamic Time Warping used? How robust is the clustering?”
>
> Response: We thank the reviewer for raising this point. We used time-series k-means with Dynamic Time Warping (DTW) because the inter-step entropy trajectories can exhibit small shifts in the epoch at which entropy increases. A Euclidean distance on raw trajectories would treat two units with the same qualitative pattern but slightly shifted in time as dissimilar, whereas DTW explicitly aligns sequences under local time-warping and clusters units based on their shared temporal evolution. We now clarify this choice in the text and cite standard DTW references and implementations [1].
>
> Regarding robustness, our goal is not to claim a unique or canonical partition of units, but to provide an exploratory summary of the heterogeneous entropy trajectories already visible in Fig. 6. In the revision, we explicitly frame the DTW k-means analysis as descriptive, and we avoid making strong claims that depend on the exact number or composition of clusters (Section 5.2.3). The main empirical support for our contributions (Hopf benchmark, neighborhood preservation, probes/ablations/MI) does not rely on this clustering step; rather, the DTW groups simply illustrate that units with persistently higher inter-step entropy form a coherent subgroup whose increased temporal sensitivity is also apparent in the raw entropy traces. A more exhaustive robustness study across architectures, datasets, and clustering hyperparameters is an important direction for future work.
>
>
> [1] Bringmann, K., Fischer, N., Hoog, I. van der, Kipouridis, E., Kociumaka, T., & Rotenberg, E. (2023). Dynamic Dynamic Time Warping (No. arXiv:2310.18128). arXiv. https://doi.org/10.48550/arXiv.2310.18128
>
> 6. Explanation of inconsistent entropy behavior (e.g., PCA vs MM-PHATE): “Figure A.12 suggests PCA entropy increases more than MM-PHATE; what does this mean?”
>
> Response: We have clarified in the revised main text that entropy in this context reflects dispersion of low-dimensional embeddings, not model performance. PCA is sensitive to global variance and may inflate entropy when variance is dominated by noise or large-amplitude directions; in contrast, MM-PHATE uses diffusion geometry to emphasize temporally smooth structure. We have added explanatory text in sections 5.1 and 5.2.
>
> 7. Figure formatting and presentation: “Use transparency… merge panels… improve labels.”
>
> Response: We thank the reviewer for these helpful formatting suggestions. We have integrated these improvements in the revised and final camera-ready versions, including clearer labels, merged redundant subpanels, and improved visualization of dense scatterplots.
>
> Regarding the suggestion that “The top and bottom subfigures in Fig. 5b and 5c should be merged, the bottom ones provide no additional information. This is probably also true for Figure 3,” we respectfully disagree. Accuracy and loss capture different aspects of model behavior and are not redundant. In particular, phase structure in training can be visible in one metric but not the other. For example, in Fig. 3 the transition before epoch 100 is clearly expressed in accuracy, whereas the loss curve over the same interval is comparatively smooth and does not exhibit a similarly sharp change. We therefore retain both accuracy and loss panels so that the reader can relate representational changes (e.g., in entropy and MM-PHATE geometry) to complementary performance metrics, rather than relying on a single aggregate measure.
>
>
> NOTE: For ease of review, in the revised manuscript we use color coding: purple text in the main body denotes entirely new analyses and results, teal text indicates substantially rephrased or clarified material, and appendix sections with yellow-highlighted titles correspond to newly added content.

---

### Review · Reviewer_8qeY · 2025-10-15

**Summary Of Contributions:**

- RNN are widely used yet still remain a largely a black box. To alleviate this issue this paper proposes Multiway Multislice PHATE (MM-PHATE), an extension of the M-PHATE approach. The authors extension considers affinities between units at a given time step as well as between time-steps.
- The paper demonstrates that the new method preserves hidden unity structure
- The paper presents interesting new evidence in support ot the information bottleneck theory for neural networks.
- The related work section summarizes the literature well.
- The authors observe improved temporal consistency when using the new method on the Human Activity Recognition (HAR) as well as the Area2Bump data-sets. The area2bump datasets consists of neural spiking activity data from macaques.

**Audience:**

Yes

**Audience Explanation:**

- Neat illustrations, the paper is relevant and well written.
- I expect the paper's experiments on neuroscience-data will fall on fertile ground within the machine learning community.
- Experiments on more widely known data-sets would broaden the paper's reach.

**Claims And Evidence:**

Yes

**Claims Explanation:**

- The presented evidence supports the paper. I saw that the supplementary code sets the pseudorandom seeds values. If the authors add the exact version of the machine learning library they used, this work can be made reproducible.
- Unfortunately the data sets used in this paper are relatively obscure.

**Requested Changes:**

- Add a citation for the first mention of the information bottleneck theory on page 2.
- The linear transformers are RNNs paper ( https://proceedings.mlr.press/v119/katharopoulos20a/katharopoulos20a.pdf ), could be cited to strengthen the last point in the conclusion.

- It would have been interesting see the method applied on more widely know data sets like WMT'14 from ( https://arxiv.org/pdf/1409.0473 ). Although this can still happen in future work. A very quick remedy would be to include adding and memory problems from https://proceedings.mlr.press/v48/arjovsky16.pdf .

- In its current form, the supplementary code is not presentable:
    - The documentation of the supplementary code is extremely sparse, I suggest the authors improve the documentation.
    - Furthermore the project's `requirements.txt` file does not specify the exact version of the machine learning framework used, which is important since changes i.e. to the pseudorandom number generator impact reproducibility.
    - The code does not apply systematic containerized testing, see
        - https://docs.pytest.org/en/stable/
    - as well as
        - https://nox.thea.codes/en/stable/index.html
        - https://tox.wiki/en/4.31.0/
    - for more information.
    - The project code is not properly packaged, this increases the risk for code duplication in the community.
        - Please take a look at https://packaging.python.org/en/latest/ for the final version of this code.
    - This can still be fixed, I would consider this a minor yet very important change since replicability and reproducability are a key components of TMLR's scientific culture.

---

> ### Author Response · Authors · 2025-12-12
>
> We sincerely thank the reviewer for their evaluation of the manuscript and for the thoughtful suggestions regarding reproducibility, broader context, and additional citations. We are glad that the reviewer found the paper well written, relevant to the ML community, and that the neuroscience results would “fall on fertile ground.” Below, we address each comment point-by-point.
>
> 1. Citation for Information Bottleneck Theory: “Add a citation for the first mention of the information bottleneck theory on page 2.”
>
> Response: We thank the reviewer for pointing this out. We have added a citation to the classical formulation of the information bottleneck (Tishby & Zaslavsky, 2015; Fischer, 2020) at its first mention.
>
> 2. Linear Transformers as RNNs: “The linear transformers paper could be cited to strengthen the last point.”
>
> Response:  We appreciate the suggestion and have cited Katharopoulos et al. (2020) in the conclusion to further emphasize the broader relevance of visualizing sequential computation architectures beyond classical RNNs.
>
> 3. Applying MM-PHATE to more widely known datasets (WMT’14, addition/memory tasks): “It would be interesting to see the method applied to WMT’14 or to adding/memory tasks.”
>
> Response: We appreciate the reviewer’s perspective on broader benchmarking. Due to computational constraints and the focus on analyzing hidden dynamics, large-scale machine translation experiments such as WMT’14 fall outside the scope of the current submission. However, we agree that controlled tasks are valuable. To address this direction, we have added (also per Reviewer J7ag’s request) a controlled dynamical system analysis based on the Hopf bifurcation, which provides a clear, interpretable test case with known ground-truth transitions. This addition strengthens the methodological foundation. We view the application to addition/memory tasks as an excellent direction for future work.
>
> 4. Reproducibility, documentation, and packaging: “Supplementary code lacks documentation, exact versions, proper packaging, and containerized testing.”
>
> Response:  We thank the reviewer for highlighting these important aspects of scientific reproducibility. In the camera-ready version of the paper and code release, we will:
>
> >Expand the documentation and in-line comments in the codebase to clarify the main pipelines and configuration options.
>
> >Provide a requirements.txt (and an optional environment file) specifying exact versions of all dependencies, including the deep learning framework, together with explicit seed-setting to support pseudorandom reproducibility.
>
> >Reorganize the repository into a minimal package-style structure to reduce duplication and make the main scripts easier to discover and reuse.
>
> >Add simple test scripts and clear instructions for running the key reproduction pipelines end-to-end (e.g., reproducing the main figures and metrics).
>
> 5. Obscurity of datasets: “The datasets are relatively obscure.”
>
> Response: To address clarity, we have revised the dataset descriptions, add proper citations, and expand the motivation for using these datasets (Area2Bump as a neuroscience-motivated benchmark of population dynamics; HAR as a real-world sequential classification dataset). We note that Area2Bump is a part of the well-known “Neural Latents Benchmark” Challenge (2021) [1] in the neuroscience community. The HAR dataset was originally introduced in [2] which has been cited 3k times, and is also part of the UCI ML repository (where it has been viewed ~196k times). We have also added the new Hopf bifurcation experiment (Section 5.1), which provides a simple, widely interpretable benchmark that complements the more specialized datasets.
>
> [1] Pei, F., J. Ye, D. Zoltowski, A. Wu, R. H. Chowdhury, H. Sohn, J. E. O’Doherty et al. "Neural Latents Benchmark’21: Evaluating latent variable models of neural population activity." Advances in Neural Information Processing Systems (NeurIPS), Track on Datasets and Benchmarks 34 (2021)
>
> [2] Anguita, D., Ghio, A., Oneto, L., Parra, X., & Reyes-Ortiz, J. L. (2013, April). A public domain dataset for human activity recognition using smartphones. In Esann (Vol. 3, No. 1, pp. 3-4).
>
>
> NOTE: For ease of review, in the revised manuscript we use color coding: purple text in the main body denotes entirely new analyses and results, teal text indicates substantially rephrased or clarified material, and appendix sections with yellow-highlighted titles correspond to newly added content.

---

> ### Comment · Reviewer_8qeY · 2026-01-19
> **Thank you for your repliy**
>
> I am sorry for the delay, since the reply was late open-review places this paper in the finished pile. I have no further questions and will update my recommendation accordingly.

---

### Review · Reviewer_f4wa · 2025-11-16

**Summary Of Contributions:**

The paper presents MM-PHATE, an extension of M-PHATE for visualizing training dynamics of RNNs by using structured kernel also across time steps of the RNN. The paper evaluates the proposed technique on two datasets, namely Area2Bump and the Human Activity Recognition, and empirically shows using visual plots that MM-PHATE preserves community structure among units and identifies information processing and compression phases during training


Strengths:
- New approach for visualizing the dynamics of RNNs throughout their training
- Results on two datasets, Area2Bump and the Human Activity Recognition, visually illustrates the benefits of the proposed approach

Weaknesses:
- Technically, this seems like a straightforward adaptation of M-PHATE, where the affinities are computed in a similar manner, just expanding the scope from across epochs and hidden unit to across epoch, hidden unit, and time step.
- There is no sufficient quantitative evaluation and comparison between approaches, making it hard to judge the claims in the paper, in particular in comparison to the baselines, instead relying on interpretation of the plots.
	- In particular, contribution #2 ("MM-PHATE preserves the community structure of hidden units by tracking their learning trajectories and capturing correlations among their activations throughout training”) is therefore not sufficiently substantiated in comparison to the baselines.
	- I note that the precursor work (M-PHATE) has provided some quantitative analysis and comparison of the preservation of neighborhood structure.
- The claim in contribution #3 ("MM-PHATE reveals phases of information processing and compression during training, consistent with the information bottleneck theory”) is not sufficiently justified:
	- This claim is not clearly formalized mathematically so it is hard to verify.
	- The experiments designed "to quantitatively confirm that the information retained by the network is reflected in the MM-PHATE embedding” are again based on looking at plots and not based on a clear quantitative comparison to the baselines.
	- The random label experiments (in the appendix) are only conducted on MM-PHATE and not on the baselines and do not support the comparison between the baselines.
- The lack of quantitative comparison, and the reliance on visual plots also restricts the experimental analysis to two datasets. It would also be nice to see analysis on different types of sequential data like, e.g., natural language data.
- Comparison with the precursor work (M-PHATE) as a baseline is very limited and only seems to include Fig A.7. It is not clear why not all comparisons with the baselines include M-PHATE

**Audience:**

Yes

**Audience Explanation:**

As noted under weaknesses, this proposed approach seems like a relatively straightforward adaptation of M-PHATE and the potential uses of this tool are not entirely clear. However, its ability to visualize the training dynamics likely makes it useful for researchers working with RNNs.

**Claims And Evidence:**

No

**Claims Explanation:**

As noted under weaknesses above, contributions #2 and #3 are not sufficiently supported by clear and convincing evidence to show the gains of the proposed approach over the baselines, and do not include potentially relevant baselines.

**Requested Changes:**

- The main request is to better support contributions #2 and #3 via quantitative measures and clear quantitative comparison with the baselines to show the gains of the proposed approach.
- Provide full comparison with M-PHATE
- It would be nice to see how such analysis works in NLP tasks (not critical)

---

> ### Author Response · Authors · 2025-12-12
> **Response sec.1**
>
> We thank the reviewer for the detailed and thoughtful feedback. The comments significantly helped us clarify both the conceptual contributions of MM-PHATE and the empirical evidence supporting them. Below, we address each concern point-by-point.
>
> 1. Quantitative evaluation and comparison to baselines:  “There is no sufficient quantitative evaluation… contributions #2 and #3 are not sufficiently supported.”
>
> Response: We appreciate this important point, and in the revised manuscript, we have added a new analysis to quantify how well MM-PHATE preserves local neighborhoods.
>
> In this analysis, we compute intra-step and inter-step k-Nearest Neighbor (k-NN) agreement—measuring, respectively, how well a method preserves local geometry within each epoch and along each hidden unit’s temporal trajectory—and evaluate all methods on a shared subset of the data where neighborhood comparisons are well-defined.
>
> These results show that:
>
> >MM-PHATE achieves the highest neighborhood preservation across almost all $k$s, demonstrating strong fidelity to the intrinsic geometry of hidden-state trajectories.
>
> >PCA and t-SNE degrade significantly as $k$ increases, consistent with their sensitivity to amplitude and global variance distortions.
>
> >UMAP, Isomap, and LLE also underperform MM-PHATE, particularly in cases when either temporal correlations are important, or $k$ is large.
>
> We have added a summary table in the main manuscript (Section 5.2.1) and a supplementary figure illustrating these trends. These quantitative results directly strengthen Contribution #2 regarding preservation of community and temporal structure. This also supports Contribution #3 in that the distinct training-phase changes we observed in the embedding are real structures preserved from the hidden states. Together with the information-processing analyses described in Response 3, this addresses the concern that Contributions #2 and #3 lacked sufficient quantitative support.
>
> 2. Comparison with M-PHATE: “Comparison with the precursor work (M-PHATE) is limited.”
>
> Response: We appreciate the reviewer highlighting this and M-PHATE is indeed a relevant method to compare. We now added a qualitative comparison (Appendix A.4) that shows the components identified by M-PHATE applied to the same data. With that, we would like to clarify that such an evaluation must be limited to qualitative comparison, as the two methods embed different objects in distinct geometric spaces, which hinders numerical comparison. In particular:
>
> >M-PHATE embeds units × epochs, but cannot incorporate within-epoch time steps.
>
> >MM-PHATE embeds units × time steps × epochs simultaneously through a multislice diffusion operator that couples these axes.
>
> >This is not merely “more data”: adding the time dimension changes the neighborhood graph, adaptive kernel bandwidths, diffusion paths, and stationary geometry of the embedding.
>
> Thus the two embeddings inhabit fundamentally different multislice structures, and forcing a one-to-one numerical comparison would be misleading.
>
> 3. Justification of information-processing and compression phases: “Claim #3 is not sufficiently formalized or justified.”
>
> Response: We fully agree that clarifying the meaning of representation “phases” and the structure captured by MM-PHATE is essential. In the revision, we added Sections 5.1, 5.2.1-5.2.2, which explain:
>
> >MM-PHATE preserves both local and global geometry (as supported by the preservation analysis in Table 1 and Sec. 5.2.1), temporal continuity, and correlations across time-step trajectories (Section 5.2.2).
>
> >MM-PHATE minimizes the impact of global amplitude or variance scaling, which can distort the embeddings produced by methods like PCA and t-SNE (as illustrated in Fig. 2 with the Hopf system), allowing for a more accurate representation of the underlying dynamics rather than trivial amplitude drift.
>
> >In our framework, intra-step entropy quantifies the geometric dispersion of hidden units in MM-PHATE space at a fixed epoch and time-step: increases correspond to expansion or diversification of representations across units, whereas decreases reflect compression or stabilization into a more compact manifold. Inter-step entropy is computed per unit across time-steps within an epoch and measures how strongly that unit differentiates among positions in the sequence; higher inter-step entropy indicates sustained temporal selectivity, while decreases correspond to a collapse toward temporally invariant responses.

---

> ### Author Response · Authors · 2025-12-12
> **Response sec.2**
>
> In response to reviewer fw4a’s suggestion, to validate that MM-PHATE reveals true underlying structure rather than artifacts, we added a controlled dynamical-systems experiment using the Hopf bifurcation (see Fig. 2 and Sec. 5.1), which shows that MM-PHATE cleanly recovers the Hopf bifurcation and emergent limit-cycle geometry.
>
>
> The Hopf bifurcation example provides strong evidence of correctness since it exhibits a known analytical transition that we can explore in the embedding. Specifically, the bifurcation has three main epochs in its temporal evolution:
>
> >Pre-bifurcation: The system has a stable fixed point (spiral sink)
>
> >At bifurcation: The fixed point loses stability as eigenvalues cross the imaginary axis
>
> >Post-bifurcation: A stable limit cycle emerges around the now-unstable fixed point
>
>
> In this controlled setting (Fig. 2), we found that running different embeddings produced very different interpretations of the system dynamics. In particular:
>
> >PCA and t-SNE distort the topology of the system because the radius of the oscillation grows with μ. These methods are dominated by amplitude changes and obscure each unit's overall temporal trajectory.
>
> >In contrast, MM-PHATE removes amplitude distortions and correctly recovers the limit-cycle geometry as a single coherent orbit, even as the raw oscillatory radius expands significantly.
>
> >MM-PHATE identifies the bifurcation point, showing a transition from a point attractor to a periodic manifold.
>
> This experiment demonstrates that MM-PHATE preserves true underlying dynamical structure even when other methods fail due to scale or variance distortions. This directly addresses the reviewer's concern:
>
> >It validates that the “phases” observed in training are not visualization artifacts.
>
> >It shows MM-PHATE captures meaningful structure when ground truth is available.
>
> >It illustrates why the multislice diffusion process is uniquely suited to sequential dynamics.
>
>
> We believe this strongly substantiates Contribution #3.
>
> 4. Dataset diversity: “Only two datasets are used; NLP tasks would be interesting.”
>
> Response: We appreciate this suggestion. Applying MM-PHATE to large-scale NLP models is an exciting direction for future work, but is beyond the scope of the current paper.  Our addition of the Hopf bifurcation analysis offers a clean, widely interpretable benchmark that complements the two real sequential datasets.
>
> NOTE: For ease of review, in the revised manuscript we use color coding: purple text in the main body denotes entirely new analyses and results, teal text indicates substantially rephrased or clarified material, and appendix sections with yellow-highlighted titles correspond to newly added content.

---

### Author Response · Authors · 2026-01-19
**Paper revision note**

Dear all,

We had discussed with the Editor an extension for the revision, however a recent response from a reviewer indicates that our paper is marked as "done" which might prevent reviewers from looking at the paper revisions and review responses that we have provided. I wanted then to update here that those have all been provided back in November and are ready for the reviewers to look at, in case the other reviewers similarly did not see the posted responses.

Thank you for your efforts as Editors/Reviewers for handling our manuscript,

Best

-The authors

---

### Decision · Action_Editor_G2dG · 2026-01-26

**Recommendation:** Accept with minor revision

**Audience:**

Yes

**Audience Explanation:**

All reviewers agree.

**Claims And Evidence:**

Yes

**Claims Explanation:**

All three reviewers initially indicated "No". After receiving the author responses, two reviewers were able to accept the claims, provided some extra experiments and extra experimental details are provided. A third reviewer remains unconvinced, also due to lack of experimental details.

I strongly suggest the authors use all of the comments of the reviewers to improve the experiments in the revision. I will check the resubmission for changes to the experiments section.

I copy the relevant comments / suggestions included in the final recommendations of all three reviewers below:

**Recommendation:**

"I appreciate the additional experiment with the Hopf bifurcation. The use of this experiment provides some evidence that the method is indeed showing what it is designed for. I would like to see the following extensions of this experiment:

Additional systems and bifurcations.
Warping type perturbations of a system.
Can the method distinguish these two types of perturbations when they are both present? See e.g. Sagodi (2025), Chen (2024), Friedman (2025).
- Sagodi, A. and Park, I.M., 2025. Dynamical archetype analysis: Autonomous computation. arXiv preprint arXiv:2507.05505.
- Chen, R., Vedovati, G., Braver, T. and Ching, S., 2024. DFORM: Diffeomorphic vector field alignment for assessing dynamics across learned models. arXiv preprint arXiv:2402.09735.
- Friedman, R., Moriel, N., Ricci, M., Pelc, G., Weiss, Y. and Nitzan, M., 2025. Characterizing Nonlinear Dynamics via Smooth Prototype Equivalences. arXiv preprint arXiv:2503.10336.

Until the method is shown to work in multiple systems and bifurcations or (preferably) theoretical guarantees are given for its performance, such claims "MM-PHATE captures the local geometric structure of hidden states across units × time steps × epochs" should rather highlight for which specific case instead of such a universal.

Overall, the method would greatly benefit from a quantitative aspect of measuring temporal-geometric structure in a system, rather than one solely based on a low-dimensional representation that requires visual assessment. Such visual assessment is not sufficiently objective. ``MM-PHATE exposes a clear qualitative transition in how within-sequence dynamics are organized across epochs (Fig. 7a)." I disagree, I do not see a clear qualitative transition."

**Recommendation:**

"I continue to believe that the data-sets used in this paper are too obscure to lead to significant community interest. However given the authors have promised to include more experiments, I think a revised version of this paper can be accepted. "